# Dependence-Aware Label Aggregation for LLM-as-a-Judge via Ising Models

**Krishnakumar Balasubramanian** [1][2]  **Aleksandr Podkopaev** [1]  **Shiva Prasad Kasiviswanathan** [1]

## Abstract

Large-scale AI evaluation increasingly relies on aggregating binary judgments from $K$ annotators, including LLMs used as judges. Most classical methods, e.g., Dawid-Skene or (weighted) majority voting, assume annotators are conditionally independent given the true label $Y \in \{0, 1\}$, an assumption often violated by LLM judges due to shared data, architectures, prompts, and failure modes. Ignoring such dependencies can yield miscalibrated posteriors and even confidently incorrect predictions. We study label aggregation through a hierarchy of dependence-aware models based on Ising graphical models and latent factors. For class-dependent Ising models, the Bayes log-odds is generally quadratic in votes; for class-independent couplings, it reduces to a linear weighted vote with correlation-adjusted parameters. We present finite-$K$ examples showing that methods based on conditional independence can flip the Bayes label despite matching per-annotator marginals. We prove separation results demonstrating that these methods remain strictly suboptimal as the number of judges grows, incurring nonvanishing excess risk under latent factors. Finally, we evaluate the proposed method on three real-world datasets, demonstrating improved performance over the classical baselines.

## 1. Introduction

Large-scale evaluation of modern AI systems increasingly relies on *aggregating binary judgments* from multiple annotators that are predominantly LLMs used as judges. Given an item with unknown label $Y \in \{0, 1\}$ and $K$ noisy votes $J = (J_1, \ldots, J_K) \in \{0, 1\}^K$, the core statistical

problem is to infer $Y$ from $J$. Many classical aggregators such as majority vote, weighted majority vote, and Dawid-Skene (DS) (Dawid & Skene, 1979) estimators are built around the *conditional independence* (CI) assumption: they treat annotators as independent voters given $Y$, i.e., $P(J|Y) = \prod_{j=1}^{K} P(J_j|Y)$. Under this assumption, the weighted majority vote estimator is Bayes-optimal, and the DS estimator provides a practical approach for estimating the weights using the popular Expectation-Maximization algorithm, subject to appropriate identifiability conditions.

The CI assumption is however increasingly mismatched to contemporary evaluation pipelines, especially for LLM judges, where correlations arise from shared pretraining corpora, architectures, prompts, and failure modes (Goel et al., 2025; Kim et al., 2025; Wenger & Kenett, 2025). Dependence is not a benign nuisance: it can fundamentally change the information content of the votes. When judges are redundant or co-vary, aggregation approaches based on the CI assumption (which we refer to as *CI-predictors* and which are typically based on weighted majority voting) can *over-count agreement* and yield systematically miscalibrated predictions. In Appendix C, we demonstrate this through a three-annotator example: when the dependence is captured via an Ising model explained shortly, the posterior prediction for the labels (assuming CI) predicts the opposite label with near-certainty, even given *correct* per-annotator marginals. This illustrates a key insight: misspecified dependence structure can dominate correct marginal modeling.

In this work, we revisit label aggregation through the lens of *Ising graphical models*[1], i.e., quadratic Markov random fields of the form

$$P(J \mid Y = y) \propto \exp\Big( J^\top h^{(y)} + J^\top W^{(y)} J \Big),$$

for $J \in \{0, 1\}^K$, $y \in \{0, 1\}$; see Figure 1 for a graphical model illustration. Here $h^{(y)} \in \mathbb{R}^K$ is a vector of *class-$y$ local fields* capturing per-judge bias/strength: $h_j^{(y)}$ controls how likely judge $j$ is to output 1 under class $y$ (holding other votes fixed). The symmetric matrix $W^{(y)} \in \mathbb{R}^{K \times K}$ (with zero diagonal) is the *class-$y$ coupling* matrix encoding

---
[1]Amazon Web Service, California, USA [2]Department of Statistics, University of California, Davis, USA. Correspondence to: Krishnakumar Balasubramanian <kbala@ucdavis.edu>.

*Proceedings of the 43rd International Conference on Machine Learning*, Seoul, South Korea. PMLR 306, 2026. Copyright 2026 by the author(s).

---
[1]We note that the multiclass setting can be handled by working with Potts models (Rozikov, 2022).

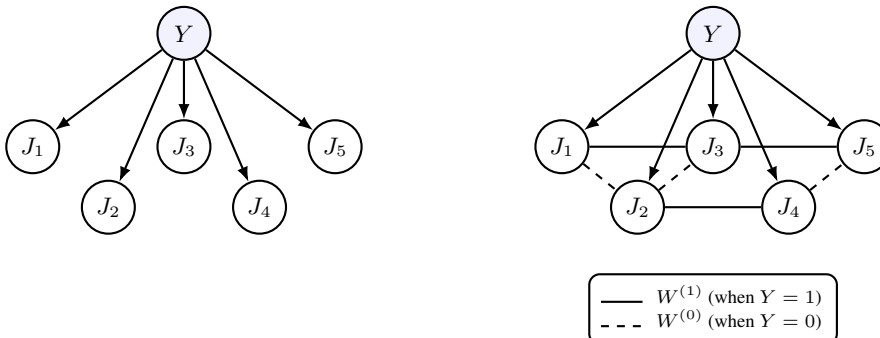

*Figure 1.* **Graphical models for LLM-as-a-judge.** Conditional independence (CI); *(left)*: judges are independent given $Y$ (represented by lack arrows connecting the Judge LLMs). Class-dependent Ising; *(right)*: judges exhibit pairwise dependence whose pattern can change with the label ($W^{(0)} \neq W^{(1)}$), enabling class information to affect directly the correlations among judges.

pairwise dependencies: $W_{jk}^{(y)} > 0$ encourages judges $j$ and $k$ to co-vote 1, while $W_{jk}^{(y)} < 0$ discourages co-voting and captures antagonistic or compensatory behavior. In LLM-as-a-judge settings, $h^{(y)}$ reflects each judge's label-conditional tendency to vote for class 1, while $W^{(y)}$ represents persistent agreement/disagreement patterns induced by shared, training data, architectures, and failure modes.

Building on this representation, we introduce the model hierarchy in Figure 2. The most expressive model is the *class-dependent Ising model*, which allows class-specific interactions ($W^{(0)} \neq W^{(1)}$) and yields Bayes log-odds that are generally *quadratic* in the votes. An important special case is the *class-independent Ising model* where couplings are shared across classes ($W^{(0)} = W^{(1)}$): in this case, the quadratic terms cancel in the likelihood ratio and the Bayes log-odds reduce to a *linear* weighted vote in $J$. Importantly, dependence still matters—correlations are absorbed into correlation-corrected weights and intercepts—allowing this model to discount redundant agreement while retaining the simplicity of a linear aggregation rule.

### 1.1. Our Contributions

- **Dependence-aware Model Hierarchy.** In Section 2, we formalize the model hierarchy spanning CI model, class-independent Ising model (shared interactions), and class-dependent Ising model (class-specific interactions). We derive the exact Bayes log-odds for class-dependent Ising model which is quadratic in votes, and show that under class-independent Ising model, the Bayes rule reduces to a linear weighted vote with dependence-adjusted parameters.

- **Separation Results.** In Section 3, we establish a sharp separation result under the class-dependent Ising model. We show that there exist regimes where each judge is better than random, yet the CI-predictor still makes a constant fraction of errors by misinterpreting correlated "wrong-mode" agreement as strong evi-

dence. In contrast, the Bayes-optimal predictor exploits the dependence structure and achieves vanishing error as the number of judges grows, creating non-vanishing risk separation between the two methods.

- **Relation to Factor Models.** In the crowdsourcing literature, another way to model judge dependence is via *latent-factor model* with a shared random effect $Z$ (independent of $Y$) such that judges are conditionally independent only given $(Y, Z)$: $P(J \mid Y, Z) = \prod_{j=1}^{K} P(J_j \mid Y, Z)$ (Whitehill et al., 2009; Xu et al., 2024). We show that the CI-predictor which ignores $Z$ can remain strictly suboptimal when data are generated from such factor model, *even as $K \to \infty$*. For weak coupling, we show that latent-factor models induce an approximately low-rank Ising structure, placing them between CI and class-dependent Ising model. The details are deferred to Appendix D.3.

- **Experimental Validation.** We evaluate the proposed aggregation method on three real-world tasks: relevance, toxicity, and summarization evaluations, using ten judge models: Claude Opus 4.5, Claude Sonnet 4.5, Claude Haiku 4.5, OpenAI gpt-oss-120b, OpenAI gpt-oss-20b, Llama 4 Maverick 17B Instruct, Llama 4 Scout 17B Instruct, Llama 3.3 70B Instruct, DeepSeek V3.2, and DeepSeek-R1. In Appendix G, we include synthetic simulations that illustrate our theoretical separation results.

We defer a discussion placing our work in the context the larger literature on LLM-as-a-judge, unsupervised label aggregation and crowdsourcing to Appendix A.

## 2. Dependent LLMs-as-a-Judge Models

Suppose that we observe $n$ independent items. Item $i$ has an unobserved label $Y_i \in \{0, 1\}$, with prior $P(Y_i = 1) = \pi \in (0, 1)$, and is annotated by $K$ judges producing a binary vote vector $J_i = (J_{i1}, \ldots, J_{iK}) \in \{0, 1\}^K$. We assume items

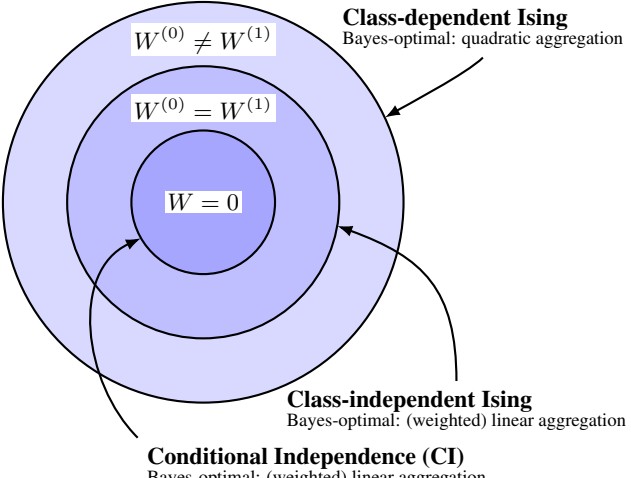

*Figure 2.* **Model hierarchy via set inclusion:** Conditional Independence (CI) $\subset$ Class-independent Ising $\subset$ Class-dependent Ising.

are independent and model *within-item dependencies* among judges via Ising (pairwise MRF) distribution conditional on the label: $\mathrm{P}(\{J_i, Y_i\}_{i=1}^n) = \prod_{i=1}^n \mathrm{P}(Y_i)\, \mathrm{P}(J_i \mid Y_i)$. This makes the label inference problem for each item separable given model parameters[2].

In LLM-as-a-judge pipelines, the within-item dependencies arise due to shared pretraining data, architectures, and prompt templates (leading to label-independent redundancy and shared failure modes), while in other settings dependence itself can be label-dependent (e.g., certain classes trigger common hallucination patterns or refusal behaviors), motivating two practically distinct regimes: a *class-dependent* structure $W^{(0)} \neq W^{(1)}$ that allows correlation patterns to change with the true label and a *class-independent* interaction structure $W^{(0)} = W^{(1)}$ that captures persistent correlations across all items.

## 2.1. Class-dependent Ising Model

In the most general form, the class-conditional distribution of the $K$ votes is allowed to differ across labels: for each $y \in \{0, 1\}$, we have

$$
\mathrm{P}(J_i \mid Y_i = y)
$$
$$
= \frac{1}{Z^{(y)}} \exp\left( \sum_{j=1}^K h_j^{(y)} J_{ij} + \frac{1}{2} \sum_{j \neq k} W_{jk}^{(y)} J_{ij} J_{ik} \right), \quad (1)
$$

where $h^{(y)} \in \mathbb{R}^K$ are class-$y$ local fields (biases), $W^{(y)} \in \mathbb{R}^{K \times K}$ is a symmetric coupling matrix with zero diagonal, and $Z^{(y)}$ is the corresponding partition function. This model captures the possibility that judges are correlated *even af-*

---

[2]The literature on Ising models use spins: $\pm 1$. Our inference algorithm is unchanged under the bijection $X_{ij} = 2J_{ij} - 1$; we keep $Y, J \in \{0, 1\}$ throughout for consistency.

*ter conditioning on the label* where the strength/pattern of correlation may itself depend on the class.

**Bayes-optimal Posterior.** For a single item, we write $J = (J_1, \ldots, J_K)$. The Bayes' rule gives posterior log-odds:

$$
\Lambda(J) := \log \frac{\mathrm{P}(Y = 1 \mid J)}{\mathrm{P}(Y = 0 \mid J)}
$$
$$
= \log \frac{\pi}{1 - \pi} + \log \frac{\mathrm{P}(J \mid Y = 1)}{\mathrm{P}(J \mid Y = 0)}
$$
$$
= \log \frac{\pi}{1 - \pi} + \sum_{j=1}^K \Delta h_j\, J_j + \frac{1}{2} \sum_{j \neq k} \Delta W_{jk}\, J_j J_k + \Delta_Z,
$$

where $\Delta h_j := h_j^{(1)} - h_j^{(0)}$, $\Delta W_{jk} := W_{jk}^{(1)} - W_{jk}^{(0)}$ and $\Delta_Z := -\log Z^{(1)} + \log Z^{(0)}$. The term $\Delta_Z$ is constant in $J$ (for fixed parameters and fixed $K$) and can be absorbed into the intercept. Since $W^{(y)}$ is symmetric with zero diagonal, one can equivalently rewrite the quadratic term as $\frac{1}{2} \sum_{j \neq k} \Delta W_{jk} J_j J_k = \sum_{1 \leq j < k \leq K} \Delta W_{jk} J_j J_k$. Therefore, the Bayes-optimal predictor takes form:

$$
g^\star(J) = \mathbf{1}\{\Lambda(J) \geq 0\}
$$
$$
= \mathbf{1}\left\{ b_0 + \sum_{j=1}^K a_j J_j + \sum_{1 \leq j < k \leq K} b_{jk} J_j J_k \geq 0 \right\}, \quad (2)
$$

where $a_j = \Delta h_j$, $b_{jk} = \Delta W_{jk}$ and $b_0 = \log \frac{\pi}{1 - \pi} + \Delta_Z$. When $W^{(1)} \neq W^{(0)}$, the optimal decision boundary is *quadratic* in the votes since class information may be present not only in marginal accuracies (fields) but also in *label-dependent correlation structure* (couplings).

## 2.2. Class-Independent Couplings

A common and interpretable special case is one in which the *dependence structure* among judges is shared across both classes but individual biases and accuracies shift with the label. Specifically, we assume

$$
\mathrm{P}(J_i \mid Y_i = y) \quad\quad\quad (3)
$$
$$
= \frac{1}{Z^{(y)}} \exp\left( \sum_{j=1}^K \left( h_j + (y - \tfrac{1}{2})c_j \right) J_{ij} + \frac{1}{2} \sum_{j \neq k} W_{jk}\, J_{ij} J_{ik} \right),
$$

where $h_j$ represents a label-independent baseline field, $c_j$ controls how the label shifts judge $j$'s field, and $W$ is a shared symmetric coupling matrix (with zero diagonal). Here, the normalizer $Z^{(y)}$ may still depend on $y$ since the fields differ across classes.

**Bayes-optimal Posterior.** Under (3), the posterior log-odds simplify substantially:

$$
\log \frac{\mathrm{P}(Y = 1 \mid J)}{\mathrm{P}(Y = 0 \mid J)}
$$
$$
= \log \frac{\pi}{1 - \pi} + \log \frac{\mathrm{P}(J \mid Y = 1)}{\mathrm{P}(J \mid Y = 0)} \qu\quad (4)
$$
$$
= \log \frac{\pi}{1 - \pi} + \sum_{j=1}^K c_j J_j + \Delta_Z,
$$

where $\Delta_Z = -\log Z^{(1)} + \log Z^{(0)}$ is a constant in $J$. The quadratic terms cancel since the coupling matrix is shared across the two classes. Therefore, the Bayes-optimal predictor is a *linear* threshold rule:

$$g^\star(J) = \mathbf{1}\Big\{ \sum_{j=1}^{K} c_j J_j + b_0 \geq 0 \Big\},$$

$$b_0 = \log \frac{\pi}{1-\pi} + \Delta_Z. \tag{5}$$

If one prefers $\pm 1$ labels instead of $\{0,1\}$, one can define centered spins $X_j := 2J_j - 1 \in \{\pm 1\}$. Then $\sum_j c_j J_j = \frac{1}{2}\sum_j c_j X_j + \frac{1}{2}\sum_j c_j$, so the rule remains a weighted vote on $X$ after absorbing $\frac{1}{2}\sum_j c_j$ into the intercept.

If $W \equiv 0$, then the judges are conditionally independent given $Y$ and (3) reduces to a product of Bernoulli marginals: $\mathrm{P}(J \mid Y = y) = \prod_{j=1}^{K} \mathrm{P}(J_j \mid Y = y)$ with $\mathrm{P}(J_j = 1 \mid Y = y) = \sigma(h_j + (y - \frac{1}{2})c_j)$, where $\sigma(t) = (1 + e^{-t})^{-1}$. In this case, the per-judge contribution to the posterior log-odds can be written as an affine function of $J_j$:

$$\log \frac{\mathrm{P}(J_j \mid Y = 1)}{\mathrm{P}(J_j \mid Y = 0)}$$

$$= J_j \Big( \mathrm{logit}(p_j^{(1)}) - \mathrm{logit}(p_j^{(0)}) \Big) + \log \frac{1 - p_j^{(1)}}{1 - p_j^{(0)}},$$

where $p_j^{(y)} := \mathrm{P}(J_j = 1 \mid Y = y)$.

Summing over $j$ recovers the classical CI or Naive-Bayes linear aggregation rule with weights equal to differences of logits; in the parameterization (3), $\mathrm{logit}(p_j^{(1)}) - \mathrm{logit}(p_j^{(0)}) = c_j$. When $W \neq 0$ but is *shared across classes*, correlations do not introduce quadratic terms into the Bayes log-odds (they cancel in (4)), and the Bayes decision remains linear in $J$. Nevertheless, correlations still matter statistically: they change the joint law of $J$ within each class and therefore affect likelihoods, partition functions (hence the intercept $b_0$), and parameter estimation from finite data. In contrast, if the couplings differ across classes ($W^{(1)} \neq W^{(0)}$), then class information is present in the dependence structure and the Bayes decision becomes quadratic as in (2).

For the sake of completeness, we discuss the CI model with asymmetric errors (which leads to weighted majority vote aggregators) in Appendix D.1. We also briefly discuss the similarities and differences with the linear and quadratic discriminant analysis model, that are standard Gaussian models (as opposed to the binary Ising models) in Appendix D.2.

## 3. Separation Results

In this section, we use our model hierarchy to clarify when standard aggregation rules are justified for LLM-as-a-judge

pipelines and when those fail. Weighted majority vote (with sufficiently accurate weights) is Bayes-optimal under conditional independence (CI) Ai et al. (2025, Theorem 1). However, as argued in the introduction and illustrated by our motivating examples (Appendix C), satisfying CI is often implausible for LLM judges due to shared pretraining corpora, similar architectures, reused prompt templates, and common safety/refusal or hallucination failure modes. These dependencies are not merely noise: they can encode item properties (e.g., difficulty or trigger patterns) and the redundancy of evidence across judges.

We show a separation (in terms of risk) between majority voting procedures and the Bayes-optimal predictor under two dependence mechanisms relevant to LLM-as-a-judge: (a) shared latent factors inducing low-rank correlations (Appendix D.3.1) and (b) interaction-driven dependence captured by the Curie-Weiss model, a special case of the Ising model, with strong agreement modes (Section 3.1). These separation results clearly demonstrate the limitations in expressive power of schemes such as majority voting. The proofs are deferred to Appendix F.

### 3.1. Sub-optimality of CI-predictor under Ising Dependence

The risk of a binary aggregator $g$ in our setting is defined as $R(g) := \mathrm{P}(g(J) \neq Y)$. We start by establishing a separation result in terms of risk between a special case of Ising model, namely the Curie-Weiss model, which has been widely used in opinion dynamics (Bahr & Passerini, 1998). Informally, we show that even with infinitely many judges, any CI-predictor remains strictly suboptimal, whereas there exists a Bayes predictor that leverages dependence (quadratic structure) to classify essentially perfectly.

**Theorem 1** (Nonvanishing Bayes vs. CI Separation for Class-conditional Ising Models)**.** *Fix a prior $\mathrm{P}(Y = 1) = \pi \in (0,1)$. For each $K \geq 1$, let $J = (J_1, \ldots, J_K) \in \{0,1\}^K$ denote the $K$ judges' votes for a single item and define the recoded spins $X_j := 2J_j - 1 \in \{-1, +1\}$ and let $M_K := \frac{1}{K}\sum_{j=1}^{K} X_j \in \left\{ -1, -1 + \frac{2}{K}, \ldots, 1 \right\}$. Assume the following class-conditional Curie-Weiss Ising model: there exist constants $0 < \beta_0 < 1 < \beta_1$ such that, conditional on $Y = y \in \{0,1\}$,*

$$\mathrm{P}(X = x \mid Y = y) = \frac{1}{Z_K(\beta_y)} \exp\Big( \frac{\beta_y}{2K} \Big( \sum_{j=1}^{K} x_j \Big)^2 \Big), \quad (6)$$

*for $x \in \{-1, +1\}^K$. Equivalently (writing $x_j = 2j_j - 1$), this is a special case of the $\{0,1\}$-Ising form (1) with $h_j^{(y)} = -2\beta_y \Big( 1 - \frac{1}{K} \Big)$ and $W_{jk}^{(y)} = \frac{4\beta_y}{K}$ $(j \neq k)$, up to an additive constant absorbed into $Z^{(y)}$.*

*Let $g_K^\star$ be the Bayes-optimal predictor under the true model (6). Let $g_K^{\mathrm{ind}}$ be the population CI-predictor that replaces the*

*true joint by the product of true one-dimensional marginals, i.e., $\mathrm{P}^{\mathrm{ind}}(J = j \mid Y = y) := \prod_{r=1}^{K} q_y^{j_r}(1 - q_y)^{1-j_r}$ with $q_y := \mathrm{P}(J_r = 1 \mid Y = y)$ (independent of $r$), and then thresholds the induced posterior $\mathrm{P}^{\mathrm{ind}}(Y = 1 \mid J)$ at $1/2$. Then the following results hold:*

1. **(CI Collapses to the Prior)** *For every $K$ and $y \in \{0, 1\}$, one has $q_y = \frac{1}{2}$. Consequently, $\mathrm{P}^{\mathrm{ind}}(Y = 1 \mid J) = \pi$ for all $J$, so $g_K^{\mathrm{ind}}(J) \equiv \mathbf{1}\{\pi \geq \frac{1}{2}\}$, and $R(g_K^{\mathrm{ind}}) = \min\{\pi, 1 - \pi\}, \forall K$.*

2. **(Bayes Risk Vanishes)** *Let $m_\star = m_\star(\beta_1) \in (0, 1)$ denote the unique positive solution to $m = \tanh(\beta_1 m)$. Then for any fixed threshold $t \in (0, m_\star^2)$, the quadratic statistic test:*

$$\tilde{g}_K(J) := \mathbf{1}\{M_K(J)^2 \geq t\},$$

$$M_K(J) = \frac{1}{K}\sum_{j=1}^{K}(2J_j - 1),$$

*satisfies $R(\tilde{g}_K) \to 0$ as $K \to \infty$. Hence $R(g_K^\star) \to 0$ as $K \to \infty$.*

3. **(Nonvanishing Separation)** *We have:*

$$\lim_{K \to \infty}\left(R(g_K^{\mathrm{ind}}) - R(g_K^\star)\right) = \min\{\pi, 1 - \pi\} > 0.$$

Theorem 1 gives a clean population-level separation between the Bayes-optimal predictor and a CI-predictor when the judges are *dependent* given the label. We consider a setting where, for each label $y \in \{0, 1\}$, the $K$ votes are drawn from a class-conditional Ising model. In the particular Curie-Weiss specialization used in the theorem, the conditional law depends only on the *global magnetization $M_K$* and the dependence strength differs across classes: the negative class is in a *high-temperature* regime ($\beta_0 < 1$), while the positive class is in a *low-temperature* regime ($\beta_1 > 1$).

In this model the two classes have the *same* one-dimensional marginals: for every judge $j$ and both labels $y \in \{0, 1\}$, $\mathrm{P}(J_j = 1 \mid Y = y) = 1/2$. Thus, looking at each judge in isolation provides no information about the label. The only distinguishing signal is in the *dependence structure*: under $Y = 1$ the votes exhibit strong global alignment (many judges tend to agree), whereas under $Y = 0$ they do not. The CI-predictor replaces the true joint likelihood by a product of these one-dimensional marginals. Since the marginals are identical across classes, under CI, the likelihood ratio is identically 1, so the posterior stays equal to the prior $\pi$ for every vote vector $J$. Consequently, the CI-predictor ignores the data entirely and achieves risk $R(g_K^{\mathrm{ind}}) = \min\{\pi, 1 - \pi\}$, independently of $K$.

In contrast, the Bayes-optimal posterior uses the correct class-conditional joint likelihood, which depends on

quadratic (pairwise) interactions among votes and, in this Curie-Weiss case, can be expressed through the statistic $M_K^2$. As $K$ grows, the magnetization concentrates at 0 under the high-temperature class ($Y = 0$) and concentrates near a nonzero value under the low-temperature class ($Y = 1$) (up to a random global sign flip). Therefore, a simple quadratic rule such as $\mathbf{1}\{M_K^2 \geq t\}$ (for any fixed $t$ between 0 and the squared limiting magnetization under $Y = 1$) separates the two classes with vanishing error as $K \to \infty$. Since the Bayes rule is optimal, its risk also converges to 0. Putting the two pieces together, the Bayes-optimal risk goes to zero while the CI risk stays bounded away from zero.

### 3.2. Extension to Informative Marginals

The Curie-Weiss separation example in Theorem 1 uses the zero-field Curie-Weiss model, which is invariant under the global flip $X \mapsto -X$. This symmetry forces $q_0 = q_1 = \mathrm{P}(J_j = 1 \mid Y = y) = \frac{1}{2}$ for both classes, so a CI-predictor that only uses one-dimensional marginals collapses to the prior. While this may look like a "corner case," the underlying failure mechanism is *not*: the CI-predictor cannot exploit class information carried by *dependence structure* (correlation/interaction patterns), and this can remain true even when individual marginals are (slightly) informative.

At a basic level, risks vary continuously with the data-generating law: if $P$ and $Q$ are two joint distributions on $(Y, J)$ and $d_{\mathrm{TV}}(P, Q) := \sup_A |P(A) - Q(A)|$ is their total variation distance, then for any classifier $g$, $|R_P(g) - R_Q(g)| \leq d_{\mathrm{TV}}(P, Q)$, and in particular we have $|R_P^\star - R_Q^\star| \leq d_{\mathrm{TV}}(P, Q)$, because $R_P(g) = P(g(J) \neq Y)$ is the probability of a measurable event, where $R_P^\star$ is Bayes-optimal risk under distribution $P$. Thus, for any fixed (moderate) $K$, the large finite-sample gaps exhibited by the symmetric Curie-Weiss example persist under small perturbations of the class-conditional distributions, including perturbations that move the marginals away from $1/2$.

The next result gives an explicit *asymptotic* variant in which each judge is individually better than random ($q_0 < 1/2 < q_1$), yet the CI risk remains bounded away from zero while the Bayes-optimal risk still vanishes as $K \to \infty$. The key idea is to retain *phase coexistence* in the low-temperature class so that, with constant probability, the judges collectively enter a "wrong-sign" agreement mode that a CI-predictor interprets as decisive evidence for the wrong label, whereas the Bayes-optimal predictor uses a dependence-sensitive statistic (here $|M_K|$) to identify the true class.

**Theorem 2** (Curie-Weiss separation with informative marginals). *Let $Y \in \{0, 1\}$ with $\mathrm{P}(Y = 1) = \pi \in (0, 1)$. For each $K \geq 1$, define spins $X = (X_1, \ldots, X_K) \in \{-1, +1\}^K$ and votes $J_j = (X_j + 1)/2 \in \{0, 1\}$. Let the class-conditional laws of $X$ be Curie-Weiss models with*

*(possibly $K$-dependent) external fields:*

$$P(X = x \mid Y = y)$$

$$= \frac{1}{Z_K^{(y)}} \exp\Big( \frac{\beta_y}{2K} \Big( \sum_{j=1}^{K} x_j \Big)^2 + h_{y,K} \sum_{j=1}^{K} x_j \Big), \quad (7)$$

*for $y \in \{0, 1\}$. Assume parameters satisfy:*

1. *(High-temperature Class) $\beta_0 \in (0, 1)$ and $h_{0,K} \equiv h_0 < 0$ is a fixed negative constant;*

2. *(Low-temperature Class with Weak Symmetry Breaking) $\beta_1 > 1$ and $h_{1,K} = c/K$ with some fixed $c > 0$.*

*Let $M_K := \frac{1}{K} \sum_{j=1}^{K} X_j \in [-1, 1]$ be the magnetization. Let $m_0 \in (-1, 0)$ be the unique solution of the mean-field equation $m_0 = \tanh(\beta_0 m_0 + h_0)$, and let $m_\star = m_\star(\beta_1) \in (0, 1)$ be the unique positive solution of $m_\star = \tanh(\beta_1 m_\star)$. Define*

$$p := \frac{e^{cm_\star}}{e^{cm_\star} + e^{-cm_\star}} = \sigma(2cm_\star) \in \Big( \frac{1}{2}, 1 \Big),$$

$$q_0 := P(J_1 = 1 \mid Y = 0) = \frac{1 + m_0}{2} \in \Big( 0, \frac{1}{2} \Big),$$

$$q_1 := P(J_1 = 1 \mid Y = 1) = \frac{1 + (2p-1)m_\star}{2} \in \Big( \frac{1}{2}, 1 \Big).$$

*Assume additionally that*

$$\frac{1 - m_\star}{2} < q_0 \iff m_\star > 1 - 2q_0. \quad (8)$$

*Let $g_K^\star$ denote the Bayes predictor under the true model (7). Let $g_K^{\mathrm{ind}}$ denote the population CI-predictor that replaces $P(J \mid Y = y)$ by the product of the true marginals $\prod_{j=1}^{K} q_y^{J_j} (1 - q_y)^{1 - J_j}$. Then, the following hold:*

1. *(**Informative Marginals**) Each judge is individually better than random: $q_0 < \frac{1}{2} < q_1$ (equivalently, specificity $1 - q_0 > \frac{1}{2}$ and sensitivity $q_1 > \frac{1}{2}$).*

2. *(**Bayes Risk Vanishes**) For any fixed threshold $t$ satisfying $|m_0| < t < m_\star$, the aggregator $\tilde{g}_K(J) := \mathbf{1}\{|M_K| \geq t\}$ has $R(\tilde{g}_K) \to 0$ as $K \to \infty$. Consequently, $R(g_K^\star) \to 0$.*

3. *(**CI Remains Bounded away from Bayes**) As $K \to \infty$, we have $R(g_K^{\mathrm{ind}}) \to \pi(1 - p)$, and hence $\lim_{K \to \infty} \big( R(g_K^{\mathrm{ind}}) - R(g_K^\star) \big) \geq \pi(1 - p) > 0$.*

**Remark 1** (Continuity viewpoint). *In Theorem 2, the limiting CI error is $\pi(1 - p)$ with $p = \sigma(2cm_\star)$, which varies continuously with the "weak symmetry-breaking" strength $c$. As $c \downarrow 0$, one has $p \to 1/2$ and $q_1 \downarrow 1/2$, recovering the symmetric example; for any $c > 0$, the marginals become informative ($q_1 > \frac{1}{2}$) yet CI still makes errors on a constant*

*fraction $(1 - p)$ of the positive-class items because it thresholds the* sign *of the vote proportion, whereas Bayes exploits the dependence-induced structure (here, large $|M_K|$ regardless of sign). Thus the separation is robust: it is driven by a persistent correlated "wrong-sign" mode, not by the exact equality $q_y = 1/2$.*

The separation results in this section are interesting for two reasons. First, they show that the limitations with the CI assumption are *structural*, not a finite-judge artifact: even as $K \to \infty$, the CI-predictor remain strictly suboptimal because it cannot exploit dependence-induced information or distinguish redundant agreement from independent evidence (Ising interactions). Second, they explain why LLM ensembles can become overconfident for the wrong reasons: if judges share a failure mode, their agreement should be discounted rather than counted multiple times, which CI weighting cannot do. Practically, this suggests evaluation pipelines should not rely solely on per-judge accuracies or majority counts, but should monitor and model dependence among judges (e.g., residual correlations after accounting for label uncertainty). When dependence is present, our results motivate moving up the hierarchy: from CI to various Ising models, to improve accuracy, calibration, and uncertainty for downstream decisions.

## 4. Experimental Evaluation

For all experimental results, we use the Expectation-Maximization algorithm for posterior label prediction. We defer the details to Appendix E. Simulation results comparing weighted (with the weights estimated by EM algorithm) and uniform majority voting are provided in Appendix G.1. Simulation results regarding CI-predictor and factor/Ising model predictors are provided in Appendix G. Below, we provide experiments on real-world datasets.

### 4.1. Real World Datasets

We consider LLMaaJ label aggregation in three tasks: relevance, toxicity, and summarization assessment. We use the following models as judges: (1) Claude Opus 4.5, (2) Claude Haiku 4.5, (3) Claude Sonnet 4.5, (4) OpenAI gpt-oss-120b, (5) OpenAI gpt-oss-20b, (OpenAI, 2025), (6) Llama 4 Maverick 17B Instruct, (7) Llama 4 Scout 17B Instruct, (8) Llama 3.3 70B Instruct, (9) DeepSeek V3.2 and (10) DeepSeek-R1 (DeepSeek-AI, 2025). For all models, the temperature parameter is set to zero. The prompts used in our evaluations are deferred to Appendix H.3.

**Relevance.** We use the WikiQA dataset (Yang et al., 2015) to evaluate query-passage relevance classification, a task which arises in the context of retrieval-augmented generation (RAG). The dataset consists of natural language questions paired with multiple sentences extracted from

| Dataset | Class-Dep. Ising | Class-Indep. Ising | CI-WMV | CI-UMV |
|---|---|---|---|---|
| Relevance | $\mathbf{0.912} \pm 0.007$ | $0.899 \pm 0.005$ | $0.820 \pm 0.004$ | $0.804 \pm 0.003$ |
| Toxicity | $\mathbf{0.792} \pm 0.004$ | $0.780 \pm 0.006$ | $0.694 \pm 0.004$ | $0.695 \pm 0.003$ |
| Summarization | $\mathbf{0.801} \pm 0.004$ | $0.806 \pm 0.005$ | $0.737 \pm 0.005$ | $0.561 \pm 0.005$ |

*Table 1.* Test accuracy for the various methods on the three datasets. Numbers are averages over 20 trials. CI-WMV and CI-UMV correspond to weighted and uniform majority vote both of which operator under conditional independence assumptions. Class-Dep. and Class-Indep. refer to class-dependent and class-independent Ising models, respectively.

Wikipedia. Each sentence is annotated with a binary label which indicates whether it correctly answers the question at hand. To construct inputs suitable for the relevance classification, we collect all the sentences associated with each question and concatenate them into a single text passage. The resulting passage is labeled relevant to the question if and only if at least one of the original sentences is labeled as a correct answer; see examples in Appendix H.1. In our evaluation, we use all available splits for the original dataset, resulting in approximately 3000 evaluation instances.

**Toxicity.** We use the Jigsaw Unintended Bias in Toxicity Classification dataset (Adams et al., 2019) for toxicity classification. We use comments from the private leaderboard test set, filtering for those with at least five human annotators and a minimum length of 100 characters. To ensure balanced representation across all toxicity levels, we use stratified sampling based on comment toxicity scores, i.e., the fraction of annotators who marked a comment as toxic. We randomly sample 1000 comments from each of the four toxicity score buckets: $[0, 0.25)$, $[0.25, 0.5)$, $[0.5, 0.75)$, $[0.75, 1]$, and label a comment as toxic (positive class) if at least half of the annotators marked it as toxic. Additional preprocessing steps are deferred to Appendix H.2.

**Summarization.** We use the CNN/DailyMail news dataset (Hermann et al., 2015; See et al., 2017) for summarization assessment. The original dataset contains news articles along with short author-written summaries. For binary summarization assessment, we use a preprocessed variant of the dataset from Arize Phoenix summarization benchmark. This benchmark augments a subset of the original articles and summaries with synthetically generated incorrect summaries designed to resemble correct ones while containing factual inconsistencies. The dataset contains 1100 instances with approximately the same number of correct and incorrect summaries.

We conducted two experiments on those datasets: (i) studying the effect of varying the number of training samples $n$ while keeping the number of judges fixed ($K = 6$), and (ii) studying effect of varying $K$ while keeping the number of training samples $n$ fixed at roughly 10%–20% of the overall dataset. We compared the class-independent Ising-predictors, class-dependent Ising-predictors, and CI-predictors (specifically, the weighted majority vote procedure resulting from the asymmetric models described in Appendix D.1). For these experiments, we randomly sampled the training data and the judges and report the average test accuracy (and standard error) over 20 trials in Figure 3.

From Figure 3, we note that the two Ising predictors invariably outperform the CI-predictor once the number of training samples and the number of judges exceed a modest threshold (and in some regimes even with fewer training samples). As a summary, in Table 1, we also report the result of comparing the aforementioned procedures, along with the uniform majority voting procedure, when all 10 judges are used and when the maximum amount of training data is used for each dataset (i.e., 2500 samples for relevance, 3000 samples for toxicity, and 1000 samples for summarization tasks). We notice that Ising models outperform (weighted) majority voting procedures. In particular, as we have large number of training (unsupervised) samples relative to the number of model parameters, the class-dependent Ising model outperforms the class-independent Ising model, illustrating the benefit of moving up the hierarchy of proposed models.

The above experiments demonstrate the practical value of explicitly modeling judge dependence: when correlations are present, interaction-aware aggregation leverage the additional structure among the pool of judges and deliver consistently lower error than any conditional-independence weighting which is based only on marginals. Additional interpretation of coupling matrices are provided in Appendix B.

## 5. Limitations and Conclusion

While our separation results are stated in terms of the Bayes-optimal posterior under the true generative model, in practice one must rely on computational procedures such as generalized EM with approximate E-steps and surrogate M-steps, e.g., pseudo-likelihood. A key next step is to extend the theory from the population Bayes rule to these algorithms: establishing conditions under which the learned parameters and induced posteriors converge to the correct decision rule or, at minimum, inherit the same separation from conditional-independence baselines.

Beyond parameter estimation, our framework motivates a hypothesis-testing view of LLM-as-a-judge evaluation:

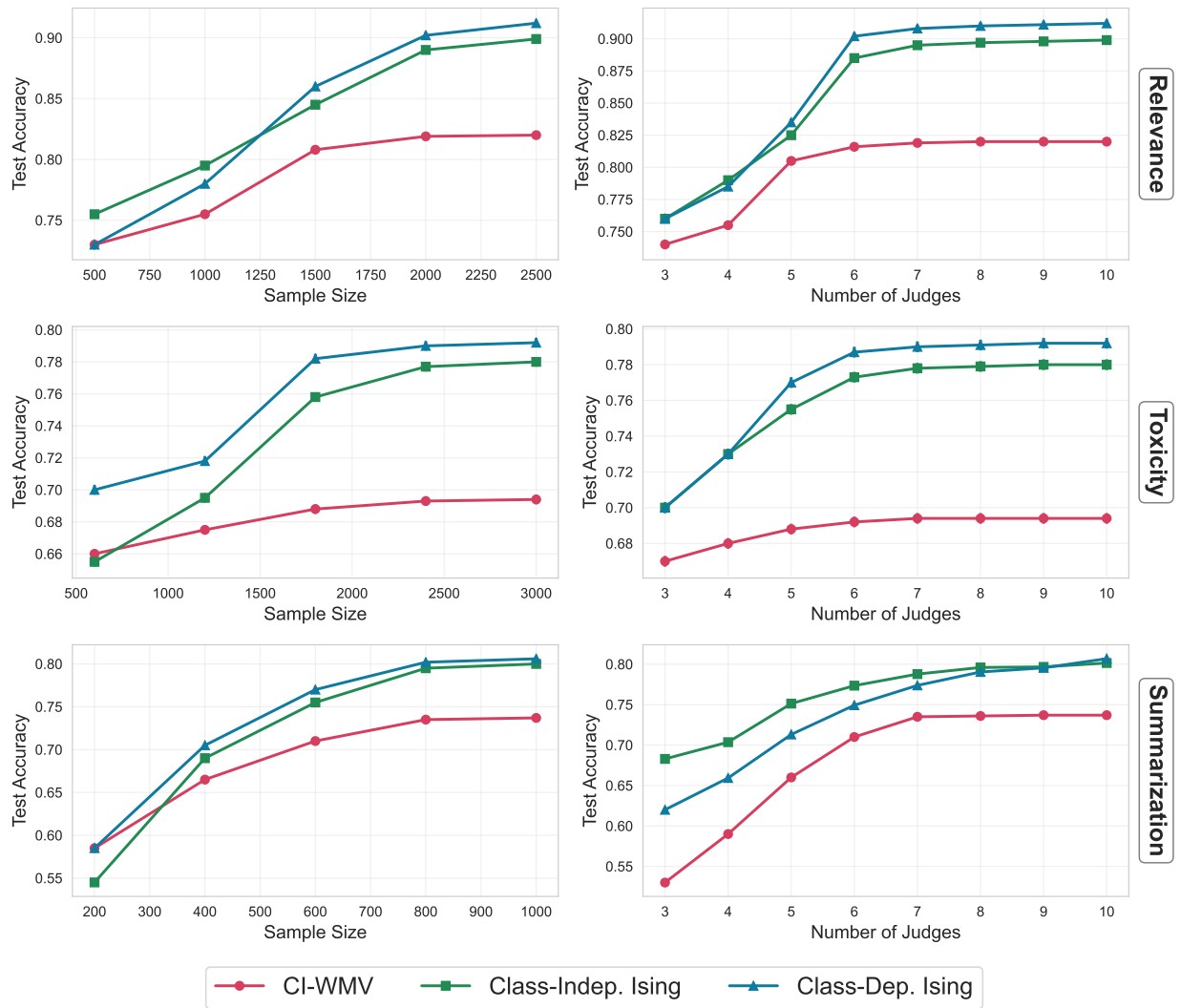

*Figure 3.* Effect of varying the number of training samples (left) and number of judges (right) on the test accuracy. Top, middle and bottom rows correspond respectively to Relevance, Toxicity and Summarization datasets. The standard errors are of small width, although they are plotted.

before fitting more expressive dependence models, practitioners can test whether class-conditional independence is plausible by checking for residual correlations among judges after accounting for label uncertainty, or by comparing CI and Ising/factor-model pseudo-likelihoods on holdout items.

From a practical standpoint, the hierarchy presented in this paper provides actionable guidance: one may start with CI as a cheap baseline and then move to latent-factor (low-rank) dependence if correlations appear global or prompt-driven. When pairwise agreement patterns are strong or class-dependent, one may escalate to class-independent or class-dependent Ising model. Simple diagnostics—estimated coupling strength, stability across random initializations, and predictive calibration on small labeled validation sets—can help determine the appropriate level of dependence model-

ing and prevent overfitting. Such considerations may make dependence-aware aggregation a reliable component of real-world evaluation pipelines.

## Impact Statement

This work provides dependence-aware aggregation methods and theoretical guarantees that improve the reliability and calibration of LLM-as-a-judge evaluations, reducing the risk of overconfident conclusions driven by correlated failure modes rather than independent evidence. By clarifying when standard independence-based aggregation is structurally unreliable and offering practical alternatives, our results can make benchmarking and model selection more robust, while also highlighting the need to audit and report judge dependence to prevent misleading evaluations.

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

## A. Related Works

The LLM-as-a-judge literature uses LLMs to approximate human preferences in open-ended evaluation, including MT-Bench/Chatbot Arena (Zheng et al., 2023) and prompt-based evaluators such as G-Eval (Liu et al., 2023), as well as benchmark suites like AlpacaEval and debiasing methods for judge preferences (e.g., length control) (Dubois et al., 2024; Li et al., 2025). See, for example, Gu et al. (2024) for a survey and Lee et al. (2025); Chen et al. (2026) for some recent works.

In the context of binary classification, however, label aggregation has been carried out using majority voting and its weighted variants much before their use in LLM-as-a-judge applications. Our focus is on the case of unsupervised label aggregation in which the unknown true label is treated as a latent variable and EM algorithm is used to infer it. The approach was proposed by Dawid & Skene (1979) and analyzed by Donmez et al. (2010); Raykar et al. (2010); Berend & Kontorovich (2015); Gao et al. (2016). A large literature on *crowdsourcing* extends this template by enriching the annotator/item structure while typically retaining conditional independence of votes given latent variables, e.g., modeling annotator expertise and item difficulty (GLAD) (Whitehill et al., 2009; Xu et al., 2024; Davani et al., 2022), multi-dimensional annotator effects (Welinder et al., 2010), and incorporating item features jointly with label aggregation (also called as "learning from crowds") (Raykar et al., 2010). Bayesian variants and alternatives include Bayesian classifier combination (Kim & Ghahramani, 2012) and models designed to detect spammers/adversaries such as MACE (Hovy et al., 2013).

Motivated by economics and customer study literatures (Chen et al., 2023; Prelec et al., 2017), recently Ai et al. (2025) study LLM aggregation beyond majority vote by proposing *optimal weight* using first-order statistics and *Inverse Surprising Popularity* using second-order cross-agent conditional response statistics with theoretical improvements and empirical gains. But their main guarantees effectively assume oracle-quality second-order information $P(J_i \mid J_j)$ for Judges $i$ and $j$ (treated as "accurate" before finite-sample estimation), which can be sample-hungry to estimate and can be brittle under distribution shift or latent-difficulty confounding that changes apparent correlations. Furthermore, they assume the judge labels are conditionally independent given the true label.

While the aforementioned works document biases and variance in LLM judging, they typically aggregate multiple judge calls using independence-based heuristics; our focus is on explicitly modeling and exploiting *dependence* among judges (e.g., shared prompts or shared failure modes) via Ising structure.

Initial steps towards handling dependencies have been taken by Jaffe et al. (2016) and Shaham et al. (2016). In particular, these works study the case of hierarchically dependent judges/annotators, a relaxation of fully conditionally independent judges. While appropriate for certain crowdsourcing applications (e.g., workers clustered by source or organization), this assumption is poorly aligned with LLM-as-a-judge settings: dependencies among LLM judges are rarely tree-structured or nested, and instead arise from overlapping pretraining data, shared architectures, and reused prompting templates that induce *dense, non-hierarchical* correlations and shared failure modes. As a result, hierarchical-dependence models can miss the dominant correlation patterns in practice and provide limited guidance for correcting over-counted agreement in modern LLM evaluation pipelines. Mazzetto et al. (2021) studied a setting where not all the judges are conditionally independent, however a subset of them are. Their approach, however, is fundamentally semi-supervised meaning that they required some amount of labeled data. We also remark that in the fully-supervised setting, several works (Steinhardt et al., 2016; Kleindessner & Awasthi, 2018) considered adversarial annotators which maybe arbitrarily correlated. Compared to these works, our focus is in the purely unsupervised setting.

Finally, we remark that recent works have used additional human (supervised) annotated labels to improve the unknown label inference via the framework of prediction powered inference; see, for example (Angelopoulos et al., 2023a;b; Boyeau et al., 2025). While we work under the purely unsupervised setting, we remark that our proposed models integrate seamlessly with the aforementioned framework.

## B. Interpreting the Estimated Couplings

The estimated couplings corresponding to the results in Section 4 contain rich diagnostic information beyond their role in improving aggregation accuracy.

**Are Claude Models More Coupled to Each other than to Others?**   Yes, unambiguously. From Table 2, we note that the Anthropic Sonnet–Opus coupling is the single strongest pair in all three datasets ($W = 0.568$ on Relevance, $0.692$ on Toxicity, $0.415$ on Summarization). By contrast, the strongest Anthropic–DeepSeek coupling is substantially weaker ($W = 0.478,\ 0.283,\ 0.026$ on the same tasks), and the strongest Anthropic–Meta coupling is weaker still in two of three

tasks. The pattern is consistent: same-provider models share more residual dependence than cross-provider pairs. This likely reflects shared training data, RLHF procedures, and architectural choices within each model family.

*Table 2.* Average Ising coupling $W$ by family relationship ($K = 10$ judges).

| Dataset | Anthropic–Anthropic | Meta–Meta | DeepSeek–DeepSeek | Cross-family |
|---|---|---|---|---|
| Relevance | 0.400 | 0.397 | 0.475 | 0.375 |
| Toxicity | 0.441 | 0.488 | 0.572 | 0.296 |
| Summarization | 0.316 | 0.152 | 0.098 | 0.111 |

**Does Coupling Strength Vary by Task?** Dramatically. On Relevance, couplings are uniformly high ($W \in [0.27, 0.57]$) and the within/across-family gap is modest ($1.28\times$). On Toxicity, the gap widens ($1.70\times$) and absolute coupling decreases. On Summarization, couplings are sparse and heterogeneous ($W \in [0.005, 0.415]$) with the largest within/across gap ($2.09\times$). This variation reflects the nature of each task: relevance judgments are relatively objective (judges agree or disagree for similar reasons regardless of provider), while summarization assessment involves more subjective criteria where provider-specific biases in what constitutes a "good summary" create family-specific failure modes.

The class-dependent structure also varies by task. On Summarization, $\|W^{(1)}\|_F / \|W^{(0)}\|_F = 3.75$: judges are nearly independent on bad summaries but strongly coupled on good ones. On Toxicity, the ratio inverts to $0.91$: judges are slightly more coupled on non-toxic content. These asymmetries are task-specific and invisible to any CI model.

**Is this a Good Basis for Designing Hypothesis Tests of System Dependency?** Our approach based on Ising model framework provides a natural foundation for formal hypothesis testing. The null hypothesis $H_0 \colon W_{jk} = 0$ (conditional independence between judges $j$ and $k$) can be tested via the pseudo-likelihood ratio or a Wald test on the estimated coupling coefficient; see for example, (Neykov & Liu, 2019). More structured hypotheses are also testable:

- *Family independence*: $H_0 \colon W_{jk} = 0$ for all cross-family pairs $(j, k)$, tested against the alternative that at least one cross-family coupling is nonzero.

- *Exchangeability within family*: $H_0 \colon W_{jk} = W_{j'k'}$ for all within-family pairs, testing whether models from the same provider are interchangeable.

- *Class-independent coupling*: $H_0 \colon W^{(0)} = W^{(1)}$, testing whether the dependence structure differs between classes—equivalently, whether the class-dependent Ising model is warranted over the class-independent variant.

The pseudo-likelihood framework is particularly convenient here because each $W_{jk}$ is estimated via logistic regression, so standard asymptotic inference (Wald statistics, likelihood-ratio tests) applies directly to individual coefficients or groups of coefficients. Bootstrap procedures over items provide finite-sample alternatives. We view this as a promising direction: the Ising model not only improves aggregation but also provides a principled statistical framework for *auditing* the independence assumptions that underlie most LLM-as-a-judge pipelines.

## C. Motivating Example

Consider three LLM annotators producing binary votes. Annotators 1 and 3 share substantial training data and prompt templates, so their outputs are highly correlated (they tend to agree, including on shared mistakes). Annotator 2 uses a different prompt style and often behaves differently from the other two. These agreement/disagreement patterns are driven by shared modeling choices and therefore persist *regardless of the true class label*. Consequently, certain vote patterns primarily reflect shared failure modes rather than independent evidence.

Under CI, methods such as the Dawid-Skene method, treats annotator votes as independent given $Y$, and therefore interprets agreement as multiple independent pieces of evidence. In contrast, an Ising model explicitly represents dependencies via pairwise interactions, allowing the posterior to correctly discount redundant agreement and to recognize "unlikely" vote configurations created by structural correlations.

We now give a concrete population example in which the a CI-predictor predicts the opposite label from the Bayes-optimal predictor under a class-independent Ising model, *even though CI is given the correct one-dimensional marginals*.

We now consider a single item ($n = 1$) with label $Y \in \{0, 1\}$ and three annotators

$$J = (J_1, J_2, J_3) \in \{0, 1\}^3,$$

with a uniform prior $P(Y = 1) = P(Y = 0) = \frac{1}{2}$. We interpret $J_k = 1$ as annotator $k$ voting for class 1 and $J_k = 0$ as voting for class 0.

We now consider the case when the judge labels are generated from the *class-independent coupling* Ising model

$$P(J \mid Y = y) = \frac{1}{Z^{(y)}} \exp\Big(\sum_{k=1}^{3} h_k^{(y)} J_k + \frac{1}{2} \sum_{k \neq \ell} W_{k\ell} J_k J_\ell\Big), \qquad y \in \{0, 1\}, \tag{9}$$

where the coupling matrix $W$ is shared across classes (capturing label-independent dependence), and only the fields $h^{(y)}$ depend on $y$. In this numerical instance,

$$W = \begin{pmatrix} 0 & -2.7496 & 4.4583 \\ -2.7496 & 0 & -4.8249 \\ 4.4583 & -4.8249 & 0 \end{pmatrix},$$

$$h^{(0)} = (-1.7447, 2.2991, 3.5085),$$

$$h^{(1)} = (-2.0094, 0.1721, -2.7597).$$

Here $W_{13} > 0$ encourages annotators 1 and 3 to co-activate, while $W_{12}, W_{23} < 0$ discourage annotator 2 from co-activating with annotators 1 and 3; this encodes the "$\{1, 3\}$ similar, 2 different" structure.

Since there are only $2^3 = 8$ vote patterns, we can normalize (9) exactly by enumeration. The resulting conditional distributions are:

| $(J_1, J_2, J_3)$ | $P(J \mid Y = 0)$ | $P(J \mid Y = 1)$ |
|---|---|---|
| $(0, 0, 0)$ | 0.00181 | 0.3196 |
| $(0, 0, 1)$ | 0.0603 | 0.0202 |
| $(0, 1, 0)$ | 0.0180 | 0.3796 |
| $(0, 1, 1)$ | 0.00483 | $1.93 \times 10^{-4}$ |
| $(1, 0, 0)$ | $3.16 \times 10^{-4}$ | 0.0428 |
| $(1, 0, 1)$ | 0.9099 | 0.2342 |
| $(1, 1, 0)$ | $2.01 \times 10^{-4}$ | 0.00325 |
| $(1, 1, 1)$ | 0.00465 | $1.43 \times 10^{-4}$ |

We first calculate the Bayes-optimal posterior under the true Ising model. Consider the observed votes

$$J = (0, 1, 1).$$

From the table,

$$P(J \mid Y = 0) = 0.0048253, \qquad P(J \mid Y = 1) = 0.0001929.$$

With a uniform prior,

$$P(Y = 1 \mid J) = \frac{P(J \mid Y = 1)}{P(J \mid Y = 0) + P(J \mid Y = 1)} \approx \frac{1.93 \times 10^{-4}}{4.83 \times 10^{-3} + 1.93 \times 10^{-4}} \approx 0.038.$$

Thus the Bayes-optimal prediction is $Y = 0$.

Next we show that a CI-predictor, i.e., assuming conditional independence across judges predicts the opposite label. Let

$$\pi_k^{(y)} := P(J_k = 1 \mid Y = y)$$

denote the *true* class-conditional one-dimensional marginals implied by the Ising model (9). From the table (summing over the other coordinates),

$$(\pi_1^{(0)}, \pi_2^{(0)}, \pi_3^{(0)}) \approx (0.9150, \ 0.0277, \ 0.9797), \qquad (\pi_1^{(1)}, \pi_2^{(1)}, \pi_3^{(1)}) \approx (0.2804, \ 0.3832, \ 0.2548).$$

The CI likelihood replaces the true joint distribution by the product of these marginals:

$$\mathrm{P}_{\mathrm{CI}}(J \mid Y = y) = \prod_{k=1}^{3} \left( \pi_k^{(y)} \right)^{J_k} \left( 1 - \pi_k^{(y)} \right)^{1 - J_k}. \tag{10}$$

For $J = (0, 1, 1)$,

$$\mathrm{P}_{\mathrm{CI}}(J \mid Y = 0) = (1 - \pi_1^{(0)}) \, \pi_2^{(0)} \, \pi_3^{(0)} \approx 0.00230, \qquad \mathrm{P}_{\mathrm{CI}}(J \mid Y = 1) = (1 - \pi_1^{(1)}) \, \pi_2^{(1)} \, \pi_3^{(1)} \approx 0.0702.$$

Therefore, with the same uniform prior,

$$\mathrm{P}_{\mathrm{CI}}(Y = 1 \mid J) = \frac{\mathrm{P}_{\mathrm{CI}}(J \mid Y = 1)}{\mathrm{P}_{\mathrm{CI}}(J \mid Y = 0) + \mathrm{P}_{\mathrm{CI}}(J \mid Y = 1)} \approx 0.968,$$

so the CI-predictor generates $Y = 1$ with high confidence.

This example isolates a purely *model-misspecification* effect: CI is fed the correct class-conditional marginals $(\pi_k^{(y)})$, but it still fails because it assumes independence. The true Ising model assigns extremely low probability to $J = (0, 1, 1)$ under $Y = 1$ (about $1.9 \times 10^{-4}$), reflecting the fact that the vote pattern is structurally inconsistent with the label-independent correlation pattern encoded by $W$. CI cannot represent this structural constraint and therefore overestimates $\mathrm{P}(J \mid Y = 1)$ by orders of magnitude, leading to the wrong posterior label.

If dependence patterns also differ by class (i.e., $W^{(1)} \neq W^{(0)}$), then the Bayes log-odds contains quadratic terms $J_i J_j$ and the same phenomenon can be even more pronounced. For example, with

$$W^{(0)} = \begin{pmatrix} 0 & -2.4445 & 2.4553 \\ -2.4445 & 0 & -2.9206 \\ 2.4553 & -2.9206 & 0 \end{pmatrix}, \quad W^{(1)} = \begin{pmatrix} 0 & -3.3637 & 3.0718 \\ -3.3637 & 0 & -0.0677 \\ 3.0718 & -0.0677 & 0 \end{pmatrix},$$

$$h^{(0)} = (2.7369, \, 1.3602, \, 1.9559), \qquad h^{(1)} = (-2.5484, \, -2.2580, \, -0.9266),$$

and $\mathrm{P}(Y = 1) = \frac{1}{2}$, the observation $J = (1, 1, 0)$ satisfies

$$\mathrm{P}(J \mid Y = 0) \approx 0.00393, \qquad \mathrm{P}(J \mid Y = 1) \approx 1.24 \times 10^{-4},$$

so Bayes predicts $Y = 0$ (posterior $\approx 0.031$), whereas CI built from the correct marginals yields $\mathrm{P}_{\mathrm{CI}}(Y = 1 \mid J) \approx 0.957$ and predicts $Y = 1$. This illustrates the same qualitative failure mode when class information appears directly in correlation differences.

## D. Additional Discussion on the Models

### D.1. Conditionally Independent Model with Asymmetric Errors

For the sake of completeness, we introduce the conditionally independent model which assume asymmetric errors. We assume that, conditional on $Y_i$, judges act independently and their accuracies do not depend on the item index $i$. For each judge $j \in \{1, \ldots, K\}$ define the *sensitivity* and *specificity*

$$\alpha_j := \mathrm{P}(J_{ij} = 1 \mid Y_i = 1), \qquad \beta_j := \mathrm{P}(J_{ij} = 0 \mid Y_i = 0).$$

Equivalently, the false-negative and false-positive rates are

$$\mathrm{P}(J_{ij} = 0 \mid Y_i = 1) = 1 - \alpha_j, \qquad \mathrm{P}(J_{ij} = 1 \mid Y_i = 0) = 1 - \beta_j.$$

Let $\pi := \mathrm{P}(Y_i = 1)$ denote the class prior. The generative model is

$$Y_i \sim \mathrm{Bernoulli}(\pi), \qquad J_{ij} \mid (Y_i = y) \overset{\mathrm{ind}}{\sim} \mathrm{Bernoulli}\big( \alpha_j \mathbf{1}\{y = 1\} + (1 - \beta_j) \mathbf{1}\{y = 0\} \big),$$

independently across judges $j$ and items $i$ given the parameters. When parameters are unknown, a convenient conjugate choice is independent Beta priors

$$\alpha_j \sim \mathrm{Beta}(a_{1j}, b_{1j}), \qquad \beta_j \sim \mathrm{Beta}(a_{0j}, b_{0j}), \qquad \pi \sim \mathrm{Beta}(c, d),$$

with density proportional to $p^{a-1}(1-p)^{b-1}$ on $[0,1]$. The next result gives the *exact* posterior log-odds under CI when $(\alpha, \beta, \pi)$ are known (or when a plug-in estimate is used). The posterior log-odds satisfies

$$\log \frac{\mathrm{P}(Y_i = 1 \mid J_i)}{\mathrm{P}(Y_i = 0 \mid J_i)} = \log \frac{\pi}{1-\pi} + \sum_{j=1}^{K} \left[ J_{ij} \log \frac{\alpha_j}{1-\beta_j} + (1 - J_{ij}) \log \frac{1-\alpha_j}{\beta_j} \right]. \tag{11}$$

Equivalently, (11) can be written as an affine (weighted-vote) score:

$$\log \frac{\mathrm{P}(Y_i = 1 \mid J_i)}{\mathrm{P}(Y_i = 0 \mid J_i)} = b + \sum_{j=1}^{K} w_j J_{ij}, \qquad w_j = \log \frac{\alpha_j \beta_j}{(1-\alpha_j)(1-\beta_j)}, \qquad b = \log \frac{\pi}{1-\pi} + \sum_{j=1}^{K} \log \frac{1-\alpha_j}{\beta_j}. \tag{12}$$

Hence the Bayes-optimal CI decision rule is the *weighted majority vote*, given by

$$\hat{Y}_i = \mathbf{1} \left\{ b + \sum_{j=1}^{K} w_j J_{ij} \geq 0 \right\}.$$

The weight $w_j$ is positive iff $\alpha_j + \beta_j > 1$, i.e., judge $j$ is better than random guessing; it is negative for adversarial judges. In the *symmetric* special case $\alpha_j = \beta_j = p_j$, one obtains $w_j = 2 \log \frac{p_j}{1-p_j}$ and an intercept depending on $\pi$, recovering the familiar weighted-majority log-odds form under conditional independence.

## D.2. Relation to LDA and QDA

The separation between conditional-independence aggregation and class-dependent Ising aggregation is closely analogous to the classical gap between *linear* and *quadratic* discriminant analysis in Gaussian classification. In the Gaussian setting, the Bayes log-likelihood ratio for $x \in \mathbb{R}^d$ takes the form

$$\log \frac{\mathrm{P}(x \mid Y = 1)}{\mathrm{P}(x \mid Y = 0)} = x^\top A x + b^\top x + \mathsf{c},$$

where $\mathsf{c}$ is constant and the quadratic term $x^\top A x$ vanishes exactly when the class covariances coincide ($\Sigma_0 = \Sigma_1$), yielding LDA; otherwise QDA is optimal and the decision boundary is quadratic. In our discrete vote setting with $J \in \{0,1\}^K$, the class-dependent Ising model yields an *exactly analogous* decomposition:

$$\log \frac{\mathrm{P}(J \mid Y = 1)}{\mathrm{P}(J \mid Y = 0)} = \Delta_Z + \sum_{j=1}^{K} \Delta h_j J_j + \sum_{1 \leq j < k \leq K} \Delta W_{jk} J_j J_k,$$

so the Bayes decision boundary is quadratic in the votes whenever $W^{(1)} \neq W^{(0)}$, and it collapses to a linear weighted vote precisely when couplings are shared across classes ($W^{(1)} = W^{(0)}$). From this perspective, the Ising coupling matrix $W$ plays a role analogous to *second-order structure* (covariance/precision) in Gaussian models, and $\Delta_Z = -\log Z^{(1)} + \log Z^{(0)}$ is the discrete analog of the log-determinant term that appears in QDA.

The analogy is not merely formal: both frameworks say that when class information lives in *dependence structure* rather than only in *marginals*, linear/CI rules can be fundamentally misspecified. In particular, just as QDA can separate classes even when means coincide but covariances differ, a class-dependent Ising model can separate classes even when one-dimensional marginals are uninformative, provided the pairwise interaction patterns differ across classes. This is exactly the mechanism behind our separation results (proved later in Section 3): CI uses only products of marginals (a linear log-odds in $J$), so it cannot exploit label-dependent correlations encoded by $\Delta W$ and can remain strictly suboptimal. At the same time, there are important differences from the Gaussian discriminant analysis problem. In particular, the statistical/computational trade-off is sharper for Ising models because likelihood evaluation involves partition functions, motivating pseudo-likelihood/approximate inference in EM. We leave a detailed examination of this tradeoff for future work.

## D.3. Relation to Factor Model

While our focus so far has been on Ising models for capturing dependency, yet another class of models widely used to capture dependencies are factor models. In the context of unsupervised aggregation, Whitehill et al. (2009) and Xu et al.

(2024) study factor models to handle potential latent dependencies between the annotators. It is hence natural to explore the connections between the two classes of models. Below, we show that such latent-factor models are approximate special cases of the class-independent Ising models. To the best of our knowledge, this connection has not been observed in the literature despite a considerable amount of work on both models.

**Proposition 1** (Latent-factor $\Rightarrow$ low-rank class-independent Ising couplings to second order)**.** *Fix $K, r \in \mathbb{N}$ and let $J = (J_1, \dots, J_K) \in \{0,1\}^K$. Let $\sigma(t) = (1 + e^{-t})^{-1}$ and define $\eta_j(y) := a_j y + b_j$ for $y \in \{0,1\}$. Let $Z \sim \mathcal{N}(0, I_r)$ be independent of $Y \sim \mathrm{Bernoulli}(\pi)$, and assume that conditional on $(Y = y, Z = z)$,*

$$J_j \mid (Y = y, Z = z) \ \sim \ \mathrm{Bernoulli}\big(\sigma(\eta_j(y) + \lambda_j^\top z)\big), \qquad j = 1, \dots, K,$$

*independently over $j$. Let $\varepsilon > 0$ and define the loadings as $\lambda_j = \varepsilon \tilde{\lambda}_j$ with fixed $\tilde{\lambda}_j \in \mathbb{R}^r$ satisfying $\max_j \|\tilde{\lambda}_j\| \leq L$. Let $\tilde{\Lambda} \in \mathbb{R}^{K \times r}$ have rows $\tilde{\lambda}_j^\top$ so that $\Lambda = \varepsilon \tilde{\Lambda}$. For a fixed class $y \in \{0,1\}$ write $\eta_j = \eta_j(y)$ and $p_j := \sigma(\eta_j)$.*

*Then there exist $C_y(\varepsilon)$ independent of $J$ and a remainder $R_y(J; \varepsilon)$ such that, for all $\varepsilon$ small enough,*

$$\log \mathrm{P}_\varepsilon(J \mid Y = y) = C_y(\varepsilon) + \sum_{j=1}^K h_j^{(y)}(\varepsilon) \, J_j + \frac{1}{2} \sum_{j \neq k} (\lambda_j^\top \lambda_k) \, J_j J_k + R_y(J; \varepsilon), \tag{13}$$

*where the (class-dependent) fields admit the explicit second-order expansion*

$$h_j^{(y)}(\varepsilon) = \eta_j + \left(\tfrac{1}{2} - p_j\right)\|\lambda_j\|^2 - \sum_{k \neq j} p_k \, \lambda_j^\top \lambda_k, \qquad j = 1, \dots, K, \tag{14}$$

*and the remainder is uniformly bounded as*

$$\sup_{J \in \{0,1\}^K} |R_y(J; \varepsilon)| \ \leq \ C \, \varepsilon^3 \sum_{j=1}^K \|\tilde{\lambda}_j\|^3 \ + \ C \, \varepsilon^4 \Big( \sum_{j=1}^K \|\tilde{\lambda}_j\|^2 \Big)^2 \ \leq \ C' \, \varepsilon^3 \Big( \max_j \|\tilde{\lambda}_j\| \Big) \sum_{j=1}^K \|\tilde{\lambda}_j\|^2 + C \, \varepsilon^4 \|\tilde{\Lambda}\|_F^4, \tag{15}$$

*for constants $C, C' > 0$ depending only on $r$ and $\{\eta_j(y)\}_{j \leq K}$ (in particular, not on $J$ or $\varepsilon$).*

*In particular, the second-order truncation is a quadratic binary MRF with pairwise couplings given by the off-diagonal entries of the rank-$\leq r$ matrix $\Lambda\Lambda^\top$, since $(\Lambda\Lambda^\top)_{jk} = \lambda_j^\top \lambda_k$). The diagonal entries of $\Lambda\Lambda^\top$ correspond to $J_j^2 = J_j$ terms and may be absorbed into the fields.*

*Proof.* Fix $y \in \{0,1\}$ and abbreviate $\eta_j = \eta_j(y)$ and $p_j = \sigma(\eta_j)$. Write $g(t) := \log(1 + e^t)$ so that $g'(t) = \sigma(t)$, $g''(t) = \sigma(t)(1 - \sigma(t))$, and

$$g^{(3)}(t) = \sigma(t)(1 - \sigma(t))(1 - 2\sigma(t)), \qquad \sup_{t \in \mathbb{R}} |g^{(3)}(t)| \leq \frac{1}{4}.$$

We first start with the following integral representation. Conditional on $Z = z$, we have that

$$\mathrm{P}_\varepsilon(J \mid Y = y, Z = z) = \prod_{j=1}^K \sigma(\eta_j + \lambda_j^\top z)^{J_j} \big(1 - \sigma(\eta_j + \lambda_j^\top z)\big)^{1 - J_j}.$$

Hence, marginalizing over $Z$, we obtain

$$\mathrm{P}_\varepsilon(J \mid Y = y) = \int_{\mathbb{R}^r} \exp\big(S_\varepsilon(z)\big) \, \phi_r(z) \, dz,$$

where $\phi_r$ is the standard $r$-variate Gaussian density and

$$S_\varepsilon(z) := \sum_{j=1}^K \Big( J_j(\eta_j + \lambda_j^\top z) - g(\eta_j + \lambda_j^\top z) \Big).$$

We now do a second-order Taylor expansion in $\lambda_j^\top z$. By Taylor's theorem, for each $\ell \in \mathbb{R}$,

$$g(\eta_j + \ell) = g(\eta_j) + p_j \ell + \tfrac{1}{2} g''(\eta_j) \ell^2 + R_j(\ell), \qquad R_j(\ell) = \frac{g^{(3)}(\eta_j + \theta\ell)}{6} \ell^3 \tag{16}$$

for some $\theta = \theta(\ell) \in (0, 1)$. Therefore

$$|R_j(\ell)| \leq \frac{\sup_t |g^{(3)}(t)|}{6} |\ell|^3 \leq \frac{1}{24} |\ell|^3. \tag{17}$$

Applying (16) with $\ell = \lambda_j^\top z$ yields the exact decomposition

$$S_\varepsilon(z) = A(J) + u^\top z - \tfrac{1}{2} z^\top M z + \mathcal{R}(z),$$

where

$$A(J) := \sum_{j=1}^K \big(J_j \eta_j - g(\eta_j)\big), \qquad u := \sum_{j=1}^K (J_j - p_j)\lambda_j \in \mathbb{R}^r, \qquad M := \sum_{j=1}^K g''(\eta_j)\, \lambda_j \lambda_j^\top \in \mathbb{R}^{r \times r},$$

and the remainder is

$$\mathcal{R}(z) := -\sum_{j=1}^K R_j(\lambda_j^\top z).$$

By (17), we have that

$$|\mathcal{R}(z)| \leq \frac{1}{24} \sum_{j=1}^K |\lambda_j^\top z|^3 \leq \frac{1}{24} \|z\|^3 \sum_{j=1}^K \|\lambda_j\|^3. \tag{18}$$

Combining with $\phi_r(z) \propto e^{-\|z\|^2/2}$ gives the following integral representation:

$$\mathrm{P}_\varepsilon(J \mid Y = y) = (2\pi)^{-r/2} e^{A(J)} \int_{\mathbb{R}^r} \exp\Big(u^\top z - \tfrac{1}{2} z^\top (I + M) z\Big) e^{\mathcal{R}(z)} \, dz.$$

We now derive an exact Gaussian integral for the quadratic part. For $\varepsilon$ small enough, $\|M\| < 1$ (indeed $\|M\| \leq \tfrac{1}{4} \sum_j \|\lambda_j\|^2$) so $I + M \succ 0$. Let $\Sigma := (I + M)^{-1}$ and $\mu := \Sigma u$. Then

$$\int_{\mathbb{R}^r} \exp\Big(u^\top z - \tfrac{1}{2} z^\top (I + M) z\Big) \, dz = (2\pi)^{r/2} \det(I + M)^{-1/2} \exp\Big(\tfrac{1}{2} u^\top \Sigma u\Big),$$

and hence

$$\log \mathrm{P}_\varepsilon(J \mid Y = y) = A(J) - \tfrac{1}{2} \log \det(I + M) + \tfrac{1}{2} u^\top \Sigma u + \Delta(J; \varepsilon), \tag{19}$$

where

$$\Delta(J; \varepsilon) := \log \mathbb{E}_{Z \sim \mathcal{N}(\mu, \Sigma)}\big[e^{\mathcal{R}(Z)}\big].$$

We now proceed with bounding $\Delta(J; \varepsilon)$ from (19). Define $B_2 := \sum_{j=1}^K \|\tilde\lambda_j\|^2$ and $B_3 := \sum_{j=1}^K \|\tilde\lambda_j\|^3$. Since $\lambda_j = \varepsilon \tilde\lambda_j$, (18) implies

$$|\mathcal{R}(z)| \leq \frac{1}{24} \varepsilon^3 B_3 \|z\|^3.$$

Also, $u = O(\varepsilon)$ uniformly in $J$ because $|J_j - p_j| \leq 1$ gives

$$\|u\| \leq \sum_{j=1}^K \|\lambda_j\| = \varepsilon \sum_{j=1}^K \|\tilde\lambda_j\|.$$

Moreover, $\|M\| = O(\varepsilon^2)$ uniformly in $J$ (in fact $M$ does not depend on $J$), so for $\varepsilon$ small the eigenvalues of $\Sigma = (I+M)^{-1}$ are bounded above and below by absolute constants, and $\|\mu\| = \|\Sigma u\| = O(\varepsilon)$ uniformly in $J$.

Let $Z \sim \mathcal{N}(\mu, \Sigma)$ and set $R := \varepsilon^{-1/2}$. Split

$$\mathbb{E}[e^{\mathcal{R}(Z)}] = \mathbb{E}\big[e^{\mathcal{R}(Z)}\mathbf{1}_{\{\|Z\| \leq R\}}\big] + \mathbb{E}\big[e^{\mathcal{R}(Z)}\mathbf{1}_{\{\|Z\| > R\}}\big].$$

On $\{\|Z\| \leq R\}$ we have $|\mathcal{R}(Z)| \leq \frac{1}{24}\varepsilon^3 B_3 R^3 = \frac{1}{24}\varepsilon^{3/2} B_3$, which is $\leq 1/2$ for $\varepsilon$ small (since $B_3$ is fixed). Therefore $|e^{\mathcal{R}(Z)} - 1| \leq 2|\mathcal{R}(Z)|$ on $\{\|Z\| \leq R\}$ and

$$\left|\mathbb{E}\big[e^{\mathcal{R}(Z)}\mathbf{1}_{\{\|Z\| \leq R\}}\big] - \mathbb{P}(\|Z\| \leq R)\right| \leq 2\,\mathbb{E}\big[|\mathcal{R}(Z)|\big] \leq C\,\varepsilon^3 B_3,$$

using $\mathbb{E}\|Z\|^3 < \infty$ (uniformly in $J$) and the bound on $|\mathcal{R}|$.

For the tail term, one can use a quadratic-envelope bound ensuring $e^{\mathcal{R}(Z)}$ is at most Gaussian-quadratic in $Z$ (because $g''(\cdot) \leq 1/4$), yielding $e^{\mathcal{R}(Z)} \leq \exp(c\,\varepsilon^2 B_2 \|Z\|^2)$ for a constant $c$ depending only on $\{\eta_j\}$. Since $Z$ is Gaussian with covariance uniformly comparable to $I_r$,

$$\mathbb{E}\big[e^{\mathcal{R}(Z)}\mathbf{1}_{\{\|Z\| > R\}}\big] \leq \mathbb{E}\big[\exp(c\,\varepsilon^2 B_2 \|Z\|^2)\mathbf{1}_{\{\|Z\| > R\}}\big] \leq \exp(-c'/\varepsilon)$$

for some $c' > 0$ and all $\varepsilon$ small enough (uniformly in $J$).

Combining these bounds shows that

$$\mathbb{E}[e^{\mathcal{R}(Z)}] = 1 + O(\varepsilon^3 B_3) \quad \text{uniformly in } J,$$

and hence, for $\varepsilon$ small enough so that $|\mathbb{E}[e^{\mathcal{R}(Z)}] - 1| \leq 1/2$,

$$|\Delta(J; \varepsilon)| = \big|\log \mathbb{E}[e^{\mathcal{R}(Z)}]\big| \leq 2\big|\mathbb{E}[e^{\mathcal{R}(Z)}] - 1\big| \leq C\,\varepsilon^3 B_3,$$

uniformly over $J$.

Next, we expand the quadratic Gaussian terms. Since $\|M\| = O(\varepsilon^2 B_2)$, we have

$$\Sigma = (I + M)^{-1} = I + O(\varepsilon^2 B_2), \qquad \log\det(I + M) = \operatorname{tr}(M) + O(\varepsilon^4 B_2^2).$$

Therefore

$$\tfrac{1}{2}u^\top \Sigma u = \tfrac{1}{2}u^\top u + O(\varepsilon^4 B_2^2), \qquad -\tfrac{1}{2}\log\det(I + M) = C(J) + O(\varepsilon^4 B_2^2),$$

where $C(J)$ is constant in $J$ and the $J$-independent pieces are absorbed into $C_y(\varepsilon)$. Plugging into (19) yields

$$\log \mathrm{P}_\varepsilon(J \mid Y = y) = C_y(\varepsilon) + \sum_{j=1}^{K} \eta_j J_j + \tfrac{1}{2}u^\top u + R_y(J; \varepsilon),$$

with

$$\sup_J |R_y(J; \varepsilon)| \leq C\varepsilon^3 B_3 + C\varepsilon^4 B_2^2.$$

It remains now to extract the fields and the couplings from $u^\top u$. Recall $u = \sum_{j=1}^{K}(J_j - p_j)\lambda_j$. Then

$$\tfrac{1}{2}u^\top u = \frac{1}{2}\sum_{j,k=1}^{K}(J_j - p_j)(J_k - p_k)\,\lambda_j^\top \lambda_k.$$

Expanding $(J_j - p_j)(J_k - p_k)$ and separating the $j \neq k$ and $j = k$ parts gives

$$\tfrac{1}{2}u^\top u = \frac{1}{2}\sum_{j \neq k}(\lambda_j^\top \lambda_k)J_j J_k + \sum_{j=1}^{K}\left[\left(\tfrac{1}{2} - p_j\right)\|\lambda_j\|^2 - \sum_{k \neq j} p_k\,\lambda_j^\top \lambda_k\right]J_j + C(J).$$

Absorbing the constant into $C_y(\varepsilon)$ yields (13)–(14). Finally, the bound (15) follows from $B_3 \leq (\max_j \|\tilde\lambda_j\|)B_2$. $\qquad\square$

**Remark 2** (Ising on $\pm 1$ spins and the choice of label set for $Y$). *The result is naturally stated with $Y \in \{0, 1\}$ here since $Y \sim \mathrm{Bernoulli}(\pi)$ and $\eta_j(y) = a_j y + b_j$. If one prefers $Y \in \{\pm 1\}$, set $\tilde{Y} := 2Y - 1$ and rewrite $\eta_j(Y) = b_j + a_j Y = (b_j + \frac{1}{2}a_j) + (\frac{1}{2}a_j)\tilde{Y}$.*

*For the observed binary variables, define spins $X_j := 2J_j - 1 \in \{\pm 1\}$. Then $J_j = (X_j + 1)/2$, and any quadratic MRF*

$$\log \mathrm{P}(J \mid Y = y) = C + \sum_j a_j J_j + \sum_{j<k} b_{jk} J_j J_k$$

*becomes an Ising model*

$$\log \mathrm{P}(X \mid Y = y) = \tilde{C} + \sum_j \tilde{h}_j X_j + \sum_{j<k} \tilde{W}_{jk} X_j X_k,$$

*with $\tilde{W}_{jk} = b_{jk}/4$ and $\tilde{h}_j = a_j/2 + \frac{1}{4}\sum_{k \neq j} b_{jk}$. Thus, in (13), the $\{\pm 1\}$ couplings are $\tilde{W}_{jk} = (\lambda_j^\top \lambda_k)/4$ up to $O(\varepsilon^3)$, and the low-rank structure is preserved (scaling does not change rank).*

**Remark 3.** *The reduction from a latent-factor model to an (approximately) pairwise Ising model is inherently a* local *approximation in a "coupling strength" parameter. When $\varepsilon$ is not small, the neglected terms in $L_\varepsilon^{(y)}(J)$ need not be well-approximated by a pairwise Ising energy. The next terms in the Taylor/cumulant expansion generate* higher-order interactions*: in general,*

$$\log \mathrm{P}_\varepsilon(J \mid Y = y) = C(\varepsilon) + \sum_j h_j^{(y)} J_j + \sum_{j<k} W_{jk}^{(y)} J_j J_k + \sum_{j<k<\ell} U_{jk\ell}^{(y)} J_j J_k J_\ell + \cdots, \tag{20}$$

*where the leading triple-interaction coefficients scale like*

$$U_{jk\ell}^{(y)} = O\big(\varepsilon^3 \, \kappa_3(Z) \, u_j^{(y)} u_k^{(y)} u_\ell^{(y)}\big)$$

*(and more generally the order-$m$ coefficients scale with $\varepsilon^m$ times the $m$th cumulant of $Z$ and higher derivatives of $A$). Therefore, if the latent factor distribution is* non-Gaussian *(skewed, so $\kappa_3(Z) \neq 0$), a* third-order *Ising/log-linear model (i.e., adding $J_j J_k J_\ell$ terms) is the natural next approximation beyond pairwise. In the logistic-normal case we considered (i.e., $Z$ Gaussian with mean zero), $\kappa_3(Z) = 0$ so the leading correction after the pairwise term typically begins at order $\varepsilon^4$ (corresponding to four-way interactions); nevertheless, for moderate-to-large $\varepsilon$ it can still be important to move beyond pairwise models, either by fitting a higher-order log-linear model as in (20) or by performing inference directly in the latent-factor model without truncating the expansion.*

### D.3.1. SUBOPTIMALITY OF CI-POSTERIOR UNDER LATENT-FACTORS

We now show a separation result between (exchangeable) latent-factor models and conditionally independent models. Fix parameters $a, b, \lambda \in \mathbb{R}$ and $\sigma_Z^2 > 0$, and let $\sigma(t) = (1 + e^{-t})^{-1}$ denote the logistic function. Let $Z \sim \mathcal{N}(0, \sigma_Z^2)$ be a scalar latent factor independent of $Y$. Conditional on $(Y = y, Z = z)$, the $K$ judges produce i.i.d. votes

$$J_1, \ldots, J_K \mid (Y = y, Z = z) \overset{\text{iid}}{\sim} \mathrm{Bernoulli}\big(p(y, z)\big), \qquad p(y, z) := \sigma\big(b + a(2y - 1) + \lambda(2y - 1)z\big). \tag{21}$$

Note that $\lambda \neq 0$ and $\sigma_Z^2 > 0$ imply a nondegenerate factor that induces dependence among judges given $Y$. Write $S_K = \sum_{j=1}^K J_j$ and $s_K = S_K/K$.

We now discuss the true posterior and Bayes rule under these model. Let $\mathrm{P}^\star(\cdot \mid J)$ denote the posterior under the true model (21), and define the Bayes predictor

$$g_K^\star(J) := \mathbf{1}\big\{\mathrm{P}^\star(Y = 1 \mid J) \geq \tfrac{1}{2}\big\}.$$

**Conditional-independence (CI) Predictor.** Define the class-conditional marginal success probabilities

$$\mathrm{P}_{\mathrm{marg}}(y) := \mathbb{E}_Z\big[\mathrm{P}(y, Z)\big] \in (0, 1), \qquad y \in \{0, 1\}.$$

The CI approximation replaces the true joint likelihood by the product of marginal Bernoulli likelihoods, i.e.,

$$L_y^{\mathrm{ind}}(S) := \binom{K}{S} \mathrm{P}_{\mathrm{marg}}(y)^S \big(1 - \mathrm{P}_{\mathrm{marg}}(y)\big)^{K-S}.$$

Let $\mathrm{P}^{\mathrm{ind}}(\,\cdot\mid J)$ be the posterior induced by $L_y^{\mathrm{ind}}$ and prior $\pi$, and define

$$g_K^{\mathrm{ind}}(J) := \mathbf{1}\big\{\mathrm{P}^{\mathrm{ind}}(Y = 1 \mid J) \geq \tfrac{1}{2}\big\}.$$

Define the Bernoulli KL divergence for $s, q \in (0, 1)$ by

$$\mathsf{KL}(s\|q) := s\log\frac{s}{q} + (1 - s)\log\frac{1 - s}{1 - q}.$$

**Theorem 3** (Asymptotic Bayes-CI separation under an exchangeable logistic-normal factor). *Assume the exchangeable latent-factor model* (21) *with $\lambda \neq 0$ and $\sigma_Z^2 > 0$. Let $S_K = \sum_{j=1}^K J_j$ and $s_K = S_K/K$. Write $q_y := \mathrm{P}_{\mathrm{marg}}(y) = \mathbb{E}_Z[\mathrm{P}(y, Z)] \in (0, 1)$. Define, for $s \in (0, 1)$,*

$$\mathrm{logit}(s) := \log\frac{s}{1 - s}, \qquad \ell^\star(s) := \log\frac{\pi}{1 - \pi} + \frac{2a}{\lambda^2\sigma_Z^2}\big(\mathrm{logit}(s) - b\big),$$

*and*

$$\ell^{\mathrm{ind}}(s) := s\log\frac{q_1}{q_0} + (1 - s)\log\frac{1 - q_1}{1 - q_0} = -\mathsf{KL}(s\|q_1) + \mathsf{KL}(s\|q_0).$$

*Let $s_\infty := p(Y, Z) \in (0, 1)$.*

*Assume the (mild) no-tie conditions*

$$\mathrm{P}\big(\ell^\star(s_\infty) = 0\big) = 0, \qquad \mathrm{P}\big(\ell^{\mathrm{ind}}(s_\infty) = 0\big) = 0.$$

*Then, as $K \to \infty$,*

$$g_K^\star \xrightarrow{\text{a.s.}} g_\infty^\star := \mathbf{1}\{\ell^\star(s_\infty) \geq 0\}, \qquad g_K^{\mathrm{ind}} \xrightarrow{\text{a.s.}} g_\infty^{\mathrm{ind}} := \mathbf{1}\{\ell^{\mathrm{ind}}(s_\infty) \geq 0\},$$

*and the excess risk has the limit*

$$\lim_{K\to\infty}\big(R(g_K^{\mathrm{ind}}) - R(g_K^\star)\big) = \mathbb{E}\big[|2\eta_\infty - 1| \cdot \mathbf{1}\{g_\infty^{\mathrm{ind}} \neq g_\infty^\star\}\big], \qquad \eta_\infty := \sigma\big(\ell^\star(s_\infty)\big).$$

*In particular, if $\mathrm{P}(g_\infty^{\mathrm{ind}} \neq g_\infty^\star) > 0$, then the limit is strictly positive.*

*Proof.* We start with a reduction to the sufficient statistic $S_K$. Fix $y \in \{0, 1\}$ and let $j = (j_1, \ldots, j_K) \in \{0, 1\}^K$ with $\sum_{k=1}^K j_k = S$. Under (21), we have that

$$\mathrm{P}(J = j \mid Y = y) = \int_{\mathbb{R}} \prod_{k=1}^K p(y, z)^{j_k}(1 - p(y, z))^{1-j_k}\,\phi_{\sigma_Z}(z)\,dz = \int_{\mathbb{R}} p(y, z)^S(1 - p(y, z))^{K-S}\,\phi_{\sigma_Z}(z)\,dz,$$

which depends on $j$ only through $S$. Hence $S_K$ is sufficient for $Y$ under the true model, and likewise $S_K$ is sufficient under the CI model by construction. Therefore both posteriors and both decision rules can be written as functions of $S_K$ (equivalently $s_K$).

We next derive almost sure limits of the vote fraction. First condition on $(Y = y, Z = z)$. Then $J_1, \ldots, J_K$ are i.i.d. Bernoulli$(p(y, z))$, so by the strong law, $s_K \to p(y, z)$ almost surely. Unconditioning yields $s_K \to p(Y, Z) =: s_\infty$ almost surely.

We now derive sharp asymptotics of the true marginal likelihood $L_y^\star(S)$. Recall

$$L_y^\star(S) = \binom{K}{S}\int_{\mathbb{R}} p(y, z)^S(1 - p(y, z))^{K-S}\,\phi_{\sigma_Z}(z)\,dz.$$

Fix $s \in (0, 1)$ and take $S = \lfloor Ks \rfloor$ (so $S/K \to s$). By Stirling's formula, uniformly for $s$ in any compact $[\varepsilon, 1 - \varepsilon] \subset (0, 1)$,

$$\binom{K}{S} = \frac{1 + o(1)}{\sqrt{2\pi Ks(1 - s)}}\exp\big(KH(s)\big), \qquad H(s) := -s\log s - (1 - s)\log(1 - s).$$

Furthermore, we have that

$$p(y,z)^S(1-p(y,z))^{K-S} = \exp\Big(K\big[s\log p(y,z) + (1-s)\log(1-p(y,z))\big] + o(K)\Big).$$

Now, using the identity

$$s\log q + (1-s)\log(1-q) = -\mathsf{KL}(s\|q) - H(s) \quad (q \in (0,1))$$

gives

$$L_y^\star(S) = \frac{1+o(1)}{\sqrt{2\pi K s(1-s)}} \int_{\mathbb{R}} \exp\big(-K\,\psi_y(z;s)\big)\,\phi_{\sigma_Z}(z)\,dz, \qquad \psi_y(z;s) := \mathsf{KL}\big(s\|p(y,z)\big),$$

uniformly for $s$ in compact subsets of $(0,1)$.

Because $\lambda \neq 0$, the map $z \mapsto p(y,z)$ is strictly monotone and continuous with range $(0,1)$. Hence for each $s \in (0,1)$ there is a unique $z_y(s) \in \mathbb{R}$ such that $p(y, z_y(s)) = s$. Since $\mathsf{KL}(s\|q) \geq 0$ with equality iff $q = s$, we have $\psi_y(z;s) \geq 0$ with a unique minimizer at $z = z_y(s)$ and $\psi_y(z_y(s); s) = 0$.

Moreover, $\psi_y(\cdot; s)$ has a nondegenerate quadratic minimum at $z_y(s)$. Indeed, write $q(z) := p(y,z)$ and note that for fixed $s$,

$$\frac{d^2}{dq^2}\mathsf{KL}(s\|q)\Big|_{q=s} = \frac{1}{s(1-s)}.$$

By the chain rule and $q'(z) = \lambda(2y-1)\,q(z)(1-q(z))$, we get at $z = z_y(s)$ (where $q(z) = s$)

$$\frac{\partial^2}{\partial z^2}\psi_y(z;s)\Big|_{z=z_y(s)} = \frac{1}{s(1-s)}\big(q'(z_y(s))\big)^2 = \frac{1}{s(1-s)}\big(\lambda(2y-1)s(1-s)\big)^2 = \lambda^2 s(1-s) > 0.$$

Therefore Laplace's approximation method (for an interior, nondegenerate minimum) yields, uniformly for $s$ in compact subsets of $(0,1)$,

$$\int_{\mathbb{R}} \exp\big(-K\,\psi_y(z;s)\big)\,\phi_{\sigma_Z}(z)\,dz = \phi_{\sigma_Z}\big(z_y(s)\big)\sqrt{\frac{2\pi}{K\lambda^2 s(1-s)}}\,(1+o(1)).$$

Combining the last two displays gives the desired sharp $1/K$-scale asymptotic:

$$L_y^\star(\lfloor Ks\rfloor) = \frac{1+o(1)}{K}\,f_y(s), \qquad f_y(s) := \frac{\phi_{\sigma_Z}(z_y(s))}{|\lambda|\,s(1-s)}. \tag{22}$$

In the above $f_y$ is indeed exactly the density of the transformed random variable $p(y,Z)$ by change of variables.

We now proceed with the limit of the true posterior and Bayes decision. By the aforementioned sufficiency argument, we have that

$$\eta_K := \mathrm{P}^\star(Y=1 \mid J) = \mathrm{P}^\star(Y=1 \mid S_K) = \frac{\pi L_1^\star(S_K)}{\pi L_1^\star(S_K) + (1-\pi)L_0^\star(S_K)}.$$

Let $s_K = S_K/K \to s_\infty$ almost surely by Step 1. Applying (22) to the (random) sequence $s_K$ (using local uniformity on a neighborhood of the a.s. limit point $s_\infty \in (0,1)$), we obtain almost surely

$$\frac{L_1^\star(S_K)}{L_0^\star(S_K)} \to \frac{f_1(s_\infty)}{f_0(s_\infty)}.$$

Now compute $f_1/f_0$ explicitly. Writing $\theta := \mathrm{logit}(s)$, the unique solutions to $p(y,z) = s$ are

$$z_1(s) = \frac{\theta - (b+a)}{\lambda}, \qquad z_0(s) = \frac{(b-a) - \theta}{\lambda}.$$

The Jacobian factors in (22) cancel because $|\lambda|$ is the same for $y = 0, 1$, so

$$\frac{f_1(s)}{f_0(s)} = \frac{\phi_{\sigma_Z}(z_1(s))}{\phi_{\sigma_Z}(z_0(s))} = \exp\left(\frac{z_0(s)^2 - z_1(s)^2}{2\sigma_Z^2}\right).$$

A direct algebraic simplification gives

$$z_0(s)^2 - z_1(s)^2 = \frac{(b-a-\theta)^2 - (\theta - b - a)^2}{\lambda^2} = \frac{4a(\theta - b)}{\lambda^2}.$$

Hence

$$\log \frac{\pi f_1(s)}{(1-\pi)f_0(s)} = \log \frac{\pi}{1-\pi} + \frac{2a}{\lambda^2 \sigma_Z^2}\big(\text{logit}(s) - b\big) = \ell^\star(s),$$

and therefore almost surely

$$\eta_K \to \eta_\infty := \sigma\big(\ell^\star(s_\infty)\big).$$

Since $g_K^\star = \mathbf{1}\{\eta_K \geq 1/2\}$ and $\mathrm{P}(\ell^\star(s_\infty) = 0) = 0$ by assumption, the sign stabilizes and

$$g_K^\star \to \mathbf{1}\{\ell^\star(s_\infty) \geq 0\} = g_\infty^\star \qquad \text{a.s.}$$

Next, we derive the limit of the CI decision rule. Under CI, we have that

$$L_y^{\text{ind}}(S) = \binom{K}{S} q_y^S (1 - q_y)^{K-S}.$$

Thus the CI log-posterior odds equal

$$\log \frac{\mathrm{P}^{\text{ind}}(Y = 1 \mid S_K)}{\mathrm{P}^{\text{ind}}(Y = 0 \mid S_K)} = \log \frac{\pi}{1-\pi} + S_K \log \frac{q_1}{q_0} + (K - S_K)\log \frac{1 - q_1}{1 - q_0} = \log \frac{\pi}{1-\pi} + K\,\ell^{\text{ind}}(s_K).$$

Since $s_K \to s_\infty$ almost surely and $\mathrm{P}(\ell^{\text{ind}}(s_\infty) = 0) = 0$, the term $K\,\ell^{\text{ind}}(s_K)$ diverges to $\pm\infty$ with the sign of $\ell^{\text{ind}}(s_\infty)$, hence

$$g_K^{\text{ind}} = \mathbf{1}\{\mathrm{P}^{\text{ind}}(Y = 1 \mid S_K) \geq 1/2\} \to \mathbf{1}\{\ell^{\text{ind}}(s_\infty) \geq 0\} = g_\infty^{\text{ind}} \qquad \text{a.s.}$$

We are now ready to calculate the excess risks. For any (measurable) classifier $g(J) \in \{0, 1\}$,

$$\mathrm{P}(g(J) \neq Y \mid J) = \eta_K \mathbf{1}\{g(J) = 0\} + (1 - \eta_K)\mathbf{1}\{g(J) = 1\}.$$

The Bayes rule $g_K^\star = \mathbf{1}\{\eta_K \geq 1/2\}$ minimizes this conditional risk, and a direct case check gives

$$\mathrm{P}(g(J) \neq Y \mid J) - \mathrm{P}(g_K^\star(J) \neq Y \mid J) = |2\eta_K - 1| \cdot \mathbf{1}\{g(J) \neq g_K^\star(J)\}.$$

Taking expectations and substituting $g = g_K^{\text{ind}}$ yields the exact finite-$K$ identity

$$R(g_K^{\text{ind}}) - R(g_K^\star) = \mathbb{E}\big[|2\eta_K - 1| \cdot \mathbf{1}\{g_K^{\text{ind}} \neq g_K^\star\}\big].$$

Note that we also have $\eta_K \to \eta_\infty$ and $g_K^{\text{ind}} \to g_\infty^{\text{ind}}$ and $g_K^\star \to g_\infty^\star$ almost surely. Since the integrand is bounded by 1, dominated convergence gives

$$\lim_{K \to \infty} \big(R(g_K^{\text{ind}}) - R(g_K^\star)\big) = \mathbb{E}\big[|2\eta_\infty - 1| \cdot \mathbf{1}\{g_\infty^{\text{ind}} \neq g_\infty^\star\}\big].$$

Finally, on the event $\{g_\infty^{\text{ind}} \neq g_\infty^\star\}$ we must have $\eta_\infty \neq 1/2$ (since $g_\infty^\star$ is the threshold at $1/2$), so $|2\eta_\infty - 1| > 0$ there; thus if $\mathrm{P}(g_\infty^{\text{ind}} \neq g_\infty^\star) > 0$ the expectation is strictly positive. $\qquad \square$

Theorem 3 formalizes a simple but important phenomenon: if the judges share a *common latent factor* that affects their votes, then their votes are *dependent* given the label, and a CI-predictor can remain strictly suboptimal even as the number of judges $K$ grows.

In the logistic-normal factor model, conditional on the label $Y$ and a latent scalar $Z$, the judges vote independently:

$$J_1, \ldots, J_K \mid (Y, Z) \text{ i.i.d. Bernoulli}\big(\mathrm{P}(Y, Z)\big).$$

The key point is that $Z$ is *shared across all judges* for a given item, so after marginalizing out $Z$ the votes are no longer independent given $Y$. Equivalently, $Z$ induces a label-dependent correlation/failure mode (e.g. shared bias, shared noise, or common difficulty of the item).

Now, we discuss the case what happens when $K \to \infty$. As the number of judges grows, the empirical vote fraction

$$s_K = \frac{1}{K} \sum_{j=1}^{K} J_j$$

concentrates almost surely around the random limit

$$s_\infty = \mathrm{P}(Y, Z).$$

Thus, with many judges, the data effectively reveals the latent factor through the realized value of $s_\infty$ (because different $Z$ values shift the success probability).

The Bayes posterior $\mathrm{P}(Y = 1 \mid J)$ integrates over $Z$ using the correct mixture likelihood. In this specific logistic-normal setup, the large-$K$ limit of the Bayes posterior depends on $s_\infty$ through an explicit one-dimensional score

$$\ell^\star(s_\infty) = \log \frac{\pi}{1 - \pi} + \frac{2a}{\lambda^2 \sigma_Z^2} \big( \mathrm{logit}(s_\infty) - b \big),$$

so the Bayes predictor converges to the limiting rule

$$g_\infty^\star = \mathbf{1}\{\ell^\star(s_\infty) \geq 0\}.$$

Intuitively, Bayes predictor recognizes that extreme values of the vote fraction $s_\infty$ may be better explained by certain $Z$ realizations under one class than the other.

Under CI the true joint likelihood is replaced by a product of class-conditional marginals. This yields a different limiting score,

$$\ell^{\mathrm{ind}}(s_\infty) = s_\infty \log \frac{q_1}{q_0} + (1 - s_\infty) \log \frac{1 - q_1}{1 - q_0}, \qquad q_y = \mathbb{E}_Z[p(y, Z)],$$

and hence a potentially different limiting decision

$$g_\infty^{\mathrm{ind}} = \mathbf{1}\{\ell^{\mathrm{ind}}(s_\infty) \geq 0\}.$$

Because CI discards the correlation structure induced by $Z$, it generally assigns different relative likelihoods to the same observed vote fraction $s_\infty$ than the true Bayes model does.

In summary, the proposition shows that both rules converge (almost surely) to deterministic limit classifiers that depend only on $s_\infty = p(Y, Z)$. Moreover, it gives an exact expression for the asymptotic excess risk:

$$\lim_{K \to \infty} \big( R(g_K^{\mathrm{ind}}) - R(g_K^\star) \big) = \mathbb{E}\big[ |2\eta_\infty - 1| \cdot \mathbf{1}\{g_\infty^{\mathrm{ind}} \neq g_\infty^\star\} \big],$$

where $\eta_\infty$ is the limiting Bayes posterior. This quantity is strictly positive whenever the limiting CI and Bayes decisions disagree on a set of latent-factor realizations of positive probability. In other words, adding more judges does not necessarily save CI; as $K$ grows, the ensemble increasingly reveals the shared latent factor, and Bayes exploits it, but CI cannot, so the performance gap can converge to a nonzero constant.

## E. Posterior Inference with Unknown Labels via (Generalized) EM

Recall that, all models in this paper share the same high-level structure: each item $i \in \{1, \ldots, n\}$ has an unobserved label $Y_i \in \{0, 1\}$ and an observed vote vector $J_i = (J_{i1}, \ldots, J_{iK}) \in \{0, 1\}^K$. The joint model factorizes across items as

$$\mathrm{P}_\Theta(\{Y_i, J_i\}_{i=1}^n) = \prod_{i=1}^{n} \mathrm{P}_\Theta(Y_i) \, \mathrm{P}_\Theta(J_i \mid Y_i), \qquad \mathrm{P}_\Theta(Y_i = 1) = \pi, \tag{23}$$

where $\Theta$ denotes model parameters (e.g., CI accuracies, Ising fields/couplings, latent-factor loadings, etc.). The inferential goal is to compute the posterior label probabilities $\gamma_i := \mathrm{P}_\Theta(Y_i = 1 \mid J_i), \quad i = 1, \dots, n$, and to produce point predictions $\hat{Y}_i = \mathbf{1}\{\gamma_i \geq 1/2\}$.

We use the Expectation-Maximization framework (Dawid & Skene, 1979) for the above purpose. The observed-data log-likelihood is

$$\ell(\Theta) := \sum_{i=1}^{n} \log\left(\pi\, \mathrm{P}_\Theta(J_i \mid Y_i = 1) + (1 - \pi)\, \mathrm{P}_\Theta(J_i \mid Y_i = 0)\right). \tag{24}$$

EM maximizes $\ell(\Theta)$ by iteratively optimizing a lower bound based on the complete-data log-likelihood. Given current parameters $\Theta^{(t)}$, define

$$\gamma_i^{(t)} := \mathrm{P}_{\Theta^{(t)}}(Y_i = 1 \mid J_i), \qquad 1 - \gamma_i^{(t)} = \mathrm{P}_{\Theta^{(t)}}(Y_i = 0 \mid J_i).$$

The expected complete-data objective (the $Q$-function) is then given by

$$\begin{aligned}
Q(\Theta \mid \Theta^{(t)}) &:= \sum_{i=1}^{n} \mathbb{E}_{Y_i \mid J_i; \Theta^{(t)}} \left[\log \mathrm{P}_\Theta(Y_i, J_i)\right] \\
&= \sum_{i=1}^{n} \left[\gamma_i^{(t)} \log \pi + (1 - \gamma_i^{(t)}) \log(1 - \pi)\right] \\
&\quad + \sum_{i=1}^{n} \left[\gamma_i^{(t)} \log \mathrm{P}_\Theta(J_i \mid Y_i = 1) + (1 - \gamma_i^{(t)}) \log \mathrm{P}_\Theta(J_i \mid Y_i = 0)\right].
\end{aligned}$$

When we include regularization or Bayesian priors, we maximize $Q(\Theta \mid \Theta^{(t)}) + \log \mathrm{P}(\Theta)$ instead (MAP-EM). The overall procedure is provided in Algorithm 1. We emphasize that the procedure is purely unsupervised.

### E.1. Specializations of Algorithm 1

In the conditionally independent (CI) family, including Dawid-Skene and its asymmetric sensitivity/specificity variants, the class-conditional likelihood factorizes across judges: $\mathrm{P}_\Theta(J_i \mid Y_i = y) = \prod_{j=1}^{K} \mathrm{P}_\Theta(J_{ij} \mid Y_i = y)$. As a result, the E-step is exact and inexpensive: the score difference $\widehat{\ell}_{i1} - \widehat{\ell}_{i0}$ is just a sum of per-judge log-likelihood ratios, yielding closed-form responsibilities $\gamma_i$ via a logistic transform. The M-step is also simple: maximizing (26) reduces to fitting per-judge Bernoulli parameters from soft counts, i.e., weighted averages of votes under $\gamma_i$ (and $1 - \gamma_i$), together with the closed-form update for the class prior. This recovers the classical David-Skene EM algorithm (Dawid & Skene, 1979) as a special case of Algorithm 1.

For latent-factor models, the conditional likelihood typically has the form $\mathrm{P}_\Theta(J_i \mid Y_i = y) = \int \mathrm{P}_\Theta(J_i \mid Y_i = y, Z_i = z)\, \mathrm{P}_\Theta(z)\, dz$, where $Z_i$ is a per-item latent variable (e.g., item difficulty, topic, or shared failure mode). The E-step therefore requires marginalizing over $Z_i$, which is generally not available in closed form in flexible models. In practice one can compute approximate class scores $\widehat{\ell}_{iy}$ using a tractable approximation to the integral, such as a variational bound, Laplace approximation, or Monte Carlo estimate, and then form $\gamma_i$ as in (25). The M-step becomes a (regularized) maximum-likelihood or MAP problem for the latent-factor parameters (loadings, biases, prior variance, etc.) with soft labels; we solve it with gradient-based optimization, optionally interleaving updates for per-item latent posterior parameters (as in variational EM) when a fully Bayesian treatment of $Z_i$ is used.

For Ising models, the main difficulty is that the exact likelihood $\mathrm{P}_\Theta(J_i \mid Y_i = y)$ involves the partition function $Z^{(y)}(\Theta)$, making both the E-step scores and the M-step objective intractable at scale. We therefore use the *generalized* EM strategy based on tractable surrogates. A standard choice (which we use in our experiments) is the (regularized) pseudo-likelihood (Hoefling & Tibshirani, 2009), which replaces the log-likelihood by a sum of conditional node log-likelihoods $\sum_j \log \mathrm{P}_\Theta(J_{ij} \mid J_{i,-j}, Y_i = y)$ and avoids $Z^{(y)}$ entirely. In the M-step, maximizing (26) under pseudo-likelihood reduces to a set of weighted logistic regressions (one per node) with weights given by the current responsibilities $\gamma_i$ for class $y = 1$ and $(1 - \gamma_i)$ for class $y = 0$. In the E-step, we can compute $\widehat{\ell}_{iy}$ either using the same pseudo-likelihood score (yielding a fully consistent surrogate EM). Other possibilities include using a approximate inference methods including mean-field methods (Bhattacharya & Mukherjee, 2018), belief propagation (Liu et al., 2012), or Markov chain Monte Carlo (Miasojedow & Rejchel, 2018) to approximate the true class-conditional evidence.

---

**Algorithm 1** Generalized EM for unsupervised binary label aggregation (CI, latent factors, Ising)

---

1: **Input:** votes $J_i \in \{0,1\}^K$ for $i = 1, \ldots, n$; model family $P_\Theta(J \mid Y)$; initialization $\Theta^{(0)}$; tolerances.
2: **for** $t = 0, 1, 2, \ldots$ until convergence **do**
3:     **E-step (posterior labels):** For each item $i$, compute class scores

$$\widehat{\ell}_{iy}^{(t)} \approx \log P_{\Theta^{(t)}}(J_i \mid Y_i = y), \qquad y \in \{0,1\},$$

    using an exact evaluator (when tractable) or an approximation (e.g. mean-field, loopy BP, Monte Carlo, or a variational bound). Set

$$\gamma_i^{(t)} \leftarrow P_{\Theta^{(t)}}(Y_i = 1 \mid J_i) = \sigma\Big( \mathrm{logit}(\pi^{(t)}) + \widehat{\ell}_{i1}^{(t)} - \widehat{\ell}_{i0}^{(t)} \Big), \tag{25}$$

    where $\sigma(u) = (1 + e^{-u})^{-1}$.
4:     **M-step (parameter update):** Update the class prior (MLE form)

$$\pi^{(t+1)} \leftarrow \frac{1}{n} \sum_{i=1}^{n} \gamma_i^{(t)} \qquad \text{(or MAP update if a Beta prior is used).}$$

    Update remaining parameters by (approximately) maximizing the weighted objective

$$\Theta^{(t+1)} \in \arg\max_{\Theta} \sum_{i=1}^{n} \Big[ \gamma_i^{(t)} \log P_\Theta(J_i \mid Y_i = 1) + (1 - \gamma_i^{(t)}) \log P_\Theta(J_i \mid Y_i = 0) \Big] + \log p(\Theta), \tag{26}$$

    using a model-appropriate solver (closed-form updates, gradient methods, pseudo-likelihood, etc.).
5: **end for**
6: **Output:** parameters $\widehat{\Theta}$ and posteriors $\widehat{\gamma}_i$; predicted labels $\widehat{Y}_i = \mathbf{1}\{\widehat{\gamma}_i \geq 1/2\}$.

---

Finally, we remark that for our experiments in Sections 4.1 and G, when using specific instantiations of Algorithm 1, we manually tune the hyperparameters (if any) for best performance. More principled ways to set them include general purpose procedures like cross-validation.

### E.2. Illustration of Theorem 2 via Algorithm 1

For the sake of the reader's convenience, we connect the magnetization-threshold rule in Theorem 2 to posterior-threshold prediction as in Algorithm 1. Define for each item $i$ the (spin) magnetization

$$M_i \; := \; M_K(J_i) \; := \; \frac{1}{K} \sum_{j=1}^{K} X_{ij}, \qquad X_{ij} := 2J_{ij} - 1 \in \{\pm 1\}.$$

Because the Curie-Weiss model is exchangeable, the Bayes posterior depends on the votes $J_i$ only through $M_i$ (equivalently through the vote fraction). In particular, if we form a (possibly approximate/plug-in) posterior based on the statistic $|M_i|$,

$$\widehat{\gamma}_i \; := \; \mathrm{P}_{\widehat{\Theta}}(Y_i = 1 \mid |M_i|) = \frac{\pi\, \widehat{p}_1(|M_i|)}{\pi\, \widehat{p}_1(|M_i|) + (1 - \pi)\, \widehat{p}_0(|M_i|)},$$

where $\widehat{p}_y(\cdot)$ denotes the (estimated) class-conditional pmf/density of $|M_K|$ under $Y = y$, then the usual posterior-threshold prediction is

$$\widehat{Y}_i \; = \; \mathbf{1}\{\widehat{\gamma}_i \geq 1/2\} \quad \Longleftrightarrow \quad \log \frac{\pi}{1 - \pi} + \log \frac{\widehat{p}_1(|M_i|)}{\widehat{p}_0(|M_i|)} \; \geq \; 0.$$

In the Curie-Weiss regimes used for the separation result in Theorem 2, the log-likelihood ratio $m \mapsto \log \frac{\widehat{p}_1(m)}{\widehat{p}_0(m)}$ is (asymptotically) increasing in $m \in [0, 1]$, so there exists a threshold $t \in (0, 1)$ such that

$$\mathbf{1}\{\widehat{\gamma}_i \geq 1/2\} \; = \; \mathbf{1}\{|M_i| \geq t\}.$$

Thus the magnetization classifier $\tilde{g}_K(J) = \mathbf{1}\{|M_K(J)| \geq t\}$ is exactly a posterior-threshold rule of the form $\widehat{Y}_i = \mathbf{1}\{\widehat{\gamma}_i \geq 1/2\}$ when $\widehat{\gamma}_i$ is computed from (or approximated by) evidence in $|M_K|$.

# F. Proofs from Section 3

## F.1. Proof of Theorem 1

**Theorem 1** (Nonvanishing Bayes vs. CI Separation for Class-conditional Ising Models). *Fix a prior* $\mathrm{P}(Y = 1) = \pi \in (0,1)$. *For each* $K \geq 1$, *let* $J = (J_1, \ldots, J_K) \in \{0,1\}^K$ *denote the* $K$ *judges' votes for a single item and define the recoded spins* $X_j := 2J_j - 1 \in \{-1, +1\}$ *and let* $M_K := \frac{1}{K}\sum_{j=1}^K X_j \in \left\{-1, -1+\frac{2}{K}, \ldots, 1\right\}$. *Assume the following* class-conditional Curie-Weiss Ising *model: there exist constants* $0 < \beta_0 < 1 < \beta_1$ *such that, conditional on* $Y = y \in \{0,1\}$,

$$\mathrm{P}(X = x \mid Y = y) = \frac{1}{Z_K(\beta_y)} \exp\left(\frac{\beta_y}{2K}\left(\sum_{j=1}^K x_j\right)^2\right), \tag{6}$$

*for* $x \in \{-1, +1\}^K$. *Equivalently (writing* $x_j = 2j_j - 1$), *this is a special case of the* $\{0,1\}$-*Ising form* (1) *with* $h_j^{(y)} = -2\beta_y\left(1 - \frac{1}{K}\right)$ *and* $W_{jk}^{(y)} = \frac{4\beta_y}{K}$ $(j \neq k)$, *up to an additive constant absorbed into* $Z^{(y)}$.

*Let* $g_K^\star$ *be the Bayes-optimal predictor under the true model* (6). *Let* $g_K^{\mathrm{ind}}$ *be the population CI-predictor that replaces the true joint by the product of true one-dimensional marginals, i.e.,* $\mathrm{P}^{\mathrm{ind}}(J = j \mid Y = y) := \prod_{r=1}^K q_y^{j_r}(1 - q_y)^{1-j_r}$ *with* $q_y := \mathrm{P}(J_r = 1 \mid Y = y)$ *(independent of* $r$), *and then thresholds the induced posterior* $\mathrm{P}^{\mathrm{ind}}(Y = 1 \mid J)$ *at* $1/2$. *Then the following results hold:*

1. *(CI Collapses to the Prior) For every* $K$ *and* $y \in \{0,1\}$, *one has* $q_y = \frac{1}{2}$. *Consequently,* $\mathrm{P}^{\mathrm{ind}}(Y = 1 \mid J) = \pi$ *for all* $J$, *so* $g_K^{\mathrm{ind}}(J) \equiv \mathbf{1}\{\pi \geq \frac{1}{2}\}$, *and* $R(g_K^{\mathrm{ind}}) = \min\{\pi, 1 - \pi\}, \forall K$.

2. *(Bayes Risk Vanishes) Let* $m_\star = m_\star(\beta_1) \in (0,1)$ *denote the unique positive solution to* $m = \tanh(\beta_1 m)$. *Then for any fixed threshold* $t \in (0, m_\star^2)$, *the quadratic statistic test:*

$$\tilde{g}_K(J) := \mathbf{1}\{M_K(J)^2 \geq t\},$$

$$M_K(J) = \frac{1}{K}\sum_{j=1}^K (2J_j - 1),$$

   *satisfies* $R(\tilde{g}_K) \to 0$ *as* $K \to \infty$. *Hence* $R(g_K^\star) \to 0$ *as* $K \to \infty$.

3. *(Nonvanishing Separation) We have:*

$$\lim_{K \to \infty}\left(R(g_K^{\mathrm{ind}}) - R(g_K^\star)\right) = \min\{\pi, 1 - \pi\} > 0.$$

*Proof.* We start with the proof of assertion (1).

Fix $y \in \{0,1\}$ and $K$. Under (6), the density depends on $x$ only through $\left(\sum_j x_j\right)^2$, and hence is invariant under the global spin-flip $x \mapsto -x$:

$$\mathrm{P}(X = x \mid Y = y) = \mathrm{P}(X = -x \mid Y = y) \qquad \forall x \in \{-1, +1\}^K.$$

Therefore, for each coordinate $r$,

$$\mathbb{E}[X_r \mid Y = y] = \sum_x x_r\,\mathrm{P}(X = x \mid Y = y) = \sum_x (-x_r)\,\mathrm{P}(X = -x \mid Y = y) = -\mathbb{E}[X_r \mid Y = y],$$

so $\mathbb{E}[X_r \mid Y = y] = 0$ and hence $\mathrm{P}(X_r = +1 \mid Y = y) = \mathrm{P}(X_r = -1 \mid Y = y) = \frac{1}{2}$. Since $J_r = (X_r + 1)/2$, we get $q_y = \mathrm{P}(J_r = 1 \mid Y = y) = \frac{1}{2}$ for both $y = 0, 1$.

With $q_0 = q_1 = \frac{1}{2}$, the CI likelihoods coincide:

$$\mathrm{P}^{\mathrm{ind}}(J \mid Y = 1) = \mathrm{P}^{\mathrm{ind}}(J \mid Y = 0) = 2^{-K} \qquad \forall J,$$

so Bayes' rule under the CI model gives $P^{\text{ind}}(Y = 1 \mid J) = \pi$ for all $J$ and $g_K^{\text{ind}}(J) \equiv \mathbf{1}\{\pi \geq \frac{1}{2}\}$. Its (true) misclassification risk is then

$$R(g_K^{\text{ind}}) = P(g_K^{\text{ind}} \neq Y) = \begin{cases} P(Y = 0) = 1 - \pi, & \pi \geq \frac{1}{2}, \\ P(Y = 1) = \pi, & \pi < \frac{1}{2}, \end{cases} = \min\{\pi, 1 - \pi\}.$$

This proves assertion (1).

Before proving assertion (2), we require some intermediate results.

We start by proving a magnetization representation and a uniform combinatorial bound for the Ising model case. Fix $\beta > 0$ and consider the Curie-Weiss law

$$P_\beta(X = x) = \frac{1}{Z_K(\beta)} \exp\Big(\frac{\beta}{2K}\Big(\sum_{j=1}^K x_j\Big)^2\Big).$$

For $m \in \{-1, -1 + 2/K, \ldots, 1\}$, let $N_K(m)$ be the number of configurations with magnetization $M_K = m$. Writing $r = \#\{j : x_j = +1\} = \frac{K(1+m)}{2}$, we have $N_K(m) = \binom{K}{r}$ and

$$P_\beta(M_K = m) = \frac{1}{Z_K(\beta)} \binom{K}{\frac{K(1+m)}{2}} \exp\Big(\frac{\beta K}{2} m^2\Big), \tag{27}$$

where $Z_K(\beta)$ is the corresponding normalizer (sum of the numerator over all admissible $m$).

Define the binary entropy (natural logs)

$$H(p) := -p \log p - (1 - p) \log(1 - p), \qquad p \in [0, 1],$$

and the mean-field objective

$$\Phi_\beta(m) := H\Big(\frac{1 + m}{2}\Big) + \frac{\beta}{2} m^2, \qquad m \in [-1, 1].$$

A standard consequence of Stirling's bounds is the uniform approximation

$$\log\Big(\frac{K}{\frac{K(1+m)}{2}}\Big) = K\, H\Big(\frac{1 + m}{2}\Big) + O(\log K), \qquad \text{uniformly over } m \in \Big\{-1, -1 + \frac{2}{K}, \ldots, 1\Big\}. \tag{28}$$

Combining (27)–(28) yields

$$P_\beta(M_K = m) = \frac{\exp\big(K\Phi_\beta(m) + O(\log K)\big)}{\sum_{m'} \exp\big(K\Phi_\beta(m') + O(\log K)\big)}, \tag{29}$$

where the sum runs over the $(K + 1)$ admissible magnetization values and the $O(\log K)$ term is uniform in $m, m'$.

We next identify the location of the maximizers of $\Phi_\beta$.

**Lemma 1.** *Let $\beta > 0$ and $\Phi_\beta(m) = H(\frac{1+m}{2}) + \frac{\beta}{2} m^2$ on $[-1, 1]$.*

1. *If $\beta < 1$, then $\Phi_\beta$ is strictly concave on $(-1, 1)$ and has a unique maximizer at $m = 0$.*

2. *If $\beta > 1$, then there exists a unique $m_\star(\beta) \in (0, 1)$ solving $m = \tanh(\beta m)$. Moreover, $\Phi_\beta$ has exactly two global maximizers at $\pm m_\star(\beta)$, and $m = 0$ is a strict local minimum.*

*Proof.* For $m \in (-1, 1)$, differentiating gives

$$\frac{d}{dm} H\Big(\frac{1 + m}{2}\Big) = \frac{1}{2} \log \frac{1 - m}{1 + m} = -\operatorname{arctanh}(m),$$

hence

$$\Phi_\beta'(m) = -\operatorname{arctanh}(m) + \beta m, \qquad \Phi_\beta''(m) = -\frac{1}{1 - m^2} + \beta.$$

If $\beta < 1$, then for all $m \in (-1, 1)$,

$$\Phi_\beta''(m) \leq -1 + \beta < 0$$

(since $1/(1 - m^2) \geq 1$), so $\Phi_\beta$ is strictly concave and has at most one critical point. Because $\Phi_\beta$ is even, $\Phi_\beta'(0) = 0$, so $m = 0$ is the unique maximizer.

If $\beta > 1$, then $\Phi_\beta''(0) = \beta - 1 > 0$, so $m = 0$ is a strict local minimum. Critical points satisfy $\Phi_\beta'(m) = 0$, i.e. $\operatorname{arctanh}(m) = \beta m$, equivalently $m = \tanh(\beta m)$. Define $f(m) := \tanh(\beta m) - m$ on $[0, 1]$. Then $f(0) = 0$ and $f'(0) = \beta - 1 > 0$, while $f(1) = \tanh(\beta) - 1 < 0$. By continuity, there exists at least one root in $(0, 1)$. Moreover, $f'(m) = \beta(1 - \tanh^2(\beta m)) - 1 = \beta(1 - m^2) - 1$ at a root (since then $\tanh(\beta m) = m$), and $m \mapsto \beta(1 - m^2) - 1$ is strictly decreasing on $[0, 1]$. This implies $f$ is strictly concave on any interval where it is positive and strictly decreasing once $m$ is large enough; in particular, $f$ can cross zero at most once in $(0, 1)$, so the positive root is unique; call it $m_\star(\beta)$. By symmetry, $-m_\star(\beta)$ is also a critical point.

At $m = \pm m_\star(\beta)$, we have $\beta(1 - m_\star^2(\beta)) < 1$ (equivalently the slope of $\tanh(\beta m)$ at the intersection is $< 1$), so $\Phi_\beta''(\pm m_\star(\beta)) = -\frac{1}{1 - m_\star(\beta)^2} + \beta < 0$, hence $\pm m_\star(\beta)$ are strict local maxima. Since $\Phi_\beta$ is continuous on compact $[-1, 1]$, it attains global maxima; the only candidates are critical points and endpoints. The endpoints satisfy $H(\frac{1 \pm 1}{2}) = 0$, hence $\Phi_\beta(\pm 1) = \frac{\beta}{2}$, while $\Phi_\beta(\pm m_\star(\beta)) > \Phi_\beta(0) = \log 2$ for $\beta > 1$ (indeed $\pm m_\star(\beta)$ are maxima and 0 is a local minimum). Thus the global maxima are exactly $\pm m_\star(\beta)$. $\qquad\square$

We next show exponential concentration of $M_K$ under $\beta_0$ and $\beta_1$. Our approach for proving concentration for Curie-Weiss models is motivated by Friedli & Velenik (2017, Chapter 2). While more sophisticated approaches are available to obtain sharp concentration bounds, we use this simpler approach as a coarse concentration result suffices to prove our separation result of interest.

**Lemma 2** (Concentration under $\beta < 1$). *Fix $\beta \in (0, 1)$ and $\delta \in (0, 1)$. Then there exists $c = c(\beta, \delta) > 0$ such that*

$$\mathrm{P}_\beta\big(|M_K| \geq \delta\big) \leq \exp(-cK) \quad \text{for all sufficiently large } K.$$

*Proof.* By assertion (1) of Lemma 1, $\Phi_\beta$ has unique maximizer at 0. By continuity of $\Phi_\beta$ and compactness of $\{m \in [-1, 1] : |m| \geq \delta\}$,

$$\Delta := \Phi_\beta(0) - \sup_{|m| \geq \delta} \Phi_\beta(m) > 0.$$

Using (29), for the numerator we bound

$$\sum_{|m| \geq \delta} \exp\big(K\Phi_\beta(m) + O(\log K)\big) \leq (K + 1) \exp\big(K \sup_{|m| \geq \delta} \Phi_\beta(m) + O(\log K)\big),$$

and for the denominator,

$$\sum_{m'} \exp\big(K\Phi_\beta(m') + O(\log K)\big) \geq \exp\big(K\Phi_\beta(0) - O(\log K)\big).$$

Taking the ratio gives

$$\mathrm{P}_\beta(|M_K| \geq \delta) \leq (K + 1) \exp\big(-K\Delta + O(\log K)\big) = \exp\big(-K\Delta + O(\log K)\big),$$

which is $\leq \exp(-cK)$ for all large $K$ for any $c < \Delta$. $\qquad\square$

**Lemma 3** (Concentration under $\beta > 1$). *Fix $\beta > 1$ and let $m_\star = m_\star(\beta) \in (0, 1)$ be the unique positive solution to $m = \tanh(\beta m)$. For any $\varepsilon \in (0, m_\star)$, there exists $c = c(\beta, \varepsilon) > 0$ such that*

$$\mathrm{P}_\beta\big(\big||M_K| - m_\star\big| \geq \varepsilon\big) \leq \exp(-cK) \quad \text{for all sufficiently large } K.$$

*Proof.* By Lemma 1(2), $\Phi_\beta$ has exactly two strict global maximizers at $\pm m_\star$. Define the closed set

$$F_\varepsilon := \{m \in [-1, 1] : \big||m| - m_\star\big| \geq \varepsilon\}.$$

Since $\pm m_\star \notin F_\varepsilon$ and $\Phi_\beta$ is continuous, compactness implies

$$\Delta := \Phi_\beta(m_\star) - \sup_{m \in F_\varepsilon} \Phi_\beta(m) > 0.$$

The same numerator/denominator bounding argument used in Lemma 2 applied to the event $\{M_K \in F_\varepsilon\}$ yields

$$\mathrm{P}_\beta(M_K \in F_\varepsilon) \le (K+1)\exp\big(-K\Delta + O(\log K)\big) = \exp\big(-K\Delta + O(\log K)\big) \le \exp(-cK)$$

for any $c < \Delta$ and all sufficiently large $K$. $\qquad\square$

We are now ready the prove assertion (2).

Consider the joint model from Theorem 1 with $\beta_0 \in (0,1)$ and $\beta_1 > 1$. Fix any $t \in (0, m_\star(\beta_1)^2)$ and define $\tilde{g}_K(J) = \mathbf{1}\{M_K(J)^2 \ge t\}$.

*Type I error under $Y = 0$:* Conditional on $Y = 0$, the spins follow $\mathrm{P}_{\beta_0}$, hence by Lemma 2 with $\delta = \sqrt{t}$,

$$\mathrm{P}(\tilde{g}_K = 1 \mid Y = 0) = \mathrm{P}_{\beta_0}(M_K^2 \ge t) = \mathrm{P}_{\beta_0}(|M_K| \ge \sqrt{t}) \longrightarrow 0.$$

*Type II error under $Y = 1$:* Conditional on $Y = 1$, the spins follow $\mathrm{P}_{\beta_1}$. Let $m_\star = m_\star(\beta_1)$ and set $\varepsilon := \frac{m_\star - \sqrt{t}}{2} > 0$. If $M_K^2 < t$, then $|M_K| < \sqrt{t} = m_\star - 2\varepsilon$, hence $\big||M_K| - m_\star\big| \ge 2\varepsilon$. Therefore, by Lemma 3 (applied with $2\varepsilon$),

$$\mathrm{P}(\tilde{g}_K = 0 \mid Y = 1) = \mathrm{P}_{\beta_1}(M_K^2 < t) \le \mathrm{P}_{\beta_1}\big(\big||M_K| - m_\star\big| \ge 2\varepsilon\big) \longrightarrow 0.$$

Combining the two conditional errors,

$$R(\tilde{g}_K) = \pi\,\mathrm{P}(\tilde{g}_K = 0 \mid Y = 1) + (1 - \pi)\,\mathrm{P}(\tilde{g}_K = 1 \mid Y = 0) \longrightarrow 0.$$

Since the Bayes predictor $g_K^\star$ minimizes misclassification risk under the true joint law,

$$R(g_K^\star) \le R(\tilde{g}_K) \to 0,$$

proving assertion (2).

Finally, we immediately have the separation limit stated in assertion (3). Indeed, assertion (1) gives $R(g_K^{\mathrm{ind}}) = \min\{\pi, 1-\pi\}$ for all $K$, while assertion (2) gives $R(g_K^\star) \to 0$. Hence

$$\lim_{K \to \infty}\big(R(g_K^{\mathrm{ind}}) - R(g_K^\star)\big) = \min\{\pi, 1-\pi\} - 0 = \min\{\pi, 1-\pi\} > 0.$$

This proves assertion (3) and completes the overall proof. $\qquad\square$

### F.2. Proof of Theorem 2

**Theorem 2** (Curie-Weiss separation with informative marginals). *Let $Y \in \{0,1\}$ with $\mathrm{P}(Y = 1) = \pi \in (0,1)$. For each $K \ge 1$, define spins $X = (X_1, \ldots, X_K) \in \{-1, +1\}^K$ and votes $J_j = (X_j + 1)/2 \in \{0,1\}$. Let the class-conditional laws of $X$ be Curie-Weiss models with (possibly $K$-dependent) external fields:*

$$\mathrm{P}(X = x \mid Y = y)$$
$$= \frac{1}{Z_K^{(y)}}\exp\Big(\frac{\beta_y}{2K}\Big(\sum_{j=1}^K x_j\Big)^2 + h_{y,K}\sum_{j=1}^K x_j\Big), \tag{7}$$

*for $y \in \{0,1\}$. Assume parameters satisfy:*

1. *(High-temperature Class) $\beta_0 \in (0,1)$ and $h_{0,K} \equiv h_0 < 0$ is a fixed negative constant;*

2. *(Low-temperature Class with Weak Symmetry Breaking) $\beta_1 > 1$ and $h_{1,K} = c/K$ with some fixed $c > 0$.*

Let $M_K := \frac{1}{K} \sum_{j=1}^{K} X_j \in [-1, 1]$ be the magnetization. Let $m_0 \in (-1, 0)$ be the unique solution of the mean-field equation $m_0 = \tanh(\beta_0 m_0 + h_0)$, and let $m_\star = m_\star(\beta_1) \in (0, 1)$ be the unique positive solution of $m_\star = \tanh(\beta_1 m_\star)$. Define

$$p := \frac{e^{cm_\star}}{e^{cm_\star} + e^{-cm_\star}} = \sigma(2cm_\star) \in \left(\frac{1}{2}, 1\right),$$

$$q_0 := P(J_1 = 1 \mid Y = 0) = \frac{1 + m_0}{2} \in \left(0, \frac{1}{2}\right),$$

$$q_1 := P(J_1 = 1 \mid Y = 1) = \frac{1 + (2p-1)m_\star}{2} \in \left(\frac{1}{2}, 1\right).$$

Assume additionally that

$$\frac{1 - m_\star}{2} < q_0 \iff m_\star > 1 - 2q_0. \tag{8}$$

Let $g_K^\star$ denote the Bayes predictor under the true model (7). Let $g_K^{\mathrm{ind}}$ denote the population CI-predictor that replaces $P(J \mid Y = y)$ by the product of the true marginals $\prod_{j=1}^{K} q_y^{J_j} (1 - q_y)^{1 - J_j}$. Then, the following hold:

1. **(Informative Marginals)** Each judge is individually better than random: $q_0 < \frac{1}{2} < q_1$ (equivalently, specificity $1 - q_0 > \frac{1}{2}$ and sensitivity $q_1 > \frac{1}{2}$).

2. **(Bayes Risk Vanishes)** For any fixed threshold $t$ satisfying $|m_0| < t < m_\star$, the aggregator $\tilde{g}_K(J) := \mathbf{1}\{|M_K| \geq t\}$ has $R(\tilde{g}_K) \to 0$ as $K \to \infty$. Consequently, $R(g_K^\star) \to 0$.

3. **(CI Remains Bounded away from Bayes)** As $K \to \infty$, we have $R(g_K^{\mathrm{ind}}) \to \pi(1-p)$, and hence $\lim_{K\to\infty} \left(R(g_K^{\mathrm{ind}}) - R(g_K^\star)\right) \geq \pi(1-p) > 0$.

*Proof.* The proof is similar to that of Theorem 1. We start with the magnetization representation. For $\beta > 0$ and field $h$, the Curie-Weiss probability of $M_K = m$ (with $m \in \{-1, -1 + \frac{2}{K}, \ldots, 1\}$) can be written as

$$P_{\beta, h}(M_K = m) = \frac{1}{Z_K(\beta, h)} \binom{K}{\frac{K(1+m)}{2}} \exp\left(\frac{\beta K}{2} m^2 + hKm\right), \tag{30}$$

where $Z_K(\beta, h)$ normalizes the mass over all admissible $m$. Let $H(p) = -p \log p - (1-p) \log(1-p)$ and define

$$\Phi_\beta(m) := H\left(\frac{1+m}{2}\right) + \frac{\beta}{2} m^2, \qquad m \in [-1, 1].$$

Stirling's formula yields

$$\log \binom{K}{\frac{K(1+m)}{2}} = K H\left(\frac{1+m}{2}\right) + O(\log K),$$

uniformly over admissible $m$, hence

$$P_{\beta, h}(M_K = m) \propto \exp\left(K \Phi_\beta(m) + hKm + O(\log K)\right). \tag{31}$$

We now compute the marginals correponding to item (1). By exchangeability, $\mathbb{E}[X_1 \mid Y = y] = \mathbb{E}[M_K \mid Y = y]$. Under $(\beta_0, h_0)$ with $\beta_0 \in (0, 1)$ and fixed $h_0 < 0$, the standard mean-field analysis implies that $M_K \to m_0$ in probability, where $m_0$ is the unique solution to $m = \tanh(\beta_0 m + h_0)$. Hence $\mathbb{E}[M_K \mid Y = 0] \to m_0 < 0$ and

$$q_0 = P(J_1 = 1 \mid Y = 0) = P(X_1 = +1 \mid Y = 0) = \frac{1 + \mathbb{E}[X_1 \mid Y = 0]}{2} \to \frac{1 + m_0}{2} < \frac{1}{2}.$$

Under $(\beta_1, h_{1,K})$ with $\beta_1 > 1$ and $h_{1,K} = c/K$, we show below (Step 2) that $M_K$ converges in distribution to a two-point mixture $p\, \delta_{m_\star} + (1-p)\, \delta_{-m_\star}$ with $p = \sigma(2cm_\star) \in (1/2, 1)$. Therefore $\mathbb{E}[M_K \mid Y = 1] \to (2p-1)m_\star > 0$ and

$$q_1 = P(J_1 = 1 \mid Y = 1) = \frac{1 + \mathbb{E}[X_1 \mid Y = 1]}{2} \to \frac{1 + (2p-1)m_\star}{2} > \frac{1}{2}.$$

This proves item (1).

We now examine the magnetization limits under the two classes. Consider $Y = 0$. Since $\beta_0 < 1$, $\Phi_{\beta_0}$ is strictly concave on $(-1, 1)$ and has a unique maximizer; adding the linear term $h_0 m$ preserves uniqueness. Thus the exponent in (31) has a unique global maximizer at $m_0$, and a standard Laplace argument yields

$$M_K \xrightarrow[K \to \infty]{\text{P}} m_0. \tag{32}$$

Now consider $Y = 1$. Here $h_{1,K} = c/K$, so $h_{1,K} K = c$ and (31) becomes

$$P(M_K = m \mid Y = 1) \propto \exp\Big(K\Phi_{\beta_1}(m) + cm + O(\log K)\Big). \tag{33}$$

For $\beta_1 > 1$, $\Phi_{\beta_1}$ has exactly two global maximizers at $\pm m_\star$ (where $m_\star = \tanh(\beta_1 m_\star)$). Fix $\varepsilon \in (0, m_\star)$ and define neighborhoods

$$U_+ := [m_\star - \varepsilon, m_\star + \varepsilon], \qquad U_- := [-m_\star - \varepsilon, -m_\star + \varepsilon], \qquad F_\varepsilon := [-1, 1] \setminus (U_+ \cup U_-).$$

By continuity and strict maximality at $\pm m_\star$, there exists $\Delta(\varepsilon) > 0$ such that $\sup_{m \in F_\varepsilon} \Phi_{\beta_1}(m) \leq \Phi_{\beta_1}(m_\star) - \Delta(\varepsilon)$. Using (33) and the same numerator/denominator bounding argument as in Lemma 3, we obtain exponential concentration:

$$P\big(M_K \in F_\varepsilon \mid Y = 1\big) \leq e^{-c_1 K} \quad \text{for some } c_1 = c_1(\varepsilon) > 0. \tag{34}$$

Hence $|M_K| \to m_\star$ in probability under $Y = 1$.

It remains to identify the limiting *mixture weights*. Let $A_{K,+}$ (resp. $A_{K,-}$) denote the total unnormalized mass in $U_+$ (resp. $U_-$) in (33). On $U_+$ we have $m = m_\star + o(1)$ and on $U_-$ we have $m = -m_\star + o(1)$, while $\Phi_{\beta_1}(m) = \Phi_{\beta_1}(m_\star) + o(1)$ in both neighborhoods. Therefore

$$\frac{A_{K,+}}{A_{K,-}} = \frac{\sum_{m \in U_+} \exp\big(K\Phi_{\beta_1}(m) + cm + O(\log K)\big)}{\sum_{m \in U_-} \exp\big(K\Phi_{\beta_1}(m) + cm + O(\log K)\big)} \longrightarrow \exp(2cm_\star),$$

because the leading $K\Phi_{\beta_1}(m_\star)$ contributions cancel and the remaining $cm$ term evaluates to $\pm cm_\star$. Combining with (34) implies

$$P(M_K > 0 \mid Y = 1) \to \frac{e^{cm_\star}}{e^{cm_\star} + e^{-cm_\star}} = p, \qquad P(M_K < 0 \mid Y = 1) \to 1 - p,$$

and thus $M_K \Rightarrow p\,\delta_{m_\star} + (1 - p)\,\delta_{-m_\star}$ under $Y = 1$.

We are now in the position to show that Bayes risk vanishes, as claimed in item (2). Pick any $t$ with $|m_0| < t < m_\star$ and define $\tilde{g}_K(J) = \mathbf{1}\{|M_K| \geq t\}$. Under $Y = 0$, (32) implies $|M_K| < t$ with probability $\to 1$. Under $Y = 1$, (34) implies $|M_K| > t$ with probability $\to 1$. Therefore $P(\tilde{g}_K \neq Y) \to 0$, i.e. $R(\tilde{g}_K) \to 0$. Since $g_K^\star$ minimizes risk, $R(g_K^\star) \leq R(\tilde{g}_K) \to 0$.

Next, we move on to proving item (3). Under the CI model with marginals $(q_0, q_1)$, the (oracle) CI log-likelihood ratio depends only on $S_K = \sum_{j=1}^{K} J_j$ or equivalently $s_K = S_K/K$:

$$\log \frac{P_{\text{ind}}(J \mid Y = 1)}{P_{\text{ind}}(J \mid Y = 0)} = K\Big(s_K \log \frac{q_1}{q_0} + (1 - s_K) \log \frac{1 - q_1}{1 - q_0}\Big).$$

Since $q_1 > q_0$, the function of $s$ in parentheses is strictly increasing and has a unique root $s_{\text{thr}} \in (q_0, q_1)$. Thus the CI-predictor satisfies (for all large $K$, ignoring the vanishing prior term at scale $K$)

$$g_K^{\text{ind}}(J) = \mathbf{1}\{s_K \geq s_{\text{thr}}\}.$$

Under $Y = 0$, we have $s_K = (1 + M_K)/2 \to (1 + m_0)/2 = q_0 < s_{\text{thr}}$ in probability, so $P(g_K^{\text{ind}} = 1 \mid Y = 0) \to 0$.

Under $Y = 1$, we have $s_K \to (1 \pm m_\star)/2$ depending on the phase. By (8), the negative-phase limit $(1 - m_\star)/2$ is strictly smaller than $q_0 < s_{\text{thr}}$, so $g_K^{\text{ind}} \to 0$ on the negative phase; while $(1 + m_\star)/2 > s_{\text{thr}}$ so $g_K^{\text{ind}} \to 1$ on the positive phase. Therefore

$$P(g_K^{\text{ind}} = 0 \mid Y = 1) \to P(\text{negative phase} \mid Y = 1) = 1 - p,$$

and hence

$$R(g_K^{\text{ind}}) = \pi\, P(g_K^{\text{ind}} = 0 \mid Y = 1) + (1 - \pi)\, P(g_K^{\text{ind}} = 1 \mid Y = 0) \to \pi(1 - p).$$

Together with $R(g_K^\star) \to 0$, this yields the claimed nonvanishing excess-risk lower bound. $\square$

### F.3. A note on Theorem 1 and Theorem 2

We remark that from our proofs it could be seen that while our main statements are stated asymptotically, the underlying mechanism is already finite-$K$: the separation develops as soon as the relevant low-dimensional statistic concentrates. Concretely, for Theorem 1 and Theorem 2, if we fix a threshold $u$ strictly between the class-typical magnetization levels (for Theorem 1, $u \in (0, m_\star)$, equivalently $M_K^2 \geq u^2$; for Theorem 2, $|m_0| < u < m_\star$), then the same free-energy comparison as in the proofs yields

$$R(\widetilde{g}_K) \leq (1 - \pi)e^{-c_0(u)K} + \pi e^{-c_1(u)K}$$

for some $c_0(u), c_1(u) > 0$, so $R(g_K^\star) \leq R(\widetilde{g}_K)$ decays exponentially in $K$. Hence, in Theorem 1,

$$R(g_K^{\mathrm{ind}}) - R(g_K^\star) \geq \min\{\pi, 1 - \pi\} - Ce^{-cK},$$

since $R(g_K^{\mathrm{ind}}) = \min\{\pi, 1 - \pi\}$ exactly for every $K$; and in Theorem 2 one similarly obtains

$$R(g_K^{\mathrm{ind}}) - R(g_K^\star) \geq \pi(1 - p) - o(1) - Ce^{-cK},$$

where the $o(1)$ term is the finite-$K$ correction in the positive/negative phase weights and can be made explicit by a sharper Laplace expansion.

In this sense, the practically relevant issue is not asymptoticity per se, but how sharply the sufficient statistic concentrates at the panel sizes of interest.

We alsoe emphasize that the same qualitative separation should extend more generally to block-wise mean-field Ising models (Berthet et al., 2019) (also called as multi-species Curie–Weiss models in the literature), where the judges are partitioned into finitely many groups $G_1, \ldots, G_B$ and $W_{jk}^{(y)} = B_{ab}^{(y)}/K$ whenever $j \in G_a$ and $k \in G_b$. In that setting, the scalar magnetization $M_K$ is replaced by the vector of block magnetizations $M_{a,K} = |G_a|^{-1} \sum_{j \in G_a}(2J_j - 1)$, and the relevant Bayes score is a quadratic function of this low-dimensional order parameter. The main technical ingredient is again concentration: under standard regularity conditions such as fixed $B$, block proportions bounded away from zero, bounded couplings, and a class-conditional block free energy with a finite set of separated maximizers and a positive gap away from them, the vector $(M_{1,K}, \ldots, M_{B,K})$ concentrates exponentially near class-dependent limit points. Once this concentration is available, the same separation proof approach goes through: a dependence-aware predictor based on the appropriate quadratic form in the block magnetizations achieves vanishing risk, whereas a CI predictor, which only uses one-dimensional marginals, can still incur non-vanishing error when the classes differ primarily through within-block and between-block correlation patterns rather than marginal vote rates. Thus, the key requirement is not full Curie–Weiss symmetry per se, but sharp concentration of a low-dimensional set of order parameters.

## G. Numerical Simulations

### G.1. CI Judges: Uniform vs. Weighted Majority Vote

We present sanity-check experiment in the *conditionally independent* (CI) setting, where the classical Dawid-Skene family is well-specified. Each item has a latent label $Y_i \in \{0, 1\}$ and $K = 6$ judges produce independent votes $J_{ij} \in \{0, 1\}$ conditional on $Y_i$. Judge $j$ is characterized by sensitivity $\alpha_j = \mathrm{P}(J_{ij} = 1 \mid Y_i = 1)$ and specificity $\beta_j = \mathrm{P}(J_{ij} = 0 \mid Y_i = 0)$. For each setup we generate $n = 200$ items from the CI model with the *true* $(\alpha, \beta)$ listed in Table 3. We consider four setups spanning strong annotators (Setup 1) and heterogeneous, partially unreliable annotators (Setups 2–4). We evaluate two aggregation rules: (i) Uniform majority vote (Uniform MV), which predicts the class supported by a strict majority of votes, and (ii) Weighted majority vote learned by the EM approach in Algorithm 1, where we fit the asymmetric CI model with Beta priors on $(\alpha_j, \beta_j)$ and then apply the induced Bayes-optimal linear rule (equivalently, a weighted vote with weights given by the estimated log-odds contributions of each judge.

Table 4 reports average 0–1 accuracy across runs for each setup. Two trends are consistent with theory. First, when all judges are strong and roughly exchangeable (Setup 1), uniform MV is already near-optimal and EM-based weighting provides only a small gain. Second, in the heterogeneous regimes (Setups 2–4), uniform MV can substantially underperform because it treats all judges as equally informative and ignores asymmetric error patterns. In contrast, CI-WMV learns to up-weight reliable judges and down-weight weak (or effectively adversarial) ones, yielding large improvements: in Setup 2 accuracy increases from 0.597 to 0.726, and in Setup 3 from 0.520 to 0.611. Even in Setup 4, where most judges are reasonably informative but still heterogeneous, CI-WMV improves over uniform MV (0.930 vs. 0.917). Overall, these CI experiments

| Setup | True sensitivities | | | | | | True specificities | | | | | |
|---|---|---|---|---|---|---|---|---|---|---|---|---|
| | $\alpha_1$ | $\alpha_2$ | $\alpha_3$ | $\alpha_4$ | $\alpha_5$ | $\alpha_6$ | $\beta_1$ | $\beta_2$ | $\beta_3$ | $\beta_4$ | $\beta_5$ | $\beta_6$ |
| #1 | 0.90 | 0.90 | 0.90 | 0.90 | 0.90 | 0.90 | 0.90 | 0.90 | 0.90 | 0.95 | 0.90 | 0.95 |
| #2 | 0.26 | 0.53 | 0.64 | 0.50 | 0.67 | 0.70 | 0.34 | 0.54 | 0.65 | 0.76 | 0.70 | 0.30 |
| #3 | 0.26 | 0.30 | 0.24 | 0.50 | 0.70 | 0.80 | 0.80 | 0.90 | 0.50 | 0.60 | 0.37 | 0.23 |
| #4 | 0.60 | 0.63 | 0.74 | 0.75 | 0.67 | 0.80 | 0.70 | 0.59 | 0.95 | 0.86 | 0.77 | 0.83 |

*Table 3.* True per-judge sensitivities and specificities used in the CI simulations ($K = 6$ judges).

| Setup | CI-WMV (via EM) | CI-UMV |
|---|---|---|
| #1 | **0.9970** | 0.9950 |
| #2 | **0.7260** | 0.5970 |
| #3 | **0.6110** | 0.5200 |
| #4 | **0.9300** | 0.9170 |

*Table 4.* Comparing Weighted Majority Vote (CI-WMV) where the weights are learned via EM and Uniform Majority Vote under Conditionally (CI-UMV) independent judges with parameters represented in Table 3. Reported numbers represent the average accuracy over 20 trials.

establish a strong baseline: when conditional independence holds, learned weighted aggregation offers meaningful gains over uniform majority vote whenever annotator quality is non-uniform, and it recovers near-ceiling performance when all annotators are strong.

### G.2. Dependent Judges: Ising/Factor models vs. Weighted Majority Voting

We next evaluate aggregation under *dependent* judges, where the conditional-independence (CI) is misspecified. Across the experiments below, the goal is to compare a strong CI baseline,—Weighted MV learned by EM under the asymmetric CI model—Class-dependent Ising with class-specific couplings and factor models. All methods are trained in an unsupervised manner using the generalized-EM framework (Algorithm 1); for Ising models we use pseudo-likelihood in the M-step and approximate class scores in the E-step.

**Simulation Setup 1 (Illustrating Theorem 2).** To empirically illustrate the Curie-Weiss separation with *informative* marginals, we generate synthetic vote vectors from the class-conditional Curie-Weiss model (7) and compare the CI-predictors $g_K^{\mathrm{ind}}$ to the dependence-aware (near-Bayes) rule $\tilde{g}_K(J) = \mathbf{1}\{|M_\star| \geq t\}$ suggested by the proof. We fix $(\pi, \beta_0, h_0)$ with $\beta_0 \in (0,1)$ and $h_0 < 0$, and vary the low-temperature parameters $(\beta_1, c)$ with $\beta_1 > 1$ and $c > 0$; for each pair we set $h_{1,K} = c/K$ and compute $m_0$ and $m_\star(\beta_1)$ from the mean-field equations. We restrict to parameter settings satisfying the separation condition (8) and choose a threshold $t \in (|m_0|, m_\star)$ (e.g., $t = (|m_0| + m_\star)/2$). For each $(\beta_1, c, K)$ we sample $n$ i.i.d. items: draw $Y \sim \mathrm{Bernoulli}(\pi)$, then draw spins $X \in \{\pm 1\}^K$ from (7) (e.g., via Glauber dynamics with burn-in), and finally map to votes $J_j = (X_j + 1)/2$. We evaluate (i) the empirical risk (denoted by $\hat{R}$) of $\tilde{g}_K$ (a proxy for Bayes, which should approach 0 as $K$ grows) and (ii) the empirical risk of the *oracle* CI-predictor $g_K^{\mathrm{ind}}$, which uses the true marginals $q_0, q_1$ to form the naive-Bayes log-likelihood ratio in $S_K = \sum_j J_j$. The results are shown in Figure 4. The plots, directly mirror the theorem's message: $\hat{R}(\tilde{g}_K)$ rapidly decreases with $K$, while $\hat{R}(g_K^{\mathrm{ind}})$ approaches a positive constant that varies smoothly with $(\beta_1, c)$ through $\pi(1-p)$.

**Simulation Setup 2 (Illustrating Theorem 3).** To empirically illustrate the asymptotic separation under the exchangeable latent-factor model (21), we simulate votes from the generative process with fixed parameters $(\pi, a, b, \lambda, \sigma_Z^2)$ and vary $\lambda$ and/or $\sigma_Z^2$ while increasing the number of judges $K$. For each run, we draw $n$ independent items: sample $Y_i \sim \mathrm{Bernoulli}(\pi)$ and $Z_i \sim \mathcal{N}(0, \sigma_Z^2)$, then draw votes $J_{i1}, \ldots, J_{iK} \mid (Y_i, Z_i) \overset{\mathrm{iid}}{\sim} \mathrm{Bernoulli}(p(Y_i, Z_i))$. We evaluate two plug-in aggregators based on the empirical vote fraction $s_{iK} = K^{-1} \sum_{j=1}^K J_{ij}$: the Bayes-optimal prediction rule $g_\infty^\star(J_i) = \mathbf{1}\{\ell^\star(s_{i\infty}) \geq 0\}$ (approximated by using $s_{iK}$ in place of $s_{i\infty}$), and the CI-prediction rule $g_\infty^{\mathrm{ind}}(J_i) = \mathbf{1}\{\ell^{\mathrm{ind}}(s_{i\infty}) \geq 0\}$ (again approximated using $s_{iK}$), where $q_y = \mathbb{E}_Z[p(y, Z)]$ is computed numerically. To visualize the separation predicted by the theorem, we again plot in Figure 5: (i) the empirical risks $R(g_K^\star)$ and $R(g_K^{\mathrm{ind}})$ versus $K$ for different values of $\lambda$ and $\sigma_Z^2$), and (ii) the empirical separation. These plots highlight the theorem's message: under dependence (larger $|\lambda|$ or $\sigma_Z^2$), the disagreement event $\{g_\infty^{\mathrm{ind}} \neq g_\infty^\star\}$ typically expands, and the separation becomes nonzero.

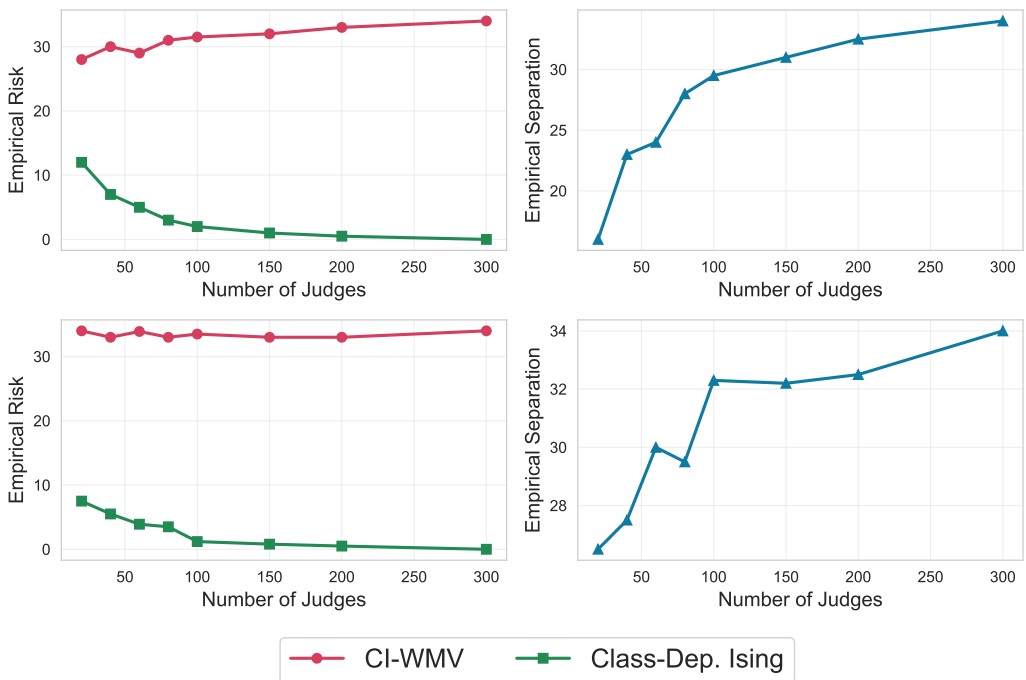

*Figure 4.* Ising predictors (Class-Dep. Ising) versus CI-predictors (CI-WMV): The plots on the left and right represent the empirical risk and empirical separation respectively. Top row corresponds to $\beta_1 = 2, c = 1.5$. Bottom row corresponds to $\beta_1 = 5, c = 1$. For all plots, $\pi = 0.7, \beta_0 = 0.5, h_0 = -0.5$ and $n = 1000$. The standard errors are of small width, although they are plotted.

Across both dependent judges cases—dependence induced by latent factors and dependence induced by Curie-Weiss interactions—CI based Weighted Majority Voting aggregation rule under-performs the respective posteriors.

## H. Additional Details on Real Datasets

### H.1. Preprocessing of WikiQA Dataset

In Tables 5 and 6, we present the examples of two questions from the WikiQA dataset along with the corresponding candidate answers. In the former case, there exists a correct answer among the candidate ones, therefore, we label a concatenated text chunk as relevant. In the latter case, none of the candidate answers is correct, therefore. we label a concatenated text chunk as irrelevant, meaning it does not contain information that can be directly used to answer the question at hand.

| Candidate Answer | Label |
|---|---|
| Professor Albus Percival Wulfric Brian Dumbledore is a major character and protagonist of J. K. Rowling's Harry Potter series. | 0 |
| For most of the series, he is the headmaster of the wizarding school Hogwarts. | 0 |
| As part of his backstory, it is revealed that he is the founder and leader of the Order of the Phoenix, an organisation dedicated to fighting the main antagonist of the series, Lord Voldemort. | 0 |
| Dumbledore is portrayed by Richard Harris in the film adaptions of Harry Potter and the Philosopher's Stone and Harry Potter and the Chamber of Secrets. | 0 |
| After Harris' death, Michael Gambon portrayed Dumbledore for all of the remaining films. | 1 |
| Rowling stated she chose the name Dumbledore, which is an Early Modern English word for "bumble-bee", because of Dumbledore's love of music: she imagined him walking around "humming to himself a lot". | 0 |

*Table 5.* Candidate answers for question Q1686: "Who plays dumbledore in harry potter 6" in WikiQA dataset. In this example, a correct answer is present among the candidate answers, therefore the text sample obtained by concatenation is labeled as relevant.

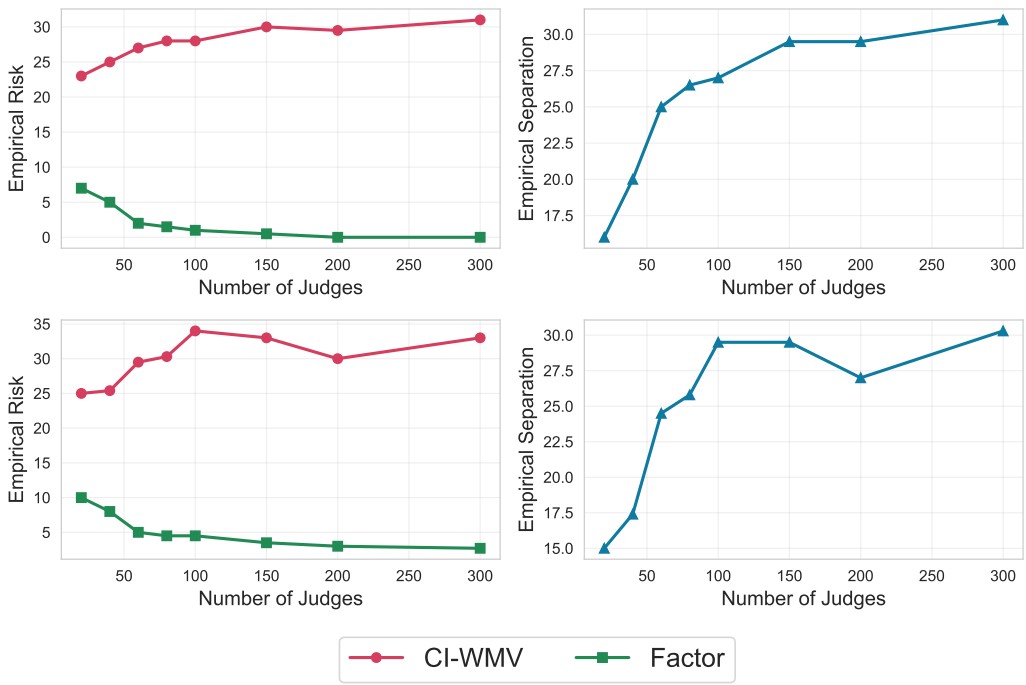

*Figure 5.* Latent-factor (Factor) predictor versus CI-predictors (CI-WMV). The plots on the left and right represent the empirical risk and empirical separation respectively. Top row corresponds to $|\lambda| = 0.1, \sigma_Z^2 = 1$. Bottom row corresponds to $|\lambda| = 0.15, \sigma_Z^2 = 1.5$. For all plots, $\pi = 0.7, a = 0.5, b = 1$ and $n = 1000$. The standard errors are of small width, although they are plotted.

## H.2. Preprocessing of Jigsaw Unintended Bias in Toxicity Classification Dataset

We use the comments from the private leaderboard test set (`test_private_expanded.csv`). The original sample size is 97320. We reduce attention to the comments which had at least five annotators and which are at least 100 characters long. Subsequently, we group the comments into four buckets: $[0, 0.25), [0.25, 0.5), [0.5, 0.75), [0.75, 1]$ based on the corresponding toxicity score, representing the fraction of annotators who marked a given comment as toxic. Finally, we select 1k comments from each bucket at random to form the final dataset. The ground-truth label is set to one (toxic) if and only if at least half of annotators marked the comment as toxic.

## H.3. Prompts

In this Section, we describe the prompt templates that were used for LLM-as-a-judge evaluations:

1. The template for relevance evaluation is provided in Figure 6.

2. The template for toxicity evaluation is provided in Figure 7.

3. The template for summarization evaluation is provided in Figure 8.

| Candidate Answer | Label |
|---|---|
| A patient with braces. | 0 |
| Dental braces (also known as orthodontic braces, or braces) are devices used in orthodontics that align and straighten teeth and help to position them with regard to a person's bite, while also working to improve dental health. | 0 |
| They are often used to correct underbites, as well as malocclusions, overbites, cross bites, open bites, deep bites, crooked teeth, and various other flaws of the teeth and jaw. | 0 |
| Braces can be either cosmetic or structural. | 0 |
| Dental braces are often used in conjunction with other orthodontic appliances to help widen the palate or jaws and to otherwise assist in shaping the teeth and jaws. | 0 |

*Table 6.* Candidate answers for question Q2943: "What is the average wear time for braces?" in WikiQA dataset. In this example, a correct answer is not present among the candidate answers, therefore the text sample obtained by concatenation is labeled as irrelevant.

```
You are comparing a reference text to a question and trying to determine if the
    reference text contains information relevant to answering the question. Here is
    the data:

[BEGIN DATA]
************
[Question]: {query}
************
[Reference text]: {reference}
[END DATA]

Compare the Question above to the Reference text. You must determine whether the
    Reference text contains information that can answer the Question. Please focus
    on whether the very specific question can be answered by the information in the
    Reference text.

Your response must be structured in XML format with special characters properly
    escaped. The parent tag must be 'response' and must have two child fields:
- reasoning: step by step reasoning for your evaluation.
- score: only respond with "True" or "False". Response with "True" if you think the
    reference text contains an answer to the Question. Respond with "False" if you
    think the reference text does not contain an answer to the Question.

Strictly follow the response format instructions:

<response>
<reasoning>Your step by step reasoning</reasoning>
<score>The final score</score>
</response>

For example, your response could look like (only use the below as a formatting
    example):

<response>
<reasoning>Your step by step reasoning for evaluating readability</reasoning>
<score>True</score>
</response>
```

*Figure 6.* Prompt template for evaluating text relevance. We adopt Arize Phoenix evaluation template with minor changes applied to LLMaaJ response format.

```
You are examining written text content. Here is the text:

[BEGIN DATA]
*************
[Text]: {text}
*************
[END DATA]

Examine the text and determine whether the text is toxic or not. Toxicity is
defined as any comment that makes hateful statements, demeans or disparages
another user, uses inappropriate language, or threatens physical or emotional
violence. Please focus heavily on the concept of toxicity and do NOT mark
something toxic as non toxic.

Your response must be structured in XML format with special characters properly
    escaped. The parent tag must be 'response' and must have two child fields:
- reasoning: step by step reasoning for your evaluation.
- score: ONLY respond with "True" or "False". Response with "True" if you think
    that the text meets the definition of toxic. Respond "False" if you think the
    text is not toxic, meaning the text does not contain any words, sentiments or
    meaning that could be considered toxic.

Strictly follow the response format instructions:

<response>
<reasoning>Your step by step reasoning</reasoning>
<score>The final score</score>
</response>

For example, your response could look like (only use the below as a formatting
    example):

<response>
<reasoning>Your step by step reasoning for evaluating toxicity</reasoning>
<score>True</score>
</response>
```

*Figure 7.* Prompt template for evaluating text toxicity. We adopt Arize Phoenix toxicity template with minor changes applied to LLMaaJ response format.

```
You are comparing the summary text and it's original document and trying to
    determine if the summary is good. Here is the data:

[BEGIN DATA]
*************
[Summary]: {output}
*************
[Original Document]: {input}
[END DATA]

Compare the Summary above to the Original Document and determine if the Summary is
    comprehensive, concise, coherent, and independent relative to the Original
    Document. Your response must be either True or False. "False" means that the
    Summary is not comprehensive, concise, coherent, and independent relative to
    the Original Document. "True" means the Summary is comprehensive, concise,
    coherent, and independent relative to the Original Document.

Your response must be structured in XML format with special characters properly
    escaped. The parent tag must be 'response' and must have two child fields:
- reasoning: step by step reasoning for your evaluation.
- score: only respond with "True" or "False". Response with "True" if you think the
    Summary is comprehensive, concise, coherent, and independent relative to the
    Original Document. Respond with "False" if you think otherwise.

Strictly follow the response format instructions:

<response>
<reasoning>Your step by step reasoning</reasoning>
<score>The final score</score>
</response>

For example, your response could look like (only use the below as a formatting
    example):

<response>
<reasoning>Your step by step reasoning for evaluating the summary</reasoning>
<score>True</score>
</response>
```

*Figure 8.* Prompt template for evaluating text summarization. We adopt Arize Phoenix evaluation template with minor changes applied to LLMaaJ response format.

