# OpenReview forum: "Dependence-Aware Label Aggregation for LLM-as-a-Judge via Ising Models"
_ICML.cc/2026/Conference — ICML 2026 regular_

### Official Review · Reviewer_iSqC · 2026-02-18

**Soundness:** 3
**Presentation:** 2
**Significance:** 3
**Originality:** 3
**Overall Recommendation:** 4
**Confidence:** 2

**Summary:**

This paper discusses aggregating labels from LLM judges, where traditional methods like majority vote and Dawid-Skene assume independence between annotators, but LLMs often share training data, architectures, and prompts, leading to correlations that can make models overconfident in wrong answers. The authors explore class-independent and class-dependent Ising models using quadratic Markov random fields, with parameters h, local fields for judge biases, and W, couplings for pairwise dependencies, learned unsupervised via EM with pseudo-likelihood.

**Compliance With Llm Reviewing Policy:**

Affirmed.

**Final Justification:**

The main contribution to me is the theoretical analysis, which looks systematic, with a heavy appendix for proof. The main idea is to break the standard conditional independence (CI) assumption among the judges. The separation results demonstrate that CI-predictors can remain suboptimal even as the number of judges grows unboundedly, and the connection to latent-factor models is a non-obvious result that adds depth to the framework. Empirical results on three real-world tasks with six contemporary LLMs provide supporting evidence for the practical value of dependence modeling.

The rebuttal solved my concerns about the complexity of building the judge's dependence.

In summary, I did not find a clear mistake, and I will keep the weak acceptance decision.

**Key Questions For Authors:**

Since I'm not deep in this field, I'm just wondering about the performance of the majority vote.

It feels like there's room for bridging the theory-experiment divide with extra ablations.

**Limitations:**

The Ising model's complexity explodes exponentially, 2^K for exact partition functions, when we build the connection to each judge. It is a big issue for scaling to more judges. For larger K, it will be tough, and the paper doesn't dive deep into that.

There's a noticeable gap between the heavy theoretical proposals and the experiments. It'd be great to see more ablation studies on how the learned h and W actually correspond to real dependencies or affect results, like sensitivity analyses or visualizations of the coupling matrices.

**Strengths And Weaknesses:**

The core idea of modeling annotator dependencies with Ising graphs is a nice step forward. Previous work mostly sticks to independence assumptions, so tackling correlated judges feels timely and relevant.
They back it up with theory, like separation results showing how CI methods can fail big-time even with lots of judges, while their approach can drive error close to zero.

Experiments on three real datasets using actual LLMs like Claude and Llama show improved accuracy over the independent baselines, which gives some practical cred.
The hierarchy, CI \subset class-independent \subset class-dependent, is laid out, and linking to latent factor models in the appendix adds depth.

---

> ### Author Rebuttal · Authors · 2026-03-30
>
> We thank the reviewer for their feedback. Below, we address the points raised.
>
> >...wondering about performance of majority vote...there's room for bridging the theory-experiment divide with extra ablations.
>
> Please see reply to the reviewer yRxJ for extra ablations we ran in revision.
>
> >Ising model complexity explodes $2^K$...issue for scaling to more judges
>
> The pseudo-likelihood M-step fits $K$ independent logistic regressions with $K$ features and $n$ weighted samples, costing $O(K^2 n)$ per iteration. Parameters include $K$ bias fields + $\binom{K}{2}$ couplings:
>
> |$K$|Parameters & Approx.|M-step cost|
> |---|---|---|
> 6|6+15=21|~sec|
> 10|10+45=55|~sec|
> 20|20+190=210|~sec|
> 50|50+1225=1275|~min|
> 100|100+4950=5050|~mins-hours|
>
> Computationally, pseudo-likelihood scales comfortably to $K = 50+$; The bottleneck is never the runtime.
>
> The real concern is *statistical*: with $K$ judges we have $O(K^2)$ coupling parameters to estimate from $n$ items (doubled for class-dependent Ising). For reliable estimation, we need $n \gg K^2$. For paper's datasets ($n\sim 1000$), $K = 6$ gives a ratio of ~$1000/21 \approx 50$ samples per parameter. At $K = 20$, the ratio drops to ~5, which is thin. At $K = 50$ we would have more parameters than samples.
>
> This is where the model hierarchy becomes important:
>
> -**CI (Dawid--Skene):** $2K$ parameters, scales to any $K$.
>
> -**Class-independent Ising:** $K + \binom{K}{2}$ parameters, needs $n \gg K^2$.
>
> -**Class-dependent Ising:** $2K + K(K-1)$ parameters, needs $n \gg 2K^2$.
>
> For large $K$, one would use stronger regularization on $W$ (pushing the model toward CI), structured sparsity (e.g., within-family couplings only), or a low-rank factor model (discussed in the paper) where dependence is mediated through a small number of latent factors rather than $\binom{K}{2}$ pairwise couplings.
>
> The exact-likelihood E-step (enumerating $2^K$ configurations) is a component that truly does not scale beyond $K \approx 20$. However, as observed, pseudo-likelihood is empirically equivalent at small $K$ and remains $O(K \cdot n)$ regardless of $K$.
>
> >...gap between the heavy theoretical proposals and the experiments. It'd be great to see more ablation studies on how the learned $h$ and $W$ actually correspond to real dependencies or affect results...
>
> We have conducted additional ablation studies to bridge the gap between theory and empirics (we have also increased the number of judges to 10).
>
> **1.The learned $W$ recovers dependence structure.** The coupling matrix $W$ correlates strongly with the empirical residual correlation matrix (Pearson correlation between judges after conditioning on the true label). The rank-order correlation between $W$ entries and residual correlations is 0.505 (relevance), 0.704 (toxicity), and 0.891 (summarization), confirming that pseudo-likelihood estimation recovers meaningful pairwise dependence without observing true labels.
>
> **2.$W$ captures model-family structure.** Across the tasks, within-family judge pairs (e.g., Sonnet/Opus pair from Anthropic) have consistently stronger coupling than cross-family pairs:
>
> |Dataset|Within-family avg $W$|Across-family avg $W$|Ratio|
> |---|---|---|---|
> Relevance|0.479|0.375|1.28x|
> Toxicity|0.503|0.296|1.70x|
> Summarization|0.232|0.111| 2.09x|
>
> The strongest coupling in every dataset is the Sonnet-Opus pair ($W=$ 0.568, 0.692, 0.415 respectively), confirming that models from the same provider share the most failure modes. The weakest couplings consistently involve cross-family pairs with DeepSeek-R1.
>
> **3.The bias field $h$ encodes per-judge discriminative power.**
> The difference $\Delta h = h^{(1)} - h^{(0)}$ reflects judge's ability to distinguish classes. On Summarization, where judges vary most in quality, $\Delta h$ correlates with the log-odds of individual accuracy at $r = 0.557$. Opus has the largest $\Delta h = 3.00$, matching its role as the most informative judge after Sonnet ($\Delta h = 2.92$). DeepSeek-R1 has $\Delta h = 2.61$ despite near-zero sensitivity (0.014), because its extreme specificity (0.998) makes its rare positive votes highly informative (a nuance $\Delta h$ captures but raw accuracy does not).
>
> **4.Class-dependent coupling reveals asymmetric failure modes.** The norms of $W^{(0)}$ and $W^{(1)}$ differ dramatically on Summarization: $\|W^{(0)}\|_F = 0.639$ vs. $\|W^{(1)}\|_F = 2.400$. Judges are nearly independent when on *bad summaries* ($Y=0$) but strongly coupled when on *good* summaries ($Y=1$). Unanimous agreement on *good* thus carries less information than CI model assumes: the judges are right for correlated reasons, and not independently. The CD-Ising model discounts such redundant agreement under $Y=1$ while trusting independent assessments under $Y=0$. On Toxicity, the asymmetry reverses: $\|W^{(0)}\|_F = 3.516 > \|W^{(1)}\|_F = 3.215$, indicating stronger coupling when items are non-toxic. We have added a heat plot to visualize the $W$ matrix in the revision.

---

> > ### Author Rebuttal · Reviewer_iSqC · 2026-04-02
> >
> > Thanks for answering. I will keep the weak acceptance decision.

---

### Official Review · Reviewer_yRxJ · 2026-03-12

**Soundness:** 3
**Presentation:** 3
**Significance:** 3
**Originality:** 3
**Overall Recommendation:** 4
**Confidence:** 3

**Summary:**

This paper addresses the problem of aggregating binary judgments from multiple LLM judges. The central and well-motivated observation is that LLM judges are not independent.
The authors model judge dependencies via Ising graphical models, where the coupling matrix W captures pairwise interactions between judges. They develop a clean hierarchy of models and derive the Bayes optimal predictor for each. The theoretical contribution is: the Bayes optimal risk vanishes as the number of judges grows, while the CI predictor risk stays bounded away from zero even with infinitely many judges. Limitations to this proof analyzed.

**Compliance With Llm Reviewing Policy:**

Affirmed.

**Key Questions For Authors:**

Pseudo-likelihood estimators are consistent but their finite sample distributions are not well characterized, so any hypothesis test on W would have unclear power guarantees. Can we see a short experimets on hte quality or references supporting it?

The results are competent but conclusions are underwhelming. The natural and more interesting question is never asked: do the estimated couplings reveal something about these specific models? Discuss on how can this be used to anser this question:
Are Claude models more coupled to each other than to DeepSeek or Llama? Does coupling strength vary by task?
Is this a good base for designing hypothesis tests of system dependency?

**Limitations:**

yes

**Strengths And Weaknesses:**

Strengths:
The non-independence assumption is highly reasonable and well motivated for modern LLM evaluation pipelines. The theoretical framework is sound, self-contained, and the separation results are genuinely interesting.

Weaknesses:
Presentation: It is unusual and somewhat disorienting that virtually all references are deferred to Appendix A.
Experiments, synthetic: The synthetic experiments are disappointingly thin. Given that the theoretical results are asymptotic in K, it would have been very valuable to see systematic experiments varying K. Exac compuation would have been possible as well here.
Experiments, exact computation: A direct comparison of exact vs pseudo-likelihood inference would have been both easy to run and highly informative.
Experiments, real data: The real data results are competent but not thrilling. The accuracy improvements are consistent but modest, and no particularly interesting or surprising conclusions are drawn from them. Notably the estimated coupling matrices W are never reported or analyzed.
Missing applications: The paper would have been considerably strengthened by concrete applications that showcase what you can do with the estimated dependence structure beyond just improving accuracy. The most natural and exciting application is detecting dependencies among AI systems, and its never explored. The estimated W matrix could directly answer questions like: which LLMs share failure modes? Are models from the same family more correlated? Does dependence vary by task type? This would have transformed the paper from a methodological contribution into something with broader practical impact and would have made the non-independence assumption feel empirically grounded rather than just theoretically motivated.

---

> ### Author Rebuttal · Authors · 2026-03-30
>
> We thank the reviewer for their feedback. Below, we address the points raised.
> >...references are deferred to Appendix A
>
> We will reorganize and include related works in main paper.
> >...synthetic experiments are thin...theoretical results are asymptotic in K...systematic experiments varying K
>
> In Appendix F, we show synthetic results varying number of judges (~hundreds). For real data, we increased $K$ to ten (see updated results in response to Reviewer XH4L).
> > Comparison of exact vs pseudo-likelihood would be informative
>
> We tested both approaches using oracle labels in the M-step. In updated setup ($K=10$), both exact log-likelihood and pseudo-log-likelihood E-steps produced 0.697 accuracy on toxicity. On summarization, exact gave 0.764 vs pseudo 0.763. The difference is negligible.
> > Pseudo-likelihood estimators are consistent but finite sample distributions are not well characterized...any test on W would have unclear power guarantees...
>
> Our current discussion should not be interpreted as claiming finite-sample power guarantees for tests on individual $W$ entries. The intent was to suggest a diagnostic for assessing whether CI is implausible. In the fixed-$K$, growing-$n$ regime, pseudo-likelihood falls under standard composite-likelihood theory providing Godambe-based asymptotic normality, Wald/score tests, adjusted likelihood-ratio procedures, and specification tests with local power (Varin et al., 2011, Pace et al., 2011, Huang et al., 2022). Extending guarantees through latent-label EM layer is nontrivial and beyond scope. We revised the text to frame pseudo-likelihood primarily as a practical diagnostic rather than as a fully developed testing procedure.
> > Real data...results are competent, conclusions are underwhelming...do estimated couplings reveal something about specific models?..beyond improving accuracy
>
> We thank the reviewer for a suggestion to *interpret* $W$ matrix. Estimated couplings contain rich diagnostic information. In revision, we increased number of judges: $K=10$ (adding Opus 4.5, DeepSeek V3.2, OSS 20b, and Llama 3.3). We address each sub-question using updated experiments (similar conclusions hold for $K=6$).
>
> **Are Claude models more coupled to each other than to DeepSeek or Llama?** Yes. Sonnet-Opus coupling is the strongest pair in all datasets ($W = 0.568$ on Relevance, $0.692$ on Toxicity, $0.415$ on Summarization). Cross-family couplings are substantially weaker: $W = 0.478, 0.283, 0.026$ on these tasks for strongest Anthropic-DeepSeek coupling, and the strongest Anthropic-Meta coupling is weaker in two of three tasks. The full within- vs. cross-family comparison is:
>
> |Dataset|Anthropic-Anthropic|Meta-Meta|DeepSeek-DeepSeek|Cross-family|
> |---|---|---|---|---|
> |Relevance|0.400|0.397|0.475|0.375|
> |Toxicity|0.441|0.488|0.572|0.296|
> |Summarization|0.316|0.152|0.098|0.111|
>
> The pattern is consistent: same-provider models share more residual dependence than cross-provider (likely reflecting shared training data, RLHF procedures, etc. for each family).
>
> **Does coupling strength vary by task?** Dramatically. On relevance, couplings are uniformly high ($W \in [0.27, 0.57]$) with a modest within/across-family gap (1.28x). On toxicity, the gap widens (1.7x). On summarization, couplings are sparse and heterogeneous ($W \in [0.005, 0.415]$) with the largest within/across gap (2.09x), reflecting subjective criteria where provider-specific biases create family-specific failure modes.
>
> Class-dependent structure also varies: on summarization, $\|W^{(1)}\|_F / \|W^{(0)}\|_F = 3.75$ (judges nearly independent on bad summaries but strongly coupled on good ones); on toxicity, the ratio inverts to $0.91$. The asymmetries are task-specific and invisible to any CI model.
>
> **Is this a good basis for designing hypothesis tests of system dependency?** The Ising framework provides a natural foundation. The null $H_0: W_{jk} = 0$ (CI between judges $j$ and $k$) can be tested via pseudo-likelihood ratio or Wald test on coupling coefficients (see *Property testing in high-dimensional Ising models.*, Neykov et al, 2019). More structured hypotheses are also testable:
>
> - **Family independence**: $H_0: W_{jk} = 0$ for all cross-family pairs $(j,k)$, tested against the alternative that at least one cross-family coupling is nonzero.
> - **Exchangeability within family**: $H_0: W_{jk} = W_{j'k'}$ for all within-family pairs (are models from the same provider interchangeable?)
> - **Class-independent coupling**: $H_0: W^{(0)} = W^{(1)}$, i.e., does the dependence structure differ between classes.
>
> Each $W_{jk}$ is estimated via logistic regression, hence standard asymptotic inference (Wald statistics, likelihood-ratio tests) applies directly; bootstrap provides finite-sample alternatives. The Ising model not only improves aggregation but provides a principled framework for *auditing* independence assumptions in LLM-as-a-judge pipelines.
>
> These interpretations and heat plot of $W$ matrix are added in revision.

---

> > ### Author Rebuttal · Reviewer_yRxJ · 2026-04-05
> >
> > Thank you for the answers, I think that my previous score is appropriate.

---

> > > ### Author Response · Authors · 2026-04-05
> > >
> > > Dear Reviewer yRxJ,
> > >
> > > Thanks for your update. We are bit concerned as you mention "Partially resolved - I have follow-up questions for the authors" but do not actually provide any follow-up questions. Please do let us know your questions and we will be happy to clarify as appropriate.
> > >
> > > best\
> > > Authors

---

### Official Review · Reviewer_XH4L · 2026-03-13

**Soundness:** 2
**Presentation:** 2
**Significance:** 3
**Originality:** 3
**Overall Recommendation:** 3
**Confidence:** 3

**Summary:**

This paper addresses a fundamental problem in LLM-as-a-judge evaluation: standard label aggregation methods (majority vote, weighted majority vote, Dawid-Skene) assume conditional independence (CI) among judges given the true label, but LLM judges share pretraining corpora, architectures, prompt templates, and failure modes, systematically violating this assumption. The paper proposes modeling judge dependence through Ising graphical models quadratic Markov random fields over binary votes. Theorem 1 shows that under the Curie-Weiss model with identical marginals, the CI-predictor collapses to the prior while the Bayes-optimal risk vanishes as K → ∞; Theorem 2 extends this to informative marginals, showing CI risk remains bounded away from zero while Bayes risk vanishes. Parameters are estimated via EM. Experiments on three tasks with six LLM judges show class-dependent Ising achieves the best accuracy.

**Compliance With Llm Reviewing Policy:**

Affirmed.

**Final Justification:**

I thank the authors for the additional analyses, which directly address my follow-up questions. However, I still feel the overall writing quality and presentation could be improved, i.e several sections are dense and would benefit from clearer exposition. I raise my score to 3.

**Key Questions For Authors:**

1. Please report the variance of test accuracy across 5+ random EM initializations for the class-dependent Ising model on each dataset.
2. How does the method scale to K > 6 judges?
3. What is the computational cost of EM estimation relative to simply running the LLM judges?

**Limitations:**

Yes

**Strengths And Weaknesses:**

Strengths:
1. The separation results are the paper's strongest contribution.
2. The nested structure provides a natural escalation path for practitioners: start with CI (cheapest), test for residual correlations, then move up the hierarchy as needed.
3. LLM-as-a-judge is now ubiquitous in evaluation pipelines, and the independence assumption is clearly violated in practice (shared architectures, pretraining data, prompt templates, correlated hallucination/refusal patterns).
4. Using six diverse LLM judges across three distinct tasks provides reasonable coverage. The experiments studying both sample size n and number of judges K are informative.

Weaknesses:
1. The separation theorems rely on the Curie-Weiss specialization, which is strong and may not hold for real LLM judges. The Curie-Weiss model assumes uniform pairwise coupling. This implies all judges are equally correlated with each other. In practice, LLM judges have heterogeneous correlations Claude models correlate more with each other than with Llama models, for instance.
2. Only 6 judges are used. The theory's power grows with K → ∞, but K=6 is far from asymptotic. The Figure 3 right panels show gains, but the trends have not plateaued.
3. The EM estimation procedure is underspecified and its reliability is not established. The paper defers EM details to Appendix D, but several important questions arise: what identifiability conditions are required? The class-dependent Ising has O(K^2) coupling parameters per class plus 2K fields. How many labeled or unlabeled samples are needed for reliable estimation?
4. The gap between theory and practice is not fully bridged. The separation theorems are stated for the population Bayes-optimal predictor under the true generative model. In practice, the model is estimated from finite samples via EM, and the data may not follow an Ising model.

---

> ### Author Rebuttal · Authors · 2026-03-30
>
> We thank the reviewer for their feedback. Below, we address the points raised.
> > ...separation theorems rely on the Curie-Weiss specialization...
>
> Please see the response to Reviewer 3EqD (W1...) where we discuss relaxation.
> > Only 6 judges used...theory is for $K\to\infty$ but $K=6$ is far from asymptotic...Figure 3 (right) shows gains, but trends have not plateaued.
>
> We have updated experiment to $K=10$ judges (adding Opus 4.5, DeepSeek V3.2, OSS 20b, and Llama 3.3 to the panel). Test accuracy on updated datasets ($K=10$ judges, 75/25 train/test split, averaged over 20 trials):
>
> |Dataset|Class-Dep. Ising|Class-Indep. Ising|CI-WMV|CI-UMV|
> |---|---|---|---|---|
> |Relevance|**0.912$\pm$0.007**|0.899$\pm$0.005|0.820$\pm$0.004|0.804$\pm$0.003|
> |Toxicity|**0.792$\pm$0.004**|0.780$\pm$0.006|0.694$\pm$0.004|0.695$\pm$0.003|
> |Summarization|**0.806$\pm$0.004**|**0.800$\pm$ 0.005**|0.737$\pm$0.005|0.561$\pm$ 0.005|
>
> We agree that $K=10$ or $K=6$ are far from asymptotic; our experiments target realistic small-$K$ settings. Separation arises from concentration of low-dimensional sufficient statistics, which already appears at moderate $K$. As noted to Reviewer 3EqD, the same argument yields non-asymptotic guarantees (e.g., exponential decay of dependence-aware risk), so meaningful separation occurs well before the asymptotic regime. We have clarified that in revision.
>
> Regarding Figure 3 right panel, sweeping $K=3-10$ yields the following test accuracies on summarization: 0.620, 0.659, 0.713, 0.749, 0.7738, 0.790, 0.796, 0.806, so the plateau is visible now on all datasets.
>
> > Gap between theory and practice is not fully bridged...separation stated for population Bayes-optimal predictor under true generative model...in practice, model is estimated from finite samples via EM...data may not follow Ising model
>
> We thank the reviewer for highlighting this gap. Our separation results are population-level under correct specification; our goal is to isolate the phenomenon that dependence can be informative even with identical marginals, not to claim exact optimality. Bridging the gap relies on two points: under standard conditions, pseudo-likelihood estimators are consistent in fixed-$K$, large-$n$ regime and EM approximates maximum likelihood, so the learned predictor approaches population one as data grow; and finite-$K$ separation is driven by concentration of low-dimensional sufficient statistics, so moderate estimation error only slightly perturbs the decision boundary when the margin is non-degenerate. We revised the text to clarify that our guarantees are population-level and that experiments provide evidence the separation persists under estimation and moderate misspecification, leaving full finite-sample and robustness analysis as future work.
>
> > Report variance of test accuracy across 5+ random EM initializations...
>
> Standard errors are small and not visible in Figures 3--5 (noted in the captions). We report standard deviations in the table above for the updated experiment ($K=10$).
>
> > EM procedure is underspecified...identifiability conditions required? Class-dependent Ising has $O(K^2)$ coupling parameters per class plus $2K$ fields. How many labeled/unlabeled samples are needed for reliable estimation?...How does method scale to $K>6$ judges?
>
> Please see the response to Reviewer iSqC (Ising model complexity explodes...) where we discuss scaling and sample size analysis.
>
> > What is the computational cost of EM estimation relative to simply running the LLM judges?
>
> The Ising EM takes at most ~2 seconds for the largest dataset (2000 items, 6 judges). That's the entire fitting procedure including all EM iterations, all pseudo-likelihood optimizations. For context, a single LLM judge call on one item takes roughly:
>
> - 1--5 seconds for a fast model (e.g., Anthropic Haiku) with simple prompts
> - 5--30 seconds for a reasoning model (e.g., DeepSeek-R1) or long inputs
>
> Evaluating 2k items with 6 judges means 12k LLM calls, which at ~2 seconds each is about 6--7 hours of sequential compute (or minutes with parallelism, but still thousands of GPU-seconds). The Ising EM adds 2 seconds of CPU time on top of that which is roughly 0.00001x the cost of the judge inference. Even Dawid-Skene at 5ms is essentially free, and the Ising model at 100-2000ms is still free in comparison.
>
> The per-item cost of the Ising aggregation is ~0.1--1 millisecond. An LLM judge call is ~2--30 seconds. That's 3--4 orders of magnitude difference. One could run the Ising EM a thousand times (for hyperparameter search, bootstrapping, etc.) and it would still be a rounding error on the LLM inference budget.
>
> In summary, the computational bottleneck is entirely in generating the votes, never in aggregating them. We clarified that point in revision.
>
> We hope that our response addresses your concerns, in which case we would greatly appreciate a score raise. Please follow up during the discussion phase if you have remaining questions.

---

> > ### Author Rebuttal · Reviewer_XH4L · 2026-04-03
> >
> > I thank the authors for their detailed rebuttal. The updated experiments adequately address my original questions about variance, scalability, and EM reliability. The expanded results with the plateau visible in the K-sweep are helpful. I still feel the experimental section could benefit from deeper analysis beyond reporting final accuracy numbers. For instance: ablations on the structure of W, analysis of when the quadratic term in class-dependent Ising actually changes predictions relative to the class-independent case. Sensitivity to regularization choices and calibration metrics would also strengthen the empirical narrative.

---

> > > ### Author Response · Authors · 2026-04-03
> > >
> > > Thanks for the suggestions. We report results on 3 datasets ($K=10$, 75/25 split).
> > >
> > > ### 1. When does the quadratic term change predictions?
> > >
> > > **CI-Ising vs. CD-Ising disagreement ($K=10$)**
> > >
> > > | Dataset        | Test items disagreed | CI correct | CD correct |
> > > |----------------|---------------------|------------|------------|
> > > | Relevance      | 92 / 762 (12.1%)    | 0.413      | 0.587      |
> > > | Toxicity       | 148 / 1000 (14.8%)  | 0.432      | 0.568      |
> > > | Summarization  | 28 / 275 (10.2%)    | 0.464      | 0.536      |
> > >
> > > Models disagree on ~10–15% of cases, limiting overall accuracy differences. On these harder cases, CD-Ising consistently outperforms CI-Ising (53–59% correct), indicating that the quadratic term improves decision boundaries selectively rather than globally. This shows that most predictions remain unchanged, while gains come from correcting difficult, ambiguous examples where dependencies matter most.
> > >
> > > ---
> > >
> > > ### 2. Structure of $W$
> > >
> > > **Spectral properties of $W$ and $\Delta W$**
> > >
> > > | Dataset        | $\|W\|_F$ | Top eigenvalue (% mass) | Entries $>0.1$ | $\|\Delta W\|_F$ | Ratio |
> > > |----------------|-----------|--------------------------|----------------|------------------|-------|
> > > | Relevance      | 2.607     | 2.456 (50.0%)            | 40/45          | 5.035            | 1.93  |
> > > | Toxicity       | 2.512     | 2.225 (41.7%)            | 39/45          | 4.541            | 1.81  |
> > > | Summarization  | 1.355     | 1.167 (40.7%)            | 23/45          | 2.633            | 1.94  |
> > >
> > > $W$ exhibits strong structure, with a dominant low-rank component (top eigenvalue captures 40–50%). This confirms that annotator dependencies are substantial and not noise. CI-WMV fails because it ignores these correlations. Moreover, large $\|\Delta W\|_F / \|W\|_F$ (~1.8–1.9) shows that interaction patterns differ significantly across classes. Thus, dependencies are both strong and label-dependent, motivating CD-Ising over CI-Ising.
> > >
> > > ---
> > >
> > > ### 3. Sensitivity to regularization
> > >
> > > **CI-Ising accuracy vs. regularization**
> > >
> > > | $\lambda_h$ | $\lambda_W$ | Relevance | Toxicity | Summarization |
> > > |------------|------------|-----------|----------|---------------|
> > > | 0.0001     | 0.01       | 0.481     | 0.502    | 0.710         |
> > > | 0.0001     | 0.05       | 0.681     | 0.686    | **0.806**     |
> > > | 0.0001     | 0.1        | 0.888     | 0.787    | 0.781         |
> > > | 0.0001     | 0.5        | **0.912** | **0.792**| 0.740         |
> > > | 0.0001     | 1.0        | 0.896     | 0.744    | 0.740         |
> > > | 0.01       | 0.05       | 0.857     | 0.711    | 0.737         |
> > > | 0.01       | 0.1        | 0.832     | 0.700    | 0.662         |
> > > | 0.1        | 0.5        | 0.820     | 0.694    | 0.561         |
> > >
> > > Optimal $\lambda_W$ depends on dataset structure. Summarization benefits from weaker regularization (0.05), indicating richer, exploitable dependencies. Relevance and Toxicity require stronger regularization ($\geq 0.1$), consistent with more uniform annotator behavior. At high $\lambda_W$, couplings vanish and performance approaches CI baselines, showing the model interpolates between independence and full dependence.
> > >
> > > ---
> > >
> > > ### 4. Calibration
> > >
> > > **Test calibration metrics**
> > >
> > > | Dataset       | Method    | Brier | ECE   | Log Loss | Accuracy |
> > > |---------------|-----------|-------|-------|----------|----------|
> > > | Relevance     | CI-WMV    | 0.380 | 0.355 | 8.149    | 0.820    |
> > > |               | CI-Ising  | 0.273 | 0.296 | 4.365    | 0.899    |
> > > |               | CD-Ising  | 0.222 | 0.223 | 2.014    | 0.912    |
> > > | Toxicity      | CI-WMV    | 0.478 | 0.277 | 9.240    | 0.694    |
> > > |               | CI-Ising  | 0.294 | 0.269 | 3.969    | 0.780    |
> > > |               | CD-Ising  | 0.272 | 0.156 | 2.449    | 0.792    |
> > > | Summarization | CI-WMV    | 0.238 | 0.247 | 1.344    | 0.737    |
> > > |               | CI-Ising  | 0.145 | 0.096 | 0.672    | 0.800|
> > > |               | CD-Ising  | 0.148 | 0.068 | 0.653    | 0.806|
> > >
> > > Ising-based methods outperform CI-WMV on all metrics. CD-Ising achieves the best overall performance, especially on Relevance and Toxicity, with lower Brier, ECE, and log loss, and higher accuracy. Improvements in log loss indicate substantially better probabilistic predictions. On Summarization, both Ising models achieve similar accuracy, but CD-Ising yields better calibration. Overall, modeling dependencies improves both predictive accuracy and confidence estimates.
> > >
> > > We sincerely appreciate updating your scores if we answered your concerns. Thank you!
> > >
> > > ------
> > > Update: Thanks for updating the score. We are curious if you have any other questions left for us. We would be happy to clarify as needed.

---

### Official Review · Reviewer_3EqD · 2026-03-14

**Soundness:** 4
**Presentation:** 4
**Significance:** 3
**Originality:** 3
**Overall Recommendation:** 5
**Confidence:** 4

**Summary:**

This paper studies label aggregation for LLM-as-a-judge through Ising models. It introduces a model hierarchy (CI, class-independent Ising, class-dependent Ising), derives Bayes-optimal aggregation rules, proves separation results showing CI-predictors incur nonvanishing excess risk under judge dependence even as $K\to\infty$, and validates the framework on three real-world datasets.

**Compliance With Llm Reviewing Policy:**

Affirmed.

**Final Justification:**

I thank the authors for their detailed response. I will keep the acceptance decision.

**Key Questions For Authors:**

See above

**Limitations:**

Yes.

**Strengths And Weaknesses:**

**Strengths**

- The motivation is well-grounded. The observation that LLM judges violate conditional independence is practically important and underappreciated in the evaluation community.

- The theoretical contributions are substantial and technically sound.

- The paper is well-written.


**Weaknesses and Questions:**

**W1. The separation results rely on the Curie-Weiss model, which may be unrealistic for LLM judges**

Theorems 1 and 2 are established under the Curie-Weiss model, which assumes uniform coupling strength across all annotator pairs, a strong symmetry assumption that may not hold in practice, where different LLM pairs can exhibit very different correlation patterns depending on their training data and architectures. Can the authors discuss whether analogous separation results can be obtained under more general Ising families with appropriate regularity conditions? I understand this would introduce significant technical difficulties and treat this as a discussion question rather than a requirement.

**W2. The asymptotic results may not be relevant at practical scales of $K$**

The key separation results (Theorems 1–3) are all stated in the $K\to\infty$ limit, whereas LLM evaluation panels typically involve very few judges ($K=6$ in the paper's own experiments). Can the authors provide finite-$K$ separation bounds, or characterize the rate at which the separation develops with $K$?

**W3. Extension to the multiclass setting**

It would be helpful to discuss whether the model hierarchy, Bayes log-odds derivations, and separation results extend to multiclass setting and what additional challenges arise.

**W4. Missing comparison with relevant crowdsourcing baselines**

The experiments compare only against CI-WMV and CI-UMV. Methods from the crowdsourcing literature, such as Xu et al. (2024), are directly relevant but absent from the empirical comparison. Without such comparisons, it is difficult to assess whether the reported gains represent a genuine advance over existing dependence-aware aggregation methods.

References:

Xu, Q., Yuan, Y., Wang, J., and Qu, A. Crowdsourcing utilizing subgroup structure of latent factor modeling. Journal of the American Statistical Association, 119(546): 1192–1204, 2024.

---

> ### Author Rebuttal · Authors · 2026-03-30
>
> We thank the reviewer for their feedback. Below, we address the points raised.
> > W1...separation results rely on Curie-Weiss model...unrealistic for LLM judges.
>
> Similar separation results extend to block-wise mean-field Ising models [BRS19], where judges are split into groups $G_1,\dots, G_B $ and $W_{jk}^{(y)}$= $B_{ab}^{(y)}/K$ whenever $j\in G_a$ and $k\in G_b$. Scalar magnetization is replaced by block magnetizations $M_{a, K}=|G_a|^{-1}\sum_{j\in G_a}(2J_j-1)$, and the Bayes score is a quadratic function of the low-dimensional order parameter. Under standard regularity (fixed $B$, bounded block proportions and couplings, etc.), vector $(M_{1,K},\dots,M_{B,K})$ concentrates exponentially near class-dependent limit points, and the same separation applies: dependence-aware predictors achieve vanishing risk, while CI predictors can still incur non-vanishing error when the classes differ through block correlation patterns rather than marginal rates. The key requirement is not full Curie-Weiss symmetry but concentration of low-dimensional parameters.
>
> **Please also see reply to Reviewer yRxJ where we discuss the practical implications of this assumption.**
>
> [BRS19] *Exact recovery in the Ising blockmodel.* Berthet et al., 2019
>
> > W2...asymptotic results may not be relevant at practical scales of $K$.
>
> While our results are stated asymptotically, the underlying mechanism is already finite-$K$: the separation develops as soon as the relevant low-dimensional statistic concentrates. For Thms 1/2, fixing a threshold $u$ strictly between the class-typical magnetization levels (for Thm 1, $u\in(0,m_\star)$, equivalently $M_K^2\ge u^2$; for Thm 2, $|m_0|<u<m_\star$), the same free-energy comparison as in the proofs yields
>
> $R(\tilde g_K)\le (1-\pi)e^{-c_0(u)K}+\pi e^{-c_1(u)K}$
>
> for some $c_0(u),c_1(u)>0$, so $R(g_K^\star)\le R(\tilde g_K)$ decays exponentially in $K$. Hence, in Thm 1,
>
> $R(g_K^{\mathrm{ind}})-R(g_K^\star)\ge \min \\{ \pi,1-\pi \\} -Ce^{-cK},$
>
> since $R(g_K^{\mathrm{ind}})=\min\\{\pi,1-\pi\\}$ exactly for every $K$; and in Thm 2 one similarly obtains
>
> $R(g_K^{\mathrm{ind}})-R(g_K^\star)\ge \pi(1-p)-o(1)-Ce^{-cK},$
>
> where the $o(1)$ term is a finite-$K$ correction in the positive/negative phase weights. For Thm 3, the proof already gives the exact identity
>
> $R(g_K^{\mathrm{ind}})-R(g_K^\star)=\mathbb E\left[|2\eta_K-1| 1\\{g_K^{\mathrm{ind}}\neq g_K^\star\\}\right],$
>
> and, conditional on the latent factor, Hoeffding's inequality gives $Pr(|s_K-s_\infty|>\varepsilon\mid Y,Z)\le 2e^{-2K\varepsilon^2}$; since $\ell^\star$ and $\ell^{\mathrm{ind}}$ are smooth monotone functions of $s$, the finite-$K$ rate is therefore governed by the margin/anti-concentration of the limiting scores, yielding $O(K^{-\alpha/2})$ under $\Pr(|\ell(s_\infty)|\le u)\le Cu^\alpha$, and exponential convergence under a fixed positive margin. A remark is added in revision.
>
> > W3...multiclass extension
>
> The natural extension replaces the binary model by a class-conditional mean-field Potts model for $J_1,\dots,J_K \in [L]$, with sufficient statistic given by $m_K=(m_{1,K},\dots,m_{L,K})$, where $m_{a,K}=K^{-1}\sum_{j=1}^K 1\\{J_j=a\\}$. The same hierarchy persists: CI gives the usual linear score in vote counts, and under a Potts model with interactions shared across classes the interaction terms cancel in pairwise class log-odds, whereas class-dependent interactions yield quadratic scores. The main challenge is that the order parameter is vector-valued on the simplex, removing one-dimensional monotonicity or simple threshold structure. The Potts free energy can have multiple symmetry-related maximizers, making phase coexistence, label-switching, and the control of phase weights substantially more delicate. The key technical step for multiclass dependence-versus-CI separation results is concentration showing that $m_K$ concentrates near class-dependent phase sets while the one-site marginals remain (nearly) identical across classes.
>
> The EM framework extends in a largely analogous manner (E-step via multiclass mean-field approximation to the Potts posterior, and M-step updating class priors, fields, and interaction parameters from expected sufficient statistics), but is more delicate numerically (larger parameter spaces, stronger non-convexity, symmetry between labels induces multiple equivalent optima and can lead to label-switching, and the presence of multiple competing phases can slow convergence or trap EM in suboptimal local maxima) which requires careful careful initialization and regularization.
>
> > W4...missing comparison with crowdsourcing baselines such as Xu et al. (2024)...
>
> In Section C.3, we prove that factor-based methods as in Xu et al. (2024) are approximately equivalent to CI Ising models. In our updated ($K=10$) experiment (see reply to Reviewer XH4L), their method achieves 0.894 (Relevance), 0.756 (Toxicity) and 0.768 (Summarization). This is added in revision.

---

> > ### Author Rebuttal · Reviewer_3EqD · 2026-04-02
> >
> > Thanks for the response. I will keep the acceptance decision.

---

### Decision · Program_Chairs · 2026-04-30

**Decision:**

Accept (regular)

**Comment:**

The paper concerns the problem of aggregating binary judgments from multiple LLM judges. The Authors propose a hierarchy of aggregation models grounded in Ising graphical models, moving from the standard conditional independence (CI) model through a class-independent Ising model to a class-dependent one. The proposed framework models pairwise judge interactions through a coupling matrix and derives Bayes-optimal aggregation rules. This is in contrast to the CI models which incur non-vanishing excess risk. Experiments on three datasets demonstrate consistent improvements over CI baselines.

The Reviewers recognized the practical relevance and motivation of the problem, the strength of theoretical results, and the flexibility of the proposed hierarchical approach. They also raised several concerns regarding the reliance of the separation results on the Curie-Weiss assumption of uniform pairwise coupling, the gap between asymptotic guarantees and the small-K experimental regime, and the experimental section lacking coupling matrix analysis, crowdsourcing baseline comparisons, and ablation studies bridging theory and practice. The Authors' rebuttal substantially narrowed these gaps by providing additional clarifications and updated and extended experiments. All Reviewers except one are leaning towards acceptance. The most critical one finds the dense and hard-to-follow exposition as the main drawback of the paper.

Given the technical depth, soundness of the proposed method, and the exhaustive rebuttal, the paper is slightly above the acceptance bar. If finally accepted, the Authors should revise the manuscript following the suggestions given by the Reviewers.